# Morphinan Alkaloids and Their Transformations: A Historical Perspective of a Century of Opioid Research in Hungary [note 1]

**DOI:** 10.3390/ijms26062736

**Published:** 2025-03-18

**Authors:** János Marton, Paul Cumming, Kenner C. Rice, Joannes T. M. Linders

**Affiliations:** 1ABX Advanced Biochemical Compounds Biomedizinische Forschungsreagenzien GmbH, Heinrich-Glaeser-Strasse 10-14, D-01454 Radeberg, Germany; 2Department of Nuclear Medicine, Bern University Hospital, Freiburgstraße 18, CH-3010 Bern, Switzerland; paul.k.cumming@gmail.com; 3School of Psychology and Counselling, Queensland University of Technology, Brisbane, QLD 4059, Australia; 4Drug Design and Synthesis Section, Molecular Targets and Medications Discovery Branch, Intramural Research Program, NIDA and the NIAAA, NIH, Department of Health and Human Services, 9800 Medical Center Drive, Bethesda, MD 20892, USA; kennerr@mail.nih.gov; 5Johnson & Johnson, Turnhoutseweg 30, B-2340 Beerse, Belgium; jlinders@its.jnj.com

**Keywords:** Alkaloida Chemical Company, azidomorphine, biosynthesis, codeine isomers, desomorphine, epimerization, 6,14-ethenomorphinans, Mitsunobu reaction, morphinan alkaloids, morphine, nomenclature, nucleophilic substitution, stereochemistry, University of Debrecen

## Abstract

The word opium derives from the ancient Greek word ὄπιον (ópion) for the juice of any plant, but today means the air-dried seed capsule latex of *Papaver somniferum*. Alkaloid chemistry began with the isolation of morphine from crude opium by Friedrich Wilhelm Adam Sertürner in 1804. More than a century later, Hungarian pharmacist János Kabay opened new perspectives for the direct isolation of morphine from dry poppy heads and straw without the labor-intensive harvesting of opium. In 2015, Kabay’s life and achievements obtained official recognition as constituting a «Hungarikum», thereby entering the national repository of matters of unique cultural value. To this day, the study of *Papaver* alkaloids is a focus of medicinal chemistry, the (perhaps unstated) aspiration of which is to obtain an opioid with lesser abuse potential and side effects, while retaining good analgesic properties. We begin this review with a brief account of opiate biosynthesis, followed by a detailed presentation of semisynthetic opioids, emphasizing the efforts of the Alkaloida Chemical Company, founded in 1927 by János Kabay, and the morphine alkaloid group of the University of Debrecen.

## Contents

1.Introduction2.Chemistry
2.1.Poppy alkaloids2.2.The stereochemistry of morphinans2.3.Biosynthesis of morphinan alkaloids2.4.The Makleit-Bognár nomenclature2.5.Early syntheses of morphine derivatives with pharmaceutical importance2.6.Reduction of thebaine2.7.Synthesis of desomorphine2.8.Nucleophilic substitution reactions in the morphine series
2.8.1.Reactions of 7,8-dihydro compounds2.8.2.Reactions of Δ^7,8^-unsaturated derivatives2.8.3.Reactions of pseudocodeine tosylate2.8.4.Neopine derivatives2.8.5.Alkyl mesylate and allyl halide structural units in the same molecule2.8.6.Substrates containing double allylic system
2.8.6.1.Substrates with allyl halide and allyl tosylate sub-units2.8.6.2.Substrates containing double allyl halide sub-structural units2.9.Azidomorphinans
2.9.1.Azidomorphine analogues2.9.2.14-Hydroxy-8-azido-8-desoxyallopseudocodeine derivatives2.9.3.Azido derivatives of 6,14-ethenomorphinans2.9.4.6-Azido-6-demethoxythebaine2.10.Fluorinated morphinans
2.10.1.Ring-C fluorinated morphinans2.10.2.1-Fluoro-substituted morphinans2.10.3.Fluorinated 6,14-ethenomorphinans2.11.Application of the Mitsunobu reaction in the morphine series
2.11.1.The first application2.11.2.Preparation of isomorphine and isocodeine derivatives2.11.3.Reactions of codeine isomers and neopine2.11.4.Synthesis of 6β-aminomorphinans2.11.5.Synthesis of 6β-succinimido derivatives2.11.6.Reaction of 14-halogenocodeines2.11.7.Novel applications2.12.Poppy alkaloids as starting materials for molecular imaging2.13.Other semi-synthetic derivatives3.Summary and conclusions

„Az ember ezt, ha egykor ellesi,Vegykonyhájában szintén megteszi. –Te nagy konyhádba helyzéd embered,S elnézed néki, hogy kontárkodik,Kotyvaszt, s magát Istennek képzeli.”“Man will certainly learn this by watchingAnd will simulate it in his kitchen. –You put into your great kitchen your manAnd of his bungling you take no notice,He brews and fancies himself to be God.”Madách Imre: Az ember tragédiájaImre Madách: Tragedy of the man (translation: Tomschey, O.)

## 1. Introduction

The word opium has its origin in the ancient Greek ὄπιον (ópion), which originally referred to the juice of any plant, belying the profound importance of the juice of *Papaver somniferum* specifically in ancient and contemporary medicine. The German apothecary and pioneer of alkaloid chemistry Friedrich Wilhelm Adam Sertürner (1783–1841) began working on opium extraction in 1804, first describing a crude preparation that he designated *principium somniferum* and later morphium, after Morphius, the Greek god of dreams. His isolation of pure morphine (**1**, Figure 1) from raw opium was the first of any alkaloid [1,2]. Throughout history, opium, morphine and its semisynthetic derivatives have been double-edged swords, being indispensable in medicine while also bringing crises of dependency and overdose. The touting of oxycodone (14-hydroxydihydrocodeinone) as an analgesic with a supposedly superior safety profile was surely a factor in the exacerbation of the opioid crisis in some countries [3], now made worse by the availability of synthetic compounds of astonishing potency such as fentanyl and the even stronger nitazines.

The *Alkaloida Chemical Company* (Alkaloida Vegyészeti Gyár) was founded by the Kabay brothers János, Péter and József in Bűdszentmihály (former Tiszabűd and Szentmihály; Tiszavasvári since 1952) in Hungary [4,5,6,7], with the signing of the foundation agreement taking place on 10 September 1926 [8,9]. The principal founder, pharmacist János Kabay (1896–1936), is notable for his 1925 patent [10,11] on the innovative «green method» for directly extracting opiate alkaloids from parts of the poppy plant, thus circumventing the inefficient and labor-intensive process of opium harvesting by the collection of the resin [10,11]. Further improvements in 1931 enabled the commercially viable extraction of the key alkaloids, morphine (**1a**) and an alkaloid mixture of mainly codeine (**2a**), and the phthalideisoquinoline derivative narcotine (**8**) from the previously worthless straw and dried capsules of the poppy plant («dry method») [10,11]. As the foundation of the Hungarian morphine alkaloid industry, Kabay’s process was adopted the world over [12,13,14].

*Kossuth Lajos University* (Kossuth Lajos Tudományegyetem, KLTE; today the University of Debrecen, Hungary) established its Institute of Organic Chemistry in 1947. The early 1950s saw a strong initiative by the Hungarian Heavy Industries and Education Ministries to form regionally important connections between new industrial centers and nearby universities. The long-standing collaboration between the Institute of Organic Chemistry of KLTE and Alkaloida arose as a direct result of these efforts. Rezső Bognár (1913–1990) [15], a former student of Emil Fischer’s (1852–1919) associate Géza Zemplén (1883–1956), received in 1950 the appointment as head of the Institute of Organic Chemistry. Early in his tenure, Bognár placed special emphasis on joint efforts with Alkaloida specialists to optimize the Kabay technology, aiming to obtain higher yields of the valuable opiate alkaloids codeine, thebaine, narceine, neopine, and narcotoline [16,17,18,19,20]. During the period of 1979–1995, Sándor Makleit (1930–2012) became head of the Department of Organic Chemistry and its morphine alkaloid research group, in which capacity he coordinated interactions between KLTE and Alkaloida. Upon Makleit’s retirement, Sándor Berényi (1947–2019) redirected research toward an emphasis on aporphine chemistry via the rearrangement of morphinans. The contributions of László Szilágyi (NMR), Gyula Batta (NMR) and Zoltán Dinya (IR, MS) were essential in the field of structural analysis of the new synthetic morphinan derivatives. Sándor Hosztafi (1952–), formerly of the Alkaloida Chemical Company, has contributed enormously to the field of semisynthetic morphinans, aporphine derivatives and other poppy alkaloids, and remains active at the Department of Pharmaceutical Chemistry of Semmelweis University, Budapest.

In this account, we cannot omit mentioning the long-standing cooperation between the Debrecen morphine alkaloid research group and the Department of Pharmacology of Semmelweis University (Budapest) for the pharmacological analysis of newly synthesized opioid receptor (OR) ligands. József Knoll (1925–2018) and Zsuzsanna Fürst (1939–) first directed these investigations. The biochemical characterization of new semisynthetic alkaloids was performed under the guidance of Anna Borsodi, Sándor Benyhe and Géza Tóth at the Biological Research Center of the Hungarian Academy of Sciences (MTA Szegedi Biológiai Központ, Biokémiai Intézet, Szeged), and at Gedeon Richter Pharmaceuticals (Richter Gedeon Vegyészeti Gyár, Budapest).

Given this historical and personal background, we now present a comprehensive account of opiate alkaloid chemistry, placing our focus on the extensive contributions in recent decades of the Hungarian research group in morphine alkaloids in Debrecen and of the Alkaloida Chemical Company in the field of morphinan syntheses, as part of a concerted search for novel structures and synthetic approaches. We begin with a brief presentation of opiate nomenclature, stereochemistry and the biosynthesis of morphinan alkaloids.

## 2. Chemistry

### 2.1. Poppy Alkaloids

The diverse alkaloids of *Papaver somniferum* are categorized based on their chemical structures [21,22,23,24,25]. The most well known of these are benzylisoquinoline-(I), morphinan-(II), aporphine-(III), benzo[c]phenantridine-(IV), papaverrubine-(V), narceine-typ-(VI), protoberberine-(VII), protopine-(VIII) and phthalideisoquinoline (IX), with ground scaffolds presented in Figure 2.

### 2.2. The Stereochemistry of Morphinans

The morphinan scaffold contains three asymmetric centers (9,13,14). The stereochemistry of morphinan derivatives and other types of OR ligands is of fundamental importance in determining their pharmacological selectivity between the OR sub-types (μ, δ, κ, NOP) or off-target binding to other receptor types, e.g., NMDA receptors. For example, levorphan (*N*^17^-methyl-3-hydroxy-morphinan), which has an *R* absolute configuration at all three asymmetric carbons (9*R*,13*R*,14*R*, Figure 3, structure IIa), is an OR agonist with seven-fold higher affinity than morphine (**1a**). Dextrorphan (*N*^17^-methyl-3-hydroxy-morphinan) with a 9*S*,13*S*,14*S* absolute configuration has a different pharmacological profile. It has selectivity for the binding site of NMDA glutamate receptors. Dextromethorphan ((9*S*,13*S*,14*S*)-*N*^17^-methyl-3-methoxy-morphinan) has little analgesic activity at μORs, but is a commonly used cough suppressant [23]. There are also semisynthetic 6,14-ethenomorphinans with up to seven asymmetric carbons (5,6,7,9,13,14,20), with the absolute configuration of the C-20 chiral carbon being of particular importance for OR affinity and selectivity. While the 20*S* version of etorphine has about 40 times higher potency than morphine (**1a**), 20*R*-etorphine surpasses its diastereomer by far, with its potency being 3200-fold that of morphine (**1a**).

There are two theoretical sterical arrangements in the case of decalin and morphinans (Figure 3), depending on the ring-junction of the A/B-ring and B/C-ring, respectively [23]. For *cis*-decalins, the H-1 and H-6 hydrogens, and for cis-morphinans, H-14 and the C^13^–C^15^ bond, are on the same face. In *trans*-decalins, the H-1 and H-6 hydrogens, and in B/C-*trans*-morphinans, H-14 and the C^13^-C^15^ bond, are on the opposite face. In naturally occurring morphinan alkaloids, the junction of the B- and C-rings is a B/C-*cis* fusion analogous to *cis*-decalin, as shown in structure **IIa** of Figure 3, where **IIb** presents the structure of a B/C-*trans*-morphinan. It is not possible to convert a morphinan (B/C-*cis*-morphinan) into an isomorphinan (B/C-*trans*-morphinan) by the rotation of molecular parts without breaking chemical bonds. The absolute configurations of the C-14 chiral carbon differ for the isomers: 14*R* (morphinan) and 14*S* (isomorphinan).

Morphine (**1a**, Figure 4) and a significant number of its derivatives have a T-shaped three-dimensional molecular geometry. The isomers of morphine (**1a**, Figure 4, Table 1) and codeine (**2a**) have five chiral carbons: positions 5, 6, 9, 13 and 14. In **1a** and **2a**, the carbon–carbon double bond is situated in position C7–C8 (Δ^7,8^), the ring-C has a flattened boat conformation, and the allylic hydroxyl group in position C-6 is pseudo-equatorial. The C-6 in **1a** and **2a** has absolute configuration *S*. In the C-6 epimer molecules isomorphine (**1b**, α-isomorphine, Figure 4) and isocodeine (**2b**), the 6-hydroxyl group is located above the plane of the ring-C in a pseudo-axial position. The C-6 in **1b** and **2b** has absolute configuration *R*.

Further Δ^6,7^ isomers for morphine (**1a**) and codeine (**2a**) are known as allopseudomorphine (**1c**, also known as β-isomorphine), γ-isomorphine (**1d**), allopseudocodeine (**2c**) and pseudocodeine (**2d**). In these compounds, the alcoholic hydroxyl group connects to position-8 of the ring-C. Of the ring-C carbons, C-5, C-6, C-7, C-8 and C-13 are in the same plane, while C-14 is located above them. The ring-C of the compounds has an envelope conformation. In allopseudomorphine (**1c**) and allopseudocodeine (**2c**), the 8-OH is pseudo-axial and the configuration of C-8 is *R*. The two isomers with a Δ^6,7^ double bond and pseudo-equatorial 8-hydroxyl group, γ-isomorphine (**1d**) and pseudocodeine (**2d**), have an 8*S* absolute configuration. Figure 4 and Table 1 provide an overview of the structure and absolute configuration of the parent compounds.

The neopine (**3a**), isoneopine (**3b**), neomorphine (**3c**) and neoisomorphine (**3d**) derivatives also have a Δ^8,14^-unsaturated bond in ring-C. These compounds have only four asymmetric carbon atoms (5, 6, 9 and 13), since the Δ^8,14^ double bond C-14 is no longer asymmetric. The C-ring is rigid and of a half-chair conformation. In the cases of neopine (**3a**) and neomorphine (**3c**), the 6-hydroxy group is in a pseudo-axial position and the carbon-6 has absolute configuration *S*. In compounds of the 6-beta series, like isoneopine (**3b**) and isoneomorphine (**3d**), the 6-OH is pseudo-equatorial and the carbon-6 has configuration *R*. The poppy alkaloids with a conjugated 6,8-dien system in ring-C—thebaine (**4**) and oripavine (**5**)—have only three asymmetric carbons, with the absolute configurations 5*R*, 9*R* and 13*S*.

### 2.3. Biosynthesis of Morphinan Alkaloids

A suite of five *O*-methyltransferases and two *N*-methyltransferase enzymes contribute to benzyisoquioline alkaloid (BIA) biosynthesis [26] in five genera of flowering plants, including *Papaveraceae* (poppy) and *Ranunculaceae* (buttercup). The phytochemical synthesis of morphine alkaloids begins with *L*-tyrosine (**10**) via a branching pathway that yields, on one side, tyramine (**11**) through the action of *L*-tyrosine decarboxylase, which is a dimeric pyridoxal-phosphate-dependent enzyme [27]. A phenolase enzyme then transforms the tyramine product (**11**) to dopamine (**13**). In the right-hand branch, *L*-tyrosine (**10**) gives rise to the keto acid 4-hydroxyphenylpyruvate (**12**) via *L*-tyrosine deaminase [28] and then to the corresponding acetaldehyde (**14**) via a specific decarboxylase. The condensation of the two products (dopamine (**13**) and 4-hydroxyphenylacetaldehyde (**14**)) to form (*S*)-norclaurine (**15**) proceeds via an asymmetric Pictet–Spengler condensation, which is catalyzed by the enzyme norclaurine synthase [29]. Mechanistic studies of the norcoclaurine synthase enzyme derived from *Thalictum flavium* (yellow meadow-rue) indicated a two-step cyclization of the putative iminium ion intermediate, which constitutes the first committed step in the formation of poppy BIAs.

Once formed, (*S*)-norclaurine (**15**, Figure 5) undergoes a series of enzymatic methylations, first by norcoclaurin-6-*O*-methyltransferase in the presence of the co-substrate *S*-adenosylmethionine (SAM), which represents a key rate-limiting step in benzyisoquioline synthesis [30], followed by tetrahydrobenzylizochinoline-*N*-methyltransferase. The product (*S*)-*N*-methyl-coclaurine (**17**) undergoes ring hydroxylation by a phenolase enzyme or a higher affinity P450-dependent monooxygenase (CYP80B1) in poppy [31] and then a further SAM-dependent *O*-methyltransferase reaction to give (*S*)-reticulin ((*S*)-**19**). Interestingly, reticulin in the traditional food *Annona muricata* (soursop) may be responsible for dopamine neuron degeneration resulting in a form of Parkinsonism that is endemic to the island of Guadeloupe [32].

A specific NADPH-dependent reductase isomerizes (*S*)-reticuline ((*S*)-**19**) to (*R*)-reticuline ((*R*)-**19**). After ring closure of (*R*)-reticuline ((*R*)-**19**, Figure 6) via an NADPH/O_2_-dependent enzyme yielding salutidarine (**7**), an NADPH/NADP^⊕^-dependent oxidoreductase yields (7*S*)-salutaridinol (**21**), which is then ring-acetylated by an acetyl coenzyme A-dependent enzyme; spontaneous deacetylation gives thebaine (**4**, paramorphine), which gains its common name from the ancient Greco-Egyptian city of Thebes. In rat brain preparations, thebaine (**4**) has only weak binding affinity for ORs [33] and, unlike the synthetic enantiomer (+)-thebaine [34], has little analgesic potency. Having been long regarded as a useless side product of poppy extraction, thebaine (**4**) now has an annual global market value of USD 1 billion, serving as starting material in the industrial production of opiate agonists (e.g., oxycodone, oxymorphone), antagonists (Nal compounds: naloxone, naltrexone, nalbuphine) and Diels–Alder type Bentley compounds (buprenorphine, etorphine, diprenorphine). ORs in the brain present important targets for molecular imaging by positron emission tomography (PET), as presented in an extensive review [35]. In particular, thebaine (**4**) is a precursor for the radiosynthesis of various PET tracers for OR imaging (see Section 2.12). It has been known since the 1970s that thebaine (**4**) predominates in *Papaver bracteatum* (the Armenian poppy), which contains only a small amount of other alkaloids [36,37]. Alternatively, Zenk and researchers at Tasmanian Alkaloids obtained the morphine (**1a**)-free *Papaver somniferum* plant designated as top1 (top: **t**hebaine–**o**ripavine–**p**oppy) through genetic modification [38]. The genetic cultivar contains predominantly thebaine (**4**, 1.65%) and oripavine (**5**, 0.43%) but no morphine (**1a**, 0%) or codeine (**2a**, 0%).

Thebaine (**4**, Figure 7) yields morphine (**1a**) by two parallel pathways in the opium poppy. In one pathway, an enol ether hydrolase enzyme forms neopinone (**23**), which equilibrates to codeinone (**24**), an alkaloid with about one-third the analgesic potency of codeine (**2a**). An NADPH-dependent reductase converts codeinone (**24**) to codeine (**2a**), which is demethylated to yield the still more potently analgesic morphine (**1a**). In the other pathway, 3-*O*-demethylation of thebaine (**4**) yields oripavine (**5**), which is also a potent analgesic, albeit with high toxicity. Oripavine (**5**) is important industrially as a precursor for synthetic opiates (i.e., etorphine and buprenorphine); in the poppy plant, oripavine (**5**) undergoes enol ether hydrolysis to yield morphinone (**25**), which has similar analgesic potency to codeine (**2a**). The conversion of morphinone (**25**) to morphine (**1a**) entails an NADPH-dependent codeinone reductase, various isoforms of which also catalyze the reduction of certain other intermediates of morphine (**1a**) biosynthesis [39]. Interestingly, morphine biosynthesis in the poppy plant entails compartmentation in several cell types [40,41]; the initial steps occur in sieve elements of the phloem, whereas the conversion of salutaridine (**7**) to thebaine (**4**) predominantly takes place in adjacent laticifers, where the main morphine accumulation occurs.

Raw opium is the dried latex from seedpods of the opium poppy, which naturally contains a mixture of thebaine alkaloids and intermediates, the composite of which contributes to its somnorific/analgesic effects and toxicity. The principle alkaloid in opium samples from South East Asia was morphine (11–23%), with lesser amounts of codeine (2–4%) and thebaine (1–3%) [42]. Poppy straw from various *Papaver somniferum* cultivars used in oil seed production in the Czech Republic had relatively invariant alkaloid concentrations across three consecutive harvests [43]. Cluster analysis distinguished cultivars with high morphine concentration (1.6%) and relatively low levels of other alkaloids from cultivars with the opposite relationships. Comestible poppy seeds also contain various alkaloids, extending over a >100-fold concentration range, depending on the source [44]. Baking substantially reduces the alkaloid content of poppy seeds, such that it would be practically impossible to experience intoxication from Bejgli, a traditional Hungarian Christmas cake, although there are case reports of individuals with opioid dependence consuming poppy seeds by the kilogram [45]. On the other hand, there are reports of lethal codeine (**2a**)/morphine (**1a**) overdose among poppy seed tea drinkers [46], and in drinkers of tea made with «home-grown» dried seedpods [47]. Thebaine (**4**) is a pro-convulsant at high doses, and poppy seeds with a particularly high content of thebaine (**4**, Figure 8) caused serious neuromuscular toxicity in tea drinkers in Australia [48]. Thus, the toxicity of poppy products depends upon factors such as the particular cultivar, growing conditions and mode of preparation.

ORs [49] regulate numerous (patho)physiological processes, being involved in pain modulation, euphoria, reward behaviors and substance abuse, with involvement in the pathophysiology of various psychiatric disorders, epilepsy and neurodegenerative conditions such Alzheimer’s disease (AD) [50]. Endogenous opioid peptide ligands and exogenous opioids activate several types of G protein-coupled (GPCR) receptors (μ-OR, δ-OR κ-OR, NOP). The concept of «biased agonism» underlies the search for OR ligands eliciting therapeutic effects but with a lesser risk of toxicity and dependence [51,52]. In general, OR agonists inhibit the enzyme adenyl cyclase (AC), thus decreasing the intracellular cAMP level and thereby inhibiting the voltage-gated Ca^2⊕^ ion channels while activating K^⊕^ efflux. The resulting hyperpolarization inhibits the presynaptic release of neurotransmitters (e.g., dopamine (DA), norepinephrine (NE), acetylcholine (ACh), γ-aminobutyric acid (GABA)), which produce a variety of effects including sedation, central analgesia and euphoria.

Morphine (**1a**) yields a number of metabolites in humans by various metabolic pathways [53]. The most important morphine metabolites are M3G (**26**, Figure 9) and M6G (**27**), which form by glucuronidation catalyzed by hepatic uridine 5′-diphosphoglucuronyltransferase (UGT). The sulfation of **1a** by hepatic sulfotransferase (SULT1A3) results in morphine-3-sulfate (**30**, M3S) and morphine-6-sulfate (**28**, M6S). The *N*^17^-demethylation of **1a** by cytochrome P450 (CYP450) gives normorphine (**29**), and oxidative dimerization leads to pseudomorphine (**6**, 2,2′-bimorphine). The SAM-dependent 3-*O*-methylation of morphine (**1a**) to codeine (**2a**) can occur in the liver, lung or kidney.

The chemical structures of some phase I and phase II metabolites of morphine are depicted in Figure 9.

### 2.4. The Makleit–Bognár Nomenclature

In 1968, Makleit and Bognár [54,55] proposed the introduction of a new nomenclature for morphine derivatives. According to this nomenclature, the naming of all ring-C-substituted semisynthetic compounds derives from the isomers of codeine (**2a**–**d**, Figure 4 and Figure 10). The Makleit–Bognár nomenclature requires only prior knowledge of the names and stereochemistry of the basic four codeine isomers: codeine (**2a**), isocodeine (**2b**), allopseudocodeine (**2c**) and pseudocodeine (**2d**).

Makleit and Bognár suggested abandoning the previously used names for morphine isomers, α-(**1b**), β-(**1c**) and γ-isomorphine (**1d**), in favor of isomorphine (**1b**), allopseudomorphine (**1c**) and pseudomorphine (**1d**), all in analogy to the codeine isomer nomenclature. They also proposed the adoption of the rational name “2,2′-bimorphine” to replace the old name of the poppy alkaloid bis-derivative pseudomorphine (**6**, Figure 1). This simplified nomenclature is generalizable to all ring-C-substituted derivatives and faithfully reflects their structure and stereochemistry. Ergo, chemists can ascertain quickly and precisely the name and exact structure of any new derivatives, knowing the names and structures of the four codeine isomers (**2a**–**d**, Figure 4 and Figure 10). We note that the position of a substituent below and above the plane ring-C differs for C-6 and C-8; if a C-6 substituent is pseudo-axial, it lies above the plane of the ring-C, while a C-8 pseudo-axial substituent is below the ring-C.

The term “deoxy-X” is a proposed term for all ring-C-modified derivatives, rendering redundant the descriptions (6α, 6β, 8α, 8β), as the name already specifies the exact structure of the compound. For the interpretation of this nomenclature, we present numerous examples in Figure 11. Like the Maat nomenclature system [56] in the field of the Diels–Alder adducts of morphinan-6,8 dienes [57], chemists have not universally adopted the rational Makleit–Bognár nomenclature [54,55], despite its considerable advantages.

### 2.5. Early Syntheses of Morphine Derivatives with Pharmaceutical Importance

Opiates are plant products formed by a series of enzymatic reactions as outlined above, whereas chemistry has built upon the rather sparse selection of poppy alkaloid scaffolds to generate countless derivatives. After the elaboration of the Kabay–Bognár technology for the isolation of morphine (**1a**) and its accompanying alkaloids from dry poppy straw and heads [16], the transformation of morphine (**1a**) to other active pharmaceutical ingredients came into focus at Alkaloida and the University of Debrecen.

Morphine (**1a**) was converted to codeine (**2a**) by methylation of the phenolic 3-hydroxyl group by aryl trimethylammonium hydroxides (Rodionov’s method [58,59]). Methods were elaborated for the synthesis of benzylmorphine (**32**, 7,8-didehydro-4,5α-epoxy-6α-hydroxy-17-methyl-3-phenylmethoxy-morphinan), ethylmorphine (**33**, 7,8-didehydro-4,5α-epoxy-6α-hydroxy-17-methyl-3-ethoxy-morphinan) and pholcodine (**34**, 7,8-didehydro-4,5-epoxy-6α-hydroxy-17-methyl-3-(2-(4-morpholinyl)ethoxy)-morphinan) [60] on an industrial scale starting from morphine (**1a**, Figure 12). The latter compound (**34**) became clinically significant because of its lesser toxicity and proclivity to induce respiratory depression in comparison to morphine (**1a**), and due to its more favorable sedative effect as compared to codeine (**2a**) or other morphine derivatives. The application of pholcodine (**34**) as an antitussive agent in the treatment of dry non-productive cough for young patients and adults has been widespread, and **34** was for a long time the second-most important active pharmaceutical ingredient (API) manufactured from morphine (**1a**) [14]. Neuromuscular blocking agents (NMBAs) are among the leading causes of life-threatening drug-induced perioperative anaphylaxis. Novel studies performed by Mertes et al. [61] confirmed a remarkable relationship between the application of pholcodine (**34**) and other quaternary ammonium derivatives and the risk for NMBA-related perioperative anaphylaxis mediated by immunoglobulin E (IgE). This association led to the withdrawal of pholcodine-containing antitussive syrups and cold medicines from the EU and UK markets [62].

Dihydromorphine (**35**, paramorphan) was synthesized by the heterogeneous catalytic hydrogenation of the Δ^7,8^ double bond of morphine (**1a**) in dilute acetic acid. Dihydromorphinone (**36**, dilaudid) was prepared either by the catalytic rearrangement of **1** or by the Oppenauer oxidation of dihydromorphine (**35**). Szabó and Bognár from KLTE developed a procedure for the preparation and purification [63] of the OR antagonist *N*^17^-allyl-normorphine (**37**, nalorphine). Synthesis starts from morphine (**1a**, route [64]: **1** → diacetylmorphine → diacetyl-*N*^17^-cyano-normorphine → normorphine → **37**) using (among other approaches) the von Braun method [65] for the *N*-demethylation step. Apomorphine (**39**), a dopamine receptor agonist that can relieve motor symptoms of Parkinson’s disease, was synthesized from morphine (**1a**) by treatment with concentrated hydrochloric acid. Berényi et al. [66,67] later investigated in detail the mechanism of the morphine (**1a**) → apomorphine (**39**) rearrangement. For details of the desomorphine (**38**) synthesis [68,69,70], see Section 2.7.

### 2.6. Reduction of Thebaine

Thebaine (**4**) comprises a 6,8-diene and an enol ether function in ring-C, which together correspond to a 1-methoxy-1,3-cyclohexadiene sub-structural unit [22,23]. The B- and C-rings together form a *trans*-3,4,8,9-tetraline system, while the C- and D-rings constitute a *trans*-1,2,3,4,5,10-hexahydro-isoquinoline subunit. An easily attacked part of the molecule is the C^5^-O bond of the dihydrofuran-ring-E, which is part of the C^4^-O-C^5^-C^6^=C^7^ allyl ether system. The characteristics of these sub-structures enable a large variety of reactions for the baine (**4**) in a manner sensitive to reaction conditions. This is particularly apparent in its reduction with various reagents or during catalytic hydrogenation under different reaction conditions, which results in a diverse array of products (Figure 13).

Indeed, the targeted reduction of thebaine is a very challenging process [71,72]. Neither the application of chemical reducing agents nor heterogeneous catalytic hydrogenation results in a sole product. In the first case, reduction in basic or acidic media yields products containing a phenolic hydroxyl group due to the opening of the E-ring. The treatment of thebaine (**4**) with sodium in hot ethanol results in dihydrothebaine-Φ (**43**, Δ^5,8^-phenolic dihydrothebaine) as the main product [73,74]. Bentley et al. [75,76] found that the application of sodium–liquid ammonia gave dihydrothebaine-Φ (**43**) in 90% yield and a small amount of β-dihydrothebaine (**40**) [77], along with approximately 10% Δ^6,8^-phenolic-dihydrothebaine (**40**). Razdan et al. [77] reported that the treatment of thebaine (**4**) with an excess of potassium in liquid ammonia gave a 1:1 mixture of **40** and **43** in 95% yield, from which **40** could be isolated in 34% overall crystalline yield. The treatment of **43** with K/liquid ammonia in the presence of Fe(NO_3_)_3_ produced a 1:1 mixture of **40** and **43** in 79% yield. When performing reduction with LiAlH_4_, in benzene–diethyl ether (or THF) [78,79], β-dihydrothebanie (**40**) was obtained in low (22–27%) yield. The third isomer Δ^5,7^-phenolic dihydrothebaine (**45**) can be prepared from codeinemethylether (**44**) by treatment with sodium ethylate [71]. The reduction of thebaine (**4**) with stannous chloride (SnCl_2_) gave thebainone-A (**41**), or metathebainone (**42**) via the rearrangement of the carbon scaffold. The yield of the reduction of thebaine (**4**) to **40** with the intact conjugated 6,8-diene system was greatly improved by the findings of Linders et al. [80], whereby reduction with zinc under alkaline conditions resulted in almost quantitative conversion to dihydrothebaine (**40**).

For the formation of metathebainone (**42**) and thebainone-A (**41**), Robinson [81,82] suggested the following reaction mechanism (Figure 14): (i) the enol ether group of the hydrochloric salt of thebaine undergoes hydrolysis and an unstable addition product (**X**) is formed via proton attack on the oxygen of the E-ring; (ii) rearrangement—heterolysis of the C^5^-O bond, electron shift and the anionotropic rearrangement of the ethanamine side-chain from C^13^ to C^14^ (**XI**); (iii) intermediate (**XI**) stabilizes via proton losing/transmission to (**XII**); and (iv) the formation of metathebainone (**42**) via the saturation of the easily accessible Δ^7,8^ double bond. The experimental conditions leading to the formation of thebainone-A (**41**) were as follows: (v) the unstable intermediate (**X**) transforms to (**XIII**) via the heterolysis of the C^5^-O bond; (vi) the formation of (**XIV**) through the uptake of H^⊕^ and 2e^⊖^; and (vii) the ketonization of (**XIV**) through a 1,4-mechanism to thebainone-A (**41**).

As noted above, thebaine (**4**) is a non-analgesic neuromuscular toxin, long regarded as a nuisance side-product of poppy extraction. However, its conversion to pharmacologically useful semisynthetic opiates has since become the objective of concerted efforts. The catalytic hydrogenation of thebaine (**4**) results in a complex mixture of products depending on the reaction conditions (e.g., pH of the solution, type of catalyst) [83]. When the hydrogenation was interrupted after the disapperance of the starting material thebaine (**4**) (from the reaction mixture) but before cessation of hydrogen uptake neopine-6-methylether (**46**) was isolated as its quaternary methiodide salt in addition to dihydothebaine (**48**), dihydrothebainone (**52**) and tetrahydrothebaine (**50**) [72]. Interestingly, the hydrolysis of dihydrothebaine-Φ (**43**) with KHSO_4_ resulted in thebainone-A (**41**, B/C-*cis*-series) and β-thebainone-A (**47**, B/C-*trans*-series).

The main driving force of these investigations in mid-20th century Hungary was to study the possibilities for the transformation of thebaine (**4**) via dihydrothebaine (**48**) to dihydrocodeinone (**49**, hydrocodone), an API (active pharmaceutical ingredient) used for the treatment of moderate-to-severe pain and as a cough suppressant. Szabó and Bognár [84,85] performed detailed research into the catalytic reduction of thebaine (**4**) as a function of reaction conditions (e.g., type of catalyst, pressure, concentration, temperature, pH, rate of hydrogen uptake). They isolated from the product mixtures a large number of morphinan derivatives, e.g., dihydrothebaine (**48**), dihydrocodeinone (**49**), dihydrothebainone (**52**), β-dihydrothebainone, etc. (see also Figure 15) [84,85]. For the characterization of the obtained products, Szabó and Bognár undertook a careful analysis of their infrared spectra and photometry combined with paper chromatography. This enabled them to detect more than twenty morphinans in the product mixture. They also performed their syntheses through independent reaction routes, and reported in detail the mechanisms of product formation [84,85].

The occurrence of parallel and consecutive side reactions during the heterogenous catalytic hydrogenation of thebaine (**4**) is explicable by the following processes (Figure 15):(A)1,2-Addition of hydrogen to the Δ^7,8^ double bond of thebaine (**4**). This results in dihydrothebaine (**48**), which is transformable to dihydrocodeinone (**49**) through hydrolysis;(B)1,4-Addition of hydrogen to the conjugated system of thebaine (**4**) in ring-C. This leads to codeine methyl ether (**44**), which is reducible to tetrahydrothebaine (**50**);(C)1,4-Addition of hydrogen to the oxygen atom of the E-ring and to C-7. This constitutes a hydrogenolysis of the allyl ether group, which results in the opening of the ether bridge. In the first step, Δ^5,8^-phenolic-dihydrothebaine (**43**, dihydrothebaine-Φ) is formed, which can be transformed to dihydrothebainone-Δ^5^-methylenolate (**51**). The enol ether hydrolysis of this compound results in dihydrothebainone (**52**). Subsequent hydrogenation leads to dihydrothebainol-6-methyl ether (**53**);(D)1,6-Addition of hydrogen to the oxygen atom of the E-ring and to C-14. This process results in Δ^5,7^-phenolic-dihydrothebaine (**45**). The following/further hydrogenation of the latter compound forms dihydrothebainone-Δ^5^-methyl enolate (**51**) and dihydrothebainol-6-methyl ether (**53**).

### 2.7. Synthesis of Desomorphine

Very recently, the chemistry, synthesis and pharmacology of desomorphine (**38**) were reviewed in considerable detail [87,88,89]. One can deduce the structure of 6-desoxymorphine derivatives from C-6 hydroxy compounds by the formal elimination of the hydroxy group or from 6-ketomorphinans by reduction. Due to their increased lipophilicity (and consequently greater brain penetration), these derivatives have higher analgesic potency compared to OR ligands with hydroxy and keto groups in position-6. This suggests that a substitution in position-6 is not essential for a high μOR affinity. Indeed, Rapoport and Bonner reported in 1951 [90] that Δ^7^-desoxymorphine (**57**, 6-desoxymorphine, desomorphine-E; for binding affinity, see [91]: K_i_ (μOR) = 2.9 nM, K_i_ (δOR) = 11.8 nM, K_i_ (κOR) = 44.5 nM, δ/μ = 4, κ/μ = 16) prepared from Δ^7^-desoxycodeine (for binding affinity, see [91]: K_i_ (μOR) = 305 nM, K_i_ (δOR) = 4520 nM, K_i_ (κOR) = 3090 nM, δ/μ = 15, κ/μ = 10) by 3-*O*-demethylation with pyridine hydrochloride at 220 °C (6 min, N_2_ atmosphere) had ~8.5-fold higher analgesic potency than morphine (**1a**), consistent with the much higher μ-OR affinity. Comparing the analgesic properties of various morphine derivatives (mouse hot-plate test), Eddy et al. [92,93] found the following order of relative potencies (in brackets): desomorphine (11.7) > dihydromorphinone (7.0) > dihydromorphine (1.2) > morphine (1.0) > dihydrodesoxycodeine (0.72) > dihydrocodeinone (0.66) > dihydrocodeine (0.17).

6-Desoxydihydromorphine (**38**, 4,5α-epoxy-17-methylmorphinan-3-ol, also known as dihydrodesoxymorphine-D, and desomorphine) differs structurally from morphine (**1a**) with respect to the absence of the alcoholic hydroxyl group in position C-6α and the saturation of the Δ^7,8^ double bond in ring-C. Desomorphine (**38**) and morphine (**1a**) both have a T-shaped 3D structure, although in **38** the ring-C has a chair conformation, while that in **1a** has a boat conformation. Desomorphine (**38**) is about 8–10 times more potent than morphine (**1a**) in the mouse hot-plate test [94] (ED_50_ (**38**) = 0.14 μmol/kg, ED_50_ (**1a**) = 3.3 μmol/kg).

Furthermore, desomorphine (**38**) has a rapid onset and shorter duration of analgesic action and is accompanied with higher toxicity (rat LD_50_ (**38**) = 27 mg/kg, LD_50_ (**1a**) = 226–318 mg/kg [95]). Desomorphine (**38**) is apparently a high-affinity μ-OR agonist, as evident by its potent analgesic effects [96], but its sub-type selectivity is unknown. The synthesis of **38** was already described in the 1930s by Small and colleagues in the course of their search for new analgesics with a lesser addictive pharmacological profile compared to morphine (**1a**) [97,98]. In the 1950s, desomorphine (**38**) served as an API under the trade names Desomorphine^®®^ and Permonid^®®^ (Hoffmann-La Roche, Little Falls, NJ, USA) in the USA and in Switzerland. The early syntheses of **38** were performed from morphine (**1a**), codeine (**2a**), dihydromorphine (**35**) or desoxymorphine-C (a Δ^6,7^-didehydro compound: 17-methyl-4,5α-epoxy-3-hydroxy-6,7-didehydro-morphinan). The 6β-chloro derivatives α-chlorocodide (**59b**, Figure 16), α-chloromorphide or α-chlorodihydromorphide were prepared with thionyl chloride (SOCl_2_) from the corresponding starting morphinan (**1a**, **2a**, **35**) (note: α- and β-descripts are not sterical but positional: α: 6-substituted and β: 8-substituted). Thereafter, the respective halogenocodides (e.g., α-chlorocodide (**59b**) or β-chlorocodide (**65a**), Figure 17) or halogenomorphides were subjected to catalytic hydrogenation [98] for the reduction of the double bond in ring-C and dehalogenation. These reactions were performed in methanol or in diluted hydrochloric acid in the presence of Pd/BaSO_4_. Depending upon the character of the substrate and the applied reaction and solvent conditions, these reactions gave desomorphine (**38**) in a yield of ~30–83%.

In 1956, Bognár and Makleit [68,69,70] developed four-step procedures for the preparation of dihydro-6-desoxymorphine (**38**) from morphine (**1a**, Figure 16, route A: **1a** → **35** → **54** → **58** → **38**; route B: **1a** → **55** → **54** → **58** → **38**). They first synthesized 3-*O*-acetyl-dihydromorphine (**54**) from morphine (**1a**) via dihydromorphine (**35**) or via 3-*O*-acetylmorphine (**55**). They performed partial acetylation by the method of Welsh [99] with acetic anhydride in the presence of an aqueous NaHCO_3_ solution. The Δ^7,8^ double bond of 3-*O*-acetylmorphine (**55**) was saturated (H_2_, Pd-C) in aqueous acidic media (HCl, pH 6–6.5). 3-*O*-Acetyl-dihydromorphine (**54**) was converted with *p*-toluene-sulfonyl chloride in pyridine to 3-*O*-acetyl-dihydromorphine tosylate (**58**), which was then treated with LiAlH_4_ according to the method of Schmid and Karrer [100] in THF. This hydrogenolyzed (hydrogen-substituted) the tosyloxy group in position-6, in concert with 3-*O*-deacetylation, giving the desired 6-desoxydihydromorphine (**38**) in an overall yield of 7.7%.

In 1975, Makleit et al. [101] published a novel route for the preparation of desomorphine (**38**) starting from morphine (route: **1a** → 3-*O*-acetylmorphine (**55**) → 3-*O*-acetylmorphine tosylate (**56**) → desoxymorphine-E (**57**) → **38**). The authors concluded that with the lithium aluminum hydride reduction of 3-*O*-acetylmorphine tosylate (**56**), the presence of the allylic system in ring-C facilitates the elimination reaction [101], thus giving higher yield compared to the analogous reaction of 3-*O*-acetyldihydromorphine tosylate (**58**) [68,69,70]. Desoxymorphine-E (**57**, Δ^7,8^-didehydro) was converted with diazomethane to desoxycodeine-E (**61**, Δ^7,8^-didehydro) or by catalytic reduction (H_2_, PtO_2_, MeOH, 1 h) quantitatively to desoxydihydromorphine-D (**38**).

In 2012, the research group of Chen [102] described a novel protocol for the synthesis of desomorphine (**38**) from codeine (**2a**, route: **2a** → **60** → **61** → **62** → **38**). Codeine mesylate (**60**) was treated with 2 equiv. of LiAlH_4_ in THF (0 °C, 30 min and RT for 1 h) to yield desoxycodeine-E (**61**) in 93% yield. The latter compound (**61**) was hydrogenated (H_2_, PtO_2_, MeOH, 4 Bar, 1 h) to dihydodesoxycodeine-D (**62**) in quantitative yield. The 3-*O*-demethylation of **62** was achieved by the application of boron tribromide in dichloromethane to yield **38** in 43% yield. The overall yield of the synthesis of **38** was 38% starting from codeine (**2a**). The authors also performed the synthesis of a deuterium-labeled version of desomorphine staring from dihydrodesoxycodeine-D in a three-step procedure.

Tragically, desomorphine (**38**) is the main active component [103] of the illicit designer drug krokodil (street name, from Russian word for crocodile: кpoкoдил, also known as Russian magic, krok/crok, flesh-eating heroin, poor man’s heroin), which has spread as a cheap substitute of heroin in former Soviet countries in the past two decades [95,102,104,105]. The illicit synthesis of desomorphine (**38**) utilizes codeine phosphate or codeine sulfate tablets sold at pharmacies as a cough suppressant. Operators reportedly use gasoline/petrol to extract the free base of **2a** liberated from the tablets by the addition of a strong alkali (KOH, NaOH). According to one street method (route: codeine (**2a**) → iodocodide (**59c**) → dihydrodesoxycodeine-D (**62**) → **38**), the extracted codeine (**2a**) is treated with iodine (iodine tincture), hydrochloric acid and red phosphorus (from match heads) for dehalogenation. The heating of the mixture gives a very complex product mixture containing **38** as the main component, along with perhaps 50 different morphinan derivatives [106], with the carryover of residual phosphorus, heavy metals and highly toxic by-products. Naloxone serves as an antidote for acute krokodil toxicity, but it is naturally ineffective against somatic injury arising from subcutaneous or intravenous injection of such a terrible concoction. Indeed, the street name krokodil refers to the necrotic skin changes associated with its use. Habitual krokodil users also suffer from horrendous wounds including deep tissue destruction, limb amputation or the auto-amputation of fingers, and osteonecrosis of facial bones and teeth.

### 2.8. Nucleophilic Substitution Reactions in the Morphine Series

The nucleophilic substitution reaction of arylsulfonyl- or alkysulfonyl-esters (tosyl, mesyl, nosyl, brosyl, mesityl) leaving groups is a fundamental transformation [107,108] for the exchange of primary and secondary hydroxyl groups to other substituents in biology, organic chemistry and radiochemistry for the production of PET tracers [109]. In 1956, Stork and Clarke [110] studied the reactions of codeine tosylate (**63a**) with Cl^⊖^, Br^⊖^, I^⊖^ and piperidine nucleophiles. The reaction of **63a** with chloride anion (LiCl, acetone, reflux, 4 h) gave the kinetic product α-chlorocodide (**59b**, 6β-chloro-Δ^7,8^-deoxycodeine, 6-chloro-6-desoxyisocodeine) in quantitative yield according to the S_N_2 mechanism (substitution with inversion). α-Chlorocodide (**59b**) is unstable and isomerizes readily to its thermodynamic isomer β-chlorocodide (**65a**, 8β-chloro-Δ^6,7^-deoxycodeine, 8-chloro-8-desoxypseudocodeine, S_N_i’). 6-Piperidocodide (**64**, 87%) was obtained by heating of codeine tosylate (**63a**) with piperidine in benzene for 36 h. In the reaction of codeine tosylate (**63a**) with bromide (LiBr, acetone, reflux, 2.5 h) or iodide (NaI, acetone, reflux, 2.5 h) anions, there was the formation of the corresponding β-halocodide derivatives (pseudocodeine series, S_N_2′ (S_N_2 + S_N_i’)) (**65b**, bromide, 98% and **65c**, iodide, 43%).

Bognár et al. [111,112] synthesized 6-deoxy-6-fluoroisocodeine (**59a**) from codeine tosylate (**63a**) with tetrabutylammonium fluoride in acetonitrile in 48% yield, and determined the configuration of its C-6 carbon by NMR spectroscopy. They found that the 6β-fluoro compound (**59a**) would not isomerize to the corresponding 8β-fluoro-derivative (β-fluorocodide, pseudocodeine series). In 1976, Makleit et al. [113,114] studied the substitution reaction of codeine tosylate (**63a**) with the Cl^⊖^ nucleophile (LiCl) in *N*,*N*-dimethylformamide. They found that the reaction temperature had a significant influence on the course of the reaction; at lower temperatures (40 °C, 5 h), there was the formation of the kinetic product α-chlorocodide (**59b**, S_N_2, 76%), while at higher temperatures (e. g., 120 °C, 10 h), the thermodynamic isomer β-chlorocodide (**65a**, S_N_2 + S_N_i’, 33%) was the predominant product. They also detected the high-temperature conversion of the 6β-chloro (**59b**) to the 8β-chloro (**65a**) isomer (120 °C, 10 h, 41%).

Makleit [115,116] thoroughly investigated the synthesis of tosyl and mesyl derivatives in the so-called morphine series (Δ^7,8^-unsaturated derivatives: morphine (**1a**), codeine (**2a**), 14-hydroxycodeine (**66**); and their 7,8-dihydro derivatives: dihydromorphine (**35**), dihydrocodeine (**67**), 14-hydroxydihydrocodeine (**68**); Figure 18). His objective was to study their nucleophilic substitution reactions, and to clarify the reaction mechanism, through the production of numerous semisynthetic morphinan derivatives [115,116]. The Makleit group further showed that the C-ring of the Δ^7,8^-unsaturated derivatives (**1a**, **2a**, **66**, Figure 18) has a flattened boat conformation, with the C-6 hydroxyl group having an allylic alcohol character in a pseudo-equatorial position. Conversely, the C-ring in the 7,8-dihydro series (**35**, **67**, **68**, Figure 18) has a chair conformation, with the C6-OH in an axial position. In both cases, the hydroxyl group connects to an asymmetric carbon in position-6. They prepared sulfonate esters of codeine (**2a**) and dihydrocodeine (**67**) by the application of previously well-established methods, resulting in codeine tosylate (**63a**, 58%, Figure 19), codeine mesylate (**63b**, 16%), dihydrocodeine tosylate (**71a**, 75%) and dihydrocodeine mesylate (**71b**, 23%), respectively [116,117,118].

#### 2.8.1. Reactions of 7,8-Dihydro Compounds

The reactions of the 7,8-dihydro-6α-sulfonate esters (**71**–**73**) with the nucleophile anions H^⊖^, N_3_^⊖^, CH_3_COS^⊖^ and CH_3_COO^⊖^ in polar aprotic solvents occur according to the S_N_2 mechanism via Walden inversion, yielding products of the *iso* series. Dihydrocodeine tosylate (**71a**) reacted with N_3_^⊖^ nucleophile resulted in 6-desoxy-6-azido-dihydroisocodeine (**74a**, Figure 20) [54,55,117,118,119,120], while 14-hydroxydihydrocodeine tosylate (**73a**) under identical conditions (10 equiv., NaN_3_, DMF, 100 °C, 24 h) with the same reagent gave 6-desoxy-6-azido-14-hydroxydihydroisocodeine (**74b**, 31%) [121,122]. The reduction of the 6-azido compounds (**74a**–**b**) with LiAlH_4_ [117,118] or by catalytic hydrogenation [121,122] led to the corresponding 6-amino (6-beta-amino or 6-amino-dihydroisocodeine) derivatives (**75a**–**b**).

Makleit and Bognár [123,124] accomplished the epimerization of the C-6 chiral centers of codeine (**2a**) and dihydrocodeine (**67**). The acetolysis of codeine tosylate (**63a**) with 10% acetic acid (reflux, 4 h) gave isocodeine (**2b**) in 16% yield [123,124]. In the 1990s, the Makleit group developed protocols resulting in higher yields for the derivatives of the iso series via the application of the Mitsunobu reaction (see Section 2.11). 6-*O*-Acetyl-dihydroisocodeine (**76**) was prepared starting from dihydrocodeine mesylate (**71b**) with acetic acid, acetic anhydride and sodium acetate (reflux, 31 h). Subsequently, the 6-*O*-deacetylation of **76** was achieved by alkaline hydrolysis to yield dihydroisocodeine (**77**), with an overall 21% yield from dihydrocodeine mesylate (**71b**)). When dihydrocodeine tosylate (**71a**) was treated with the CH_3_COS^⊖^ nucleophile (KSCOCH_3_, DMF), 6-acetylthio-dihydroisocodeine (**78**) was obtained [125,126,127,128]. The latter compound was deacetylated with sodium methylate to 6-desoxy-6-thiol-dihydroisocodeine (**79**). Upon reacting 3-*O*-acetyl-dihydromorphine tosylate (**72a**) with hydride anion (LiAlH_4_), there was the elimination of the tosyloxy group to yield 6-desoxydihydromorphine (**38**) [68,69,70].

#### 2.8.2. Reactions of Δ^7,8^-Unsaturated Derivatives

Sulfonate esters (6α-*O*-tosyl and 6α-*O*-mesyl) of Δ^7,8^-unsaturated derivatives react with different nucleophiles either according to an S_N_2 mechanism to give 6β-substituted derivatives, or via an S_N_2 reaction accompanied by allylic rearrangement (an S_N_2 + S_N_i’ mechanism) to result in 8-substituted-8-desoxypseudo compounds (see supposed mechanism in Section 2.9.1). Both the character of the nucleophile and the stereochemical properties of the substrates are of crucial importance in determining the reaction mechanism. For example, codeine tosylate (**63a**) with LiCl gave α-chlorocodide (**59b**) [110], whereas the reaction of isocodeine tosylate (**81**) under identical conditions with the same reagent resulted in β-chlorocodide (**65a**) [121,122].

Codeine tosylate (**63a**) with tetrabutylammonium fluoride in acetonitrile [111,112] led to the formation of 6-fluoro-6-desoxyisocodeine (**59a**, Figure 21). The isolated 6α-fluoro-type compound (**59a**, α-fluorocodide) cannot isomerize to the 8β-fluoro-type (β-fluorocodide) derivative. Compound **59a** was also prepared from pseudocodeine tosylate (see structure **97** in Section 2.8.3) with tetrabutylammonium fluoride (5.4 equiv., CH_3_CN, reflux, reflux, 29 h) in poor yield (10%) [129,130]. 6-Fluoro-6-desoxy-14-hydroxyisocodeine (**80**) was prepared from 14-hydroxycodeine tosylate (**70a**) under identical conditions [121,122]. When codeine tosylate (**63a**) was treated with 10% acetic acid (reflux, 4 h), the main direction of the reaction was toward the formation of isocodeine (**2b**, 16% isolated yield) [123,124] without allylic rearrangement. This method was also applicable for the synthesis of isomorphine (**1b**, 16%) from 3-*O*-acetylmorphine tosylate (**69a**) with the subsequent preparation of dihydroisomorphine (**82**) through the catalytic reduction of isomorphine (**1b**) [131,132].

Codeine tosylate (**63a**) with azide anion (NaN_3_) in *N*,*N*-dimethylformamide gave 8-azido-8-desoxypseudocodeine (**83**, 8-azidocodide, Figure 22), which was converted to 8-amino-8-desoxypseudocodeine (**84**, 8-aminocodide) with lithium aluminum hydride [54,55,117,118]. The azidolysis of 14-hydroxycodeine tosylate (**70a**) with 1.25 equiv. of sodium azide gave 6-azido-6-desoxy-14-hydroxyisocodeine (**87**, 38%) when the reaction mixture was heated at 100 °C for 4 h [121,122]. Meanwhile, the application of the same 0.25% NaN_3_ excess and a longer reaction time (100 °C, 8h) led to 8-azido-8-desoxy-14-hydoxypseudocodeine (**85**, 46%). 6-Azido-6-desoxy-14-hydoxyisocodeine (**87**) was isomerized by heating in DMF (100 °C, 6 h) 83%) to **85** in 83% yield [121,122]. The reduction of 8-azido-8-desoxy-14-hydoxypseudocodeine (**85**) with LiAlH_4_ gave 8-amino-8-desoxy-14-hydoxypseudocodeine (**86**), although the catalytic hydrogenation led to 8-amino-8-desoxy-14-hydoxydihydropseudocodeine (**88**) [121,122].

The treatment of codeine tosylate (**63a**) with the CH_3_COS^⊖^ nucleophile (KSCOCH_3_, DMF) [127,128] led to 8-acetylthio-8-desoxypseudocodeine (**89**, Figure 23), which was deacetylated with sodium methylate to give 8-thiol-8-desoxypseudocodeine (**90**). The oxidation of the latter compound gave an 8,8′-didesoxy-8,8′-pseudocodeinyl disulfide-type compound. Surprisingly, the reaction of codeine tosylate (**63a**) with thiocyanate anion (^⊖^S=C=N, KSCN, in acetone) gave (in a kinetically second-order reaction) the product 8-isothiocyanato-8-desoxypseudocodeine (**92**, -NCS) instead of the expected 8-thiocyanato-8-desoxyspeudocodeine (**91**, -SCN) [127,128].

The analogous reaction of 3-*O*-acetylmorphine tosylate (**69a**, Figure 24) with CH_3_COS^⊖^ led to 3-*O*-acetyl-8-acetylthio-8-desoxypseudomorphine (**93**), which was deacetylated with sodium methylate to 8-thiol-8-desoxypseudomorphine (**94**) [127,128]. The treatment of **69a** with potassium thiocyanate resulted in 3-*O*-acetyl-8-isothiocyanato-8-desoxypseudomorphine (**95**) [133,134]. The reaction of **69a** with N_3_^⊖^ anion gave 8-azido-8-desoxypseudomorphine (**96**, Makleit–Bognár nomenclature, 8-azido-8-desoxy-γ-isomorphine).

#### 2.8.3. Reactions of Pseudocodeine Tosylate

In the early 1970s, Makleit et al. [117,118] extended their studies on the nucleophilic substitution reaction of sulfonate esters to the Δ^6,7^ double bond containing pseudocodeine tosylate (**97**, Figure 25). Pseudocodeine (**2d**) was prepared from α-chlorocodide (**59b**) by hydrolysis according to the method of Small and Lutz [135]. The tosyl ester (**97**) was prepared from **2d** in 76% yield. They investigated the substitution reactions by the application of a large variety of nucleophilic reagents. Using type-X reagents (F^⊖^, Cl^⊖^ or piperidine), *isocodeine* derivatives (**99**) were formed either by an S_N_1′ or S_N_2′ mechanism in a yield in the range of 10–65%. Using type-Y nucleophiles (Br^⊖^, I^⊖^, NCS^⊖^, N_3_^⊖^, CH_3_COS^⊖^), *pseudocodeine* derivatives (**98**) were isolated in poor yields (10–52%). In these latter cases, the mechanism of the reaction was either S_N_1 accompanied by retention or a combination of S_N_2′ and S_N_i’.

The corresponding products were isolated in poorer yields despite more vigorous reaction conditions and longer reaction times in comparison to the analogous reactions of codeine tosylate (**63a**). The authors undertook comparative solvolysis of pseudocodeine tosylate (**97**) and codeine tosylate (**63a**) in an acetic acid–potassium acetate system, finding that the solvolysis of 8β-*O*-tosylate (**97**) occurred slower than that of 6α-*O*-Tos (**63a**) following, in both cases, a first-order reaction. They investigated more thoroughly the reaction of pseudocodeine tosylate (**97**) with piperidine, definitively showing the formation of the 6β-(1-piperidinyl) derivative (**64**, 6-piperidocodide) by an S_N_1′ reaction mechanism.

#### 2.8.4. Neopine Derivatives

Neopine derivatives (e.g., isoneopine (**3b**), Figure 26) are critical starting materials for the synthesis of B/C-*trans*-morphinans, which can be very difficult to produce [136,137]. In the 1970s, Berényi, Makleit and Bognár worked out numerous syntheses for the preparation of neopine (**3a**) [138,139] and isoneopine (**3b**) [140,141]. Furthermore, Berényi, Makleit and Dobány developed an improved method for the isolation of natural neopine (**3a**) from poppy straw [142], without interfering with the original Kabay process. In 1977, Makleit et al. [138,139] reported an efficient method for the preparation of neopine (**3a**) from thebaine (**4**) via 14-chlorocodeine. The reduction of the latter compound (**124a**) was achieved with sodium bis(2-methoxyethoxy)aluminum hydride. The overall yield for neopine (**3a**) from thebaine (**4**) was 72%. In 1980, Berényi and Makleit [140,141] published a novel method for the preparation of isoneopine (**3b**), wherein the azidolysis of neopine mesylate (**100**) resulted in 6-deoxy-6-azido-isoneopine (**101**) in 75% yield. The latter compound was reduced with Zn/NaI/DMF to 6-deoxy-6-amino-isoneopine (**102**, 65%). The treatment of the amino compound (**102**) with NaNO_2_ in acetic acid gave 6-*O*-acetylisoneopine (**103**, 90%), and finally, the alkaline hydrolysis of the 6-*O*-acetyl group resulted quantitatively in isoneopine (**3b**). The overall yield from neopine (**3a**) was 44%.

Taking advantage of the availability of neopine (**3a**) in large quantities, Berényi et al. [143,144] prepared 6-*O*-mesylneopine (**100**, Figure 27) for the investigation of its substitution reactions with nucleophile partners (F^⊖^, Cl^⊖^, Br^⊖^, I^⊖^, N_3_^⊖^). When 6-*O*-mesylneopine (**100**) was reacted with tetrabutylammonium fluoride in acetonitrile (reflux, 2 h), 6-demethoxythebaine (**105**, 77%) was obtained. In the reaction of neopine mesylate (**100**) with iodide anions (10 equiv. NaI, DMF, 120 °C, 30 h), 6-demethoxythebaine (**105**, 21%) was isolated, with 36% recovery of the starting material (**100**).

Through the application of chloride anions (10 equiv. LiCl, DMF, 120 °C, 6 h), 6-deoxy-6-chloro-isoneopine (**104a**) and 6-demethoxythebaine (**105**) were formed in a ratio of 6:4 [143,144]. Through the use of bromide anions as a nucleophile (10 equiv. LiBr, DMF, 120 °C, 11h), 6-deoxy-6-bromo-isoneopine (**104b**, 32%) and 6-demethoxythebaine (**105**, 31%) were isolated, with 10% recovery of the starting material (**100**). The 6-β position of the halogen substituent in compounds **104a** and **104b** (Cl, Br) was substantiated by NMR spectroscopy, in which the value of the geminal coupling of the H-5 and H-6 protons (^3^*J*_5,6_ = 9.5–10 Hz) was characteristic of both compounds. The azidolysis of neopine mesylate (**100**) with 10 equivalents of NaN_3_ in DMF (120 °C, 24 h) resulted in 6-deoxy-6-azido-isoneopine (**101**, 55%). The latter compound (**101**) was converted to 6-deoxy-6-amino-isoneopine (**102**) by heterogenous catalytic reduction (H_2_, 50% acetic acid, platinum oxide). The authors concluded that the first step of these reactions is an S_N_2 nucleophile substitution resulting in the corresponding 6-desoxy-6-haloisoneopines (**104a**–**b**), followed by a second elimination step resulting in 6-demethoxythebaine (**105**). 6-Desoxy-6-bromo-isoneopine (**104b**) was converted to 6-demethoxythebaine (**105**, reflux, 4 h, 72%) with tetrabutylammonium fluoride, while 6-deoxy-6-chloro-isoneopine (**104a**) gave, according to 50% conversion, a 1:1 mixture of 6-demethoxythebaine (**105**) and the starting material (**104a**). Based on these observations, Hutchins et al. and Beyerman, Maat and Crabbendam reported improved preparations of 6-demethoxythebaine (**105**) in high yields either from neopine (**3a**) or from codeine (**2a**) via isocodeine (**2b**) or β-bromocodide (**65b**) [145,146,147]. The reaction of thebaine (**4**) with peracids, peracetic acid (CH_3_CO-OOH) or performic acid (HCO-OOH) [148,149] resulted in 14-hydroxycodeinone by the 1,4-addition of hydroxyl groups to the 6,8-conjugated system followed by methanol elimination from the 6-hemiacetal intermediate.

The analogous treatment of 6-demethoxythebaine (**105**, Figure 27) with formic acid and hydrogen peroxide [143,144] gave 14-hydroxy-allopseudocodeine (**106**, 71%). Th latter compound was also prepared previously by Currie et al. [150] from 6-*O*-tosyl-14-hydroxycodeine (**70a**, Figure 28) in an S_N_2′ reaction (70% AcOH) through 8-*O*-acetyl-14-hydroxy-allopseudocodeine (**107**) and subsequent alkaline hydrolysis. Seki [151] prepared **106** from 14-hydroxy-α-chlorocodide (**108**) via acetolysis with 10% AcOH and subsequent hydrolysis.

#### 2.8.5. Alkyl Mesylate and Allyl Halide Structural Units in the Same Molecule

Studying the relative reactivity of alkyl mesylates (C^8^-C^7^-C^6^-OMs) and allyl halides (C^14^=C^8^-C^7^-X), Simon et al. [152,153] investigated the nucleophilic substitution reactions of 6-*O*-mesyl-7α-haloneopine derivatives (**112a**–**b**) containing both structural units in the same molecule. The starting materials 7α-chloroneopine (**111a**) and 7α-bromoneopine (**111b**) were prepared in three steps from thebaine (**4**) by the method of Abe et al. [154] (Figure 29). In brief, thebaine (**4**) was reacted with *N*-bromosuccinimide to give 14-bromocodeinone (**109**) [155]. The latter compound was stereoselectively reduced with NaBH_4_ to 14-bromocodeine (**110**). The treatment of **110** with concentrated hydrochloric acid or 48% hydrobromic acid resulted in the corresponding 7α-haloneopines (**111a**–**b**).

Nucleophilic substitution reactions of 6-*O*-mesyl-7α-chloro-neopine (**112a**) and 6-*O*-mesyl-7α-bromoneopine (**112b**) were studied with the following nucleophiles: Cl^⊖^, Br^⊖^, I^⊖^ and N_3_^⊖^ [152,153] (Figure 30). When 6-*O*-mesyl-7α-chloroneopine (**112a**, Figure 30) was reacted with azido anions (5 equiv. NaN_3_), there was the formation of a complex product mixture. Column chromatography isolated the main product 7-chloro-6-demethoxythebaine (**113a**, 31%) and the side products 6-*O*-mesyl-7β-azidoneopine (**114**, 10%) and 6-*O*-mesyl-6-demethoxythebaine (**115**, 5%).

When 6-*O*-mesyl-7α-bromoneopine (**112b**, Figure 30) was reacted with 5 equiv. of sodium azide in *N*,*N*-dimethylformamide (addition of water was necessary to dissolve NaN_3_), the main product was 6-*O*-mesyl-7β-azidoneopine (**114**, 48%). The isolated side products were 6-*O*-mesyl-6-demethoxythebaine (**115**, 14%) and 9α-azido-6-*O*-mesyl-indolinocodeine (**116**, 3%).

In studies of the reactions of 6-*O*-mesyl-7α-chloroneopine (**112a**, Figure 31) with 5 equivalents of reagent (LiCl, LiBr, KI) [152,153], the main product with chloride anions (5 equiv. LiCl, DMF, 100 °C, 24 h) was 7-chloro-6-demethoxythebaine (**113a**, 39%), with the isolation of 6-chloro-6-demethoxythebaine (**118a**, 18%) and 6-*O*-mesyl-7β-chloroneopine (**117a**, 3%) side products. When LiBr was used as the source of nucleophile under otherwise identical conditions, the main reaction was methanesulfonic acid elimination, which resulted in 7-chloro-6-demethoxythebaine (**113a**, 31%) and 6-bromo-6-demethoxythebaine (**118b**, 18%), as well as 6-*O*-mesyl-7β-bromoneopine (**117b**, 13%). In the presence of iodine anions (5 equiv. KI, DMF, 100 °C, 48 h), the main product was 6-demethoxythebaine (**105**, 34%).

Shorter reaction times (20–40 min) were applied in studying the reactions of 7α-bromoneopine mesylate (**112b**, Figure 32) with Cl^⊖^, Br^⊖^ and I^⊖^ nucleophiles. With LiCl, the main product was 6-*O*-mesyl-7β-chloroneopine (**117a**, 19%), with the additional isolation of 6-*O*-mesyl-6-demethoxythebaine (**115**, 7%). When **112b** was reacted with LiBr, the products were 6-*O*-mesyl-7β-bromoneopine (**117b**, 24%), 7-bromo-6-demethoxythebaine (**113b**, 3%) and 6-*O*-mesyl-6-demethoxythebaine (**115**, 6%). When potassium iodide was applied, 6-demethoxythebaine (**105**, 72%) was the single isolated product. The study [152,153] also presented an elegant and efficient method for the preparation of 7-halo-6-demethoxythebaines (**113a**–**b**).

When 6-*O*-mesyl-7α-chloroneopine (**112a**, Figure 33) or 6-*O*-mesyl-7α-bromoneopine (**112b**) was boiled in the presence of potassium *tert*-butylate in methanol, the main reaction was methanesulfonic acid elimination (E_1,2_ (-CH_3_SO_2_OH)). 7-Chloro-6-demethoxythebaine (**113a**) was prepared as the sole product in 95% yield from **112a**, and 7-bromo-6-demethoxythebaine (**113b**) was the main product in 50% yield from **112b**.

#### 2.8.6. Substrates Containing Double Allylic System

##### Substrates with Allyl Halide and Allyl Tosylate Subunits

Berényi et al. [156,157,158] were the first to study the nucleophilic substitution reactions of 6-*O*-tosyl-14-halocodeine substrates (**119a**–**b**), which possess a double allylic system: allyl halide (C^7^=C^8^-X^14^) and allyl tosylate (C^8^=C^7^-C^6^-OTos) in the same molecule. They investigated reactions of 14-chlorocodeine tosylate (**119a**, Figure 34) [156,157,158] and 14-bromocodeine tosylate (**119b**) [155] with Cl^⊖^, Br^⊖^ and N_3_^⊖^ nucleophiles. Surprisingly, 6-chloro-6-demethoxythebaine (**118a**, 68%) as well as 6-bromo-6-demethoxythebaine (**180b**, 74%) was isolated when the more stable 14-chlorocodeine tosylate (**119a**) was reacted with Cl^⊖^ or Br^⊖^ anions (LiX, X = Cl. Br, DMF, 100 °C, 24 h).

Interestingly, 6-*O*-tosyl-14-bromocodeine (**119b**) gave 6-*O*-tosyl-6-demethoxythebaine (**120**) under similar conditions. Heating 6-*O*-tosyl-14-bromocodeine (**119b**) in DMF (100 °C, 2 h) without any nucleophilic reagent gave the latter compound (**120**, 26%). A further surprising result was that 6-*O*-tosyl-6-demethoxythebaine (**120**) did not react with Cl^⊖^ or Br^⊖^ anions.

The authors [156,157,158] explained the complicated formation of 6-halo-6-demethoxythebaine derivatives (**118a**–**b**) by the following elementary steps (Figure 35): (i): S_N_2′ reaction: nucleophilic attack at position-7, resulting in 6-*O*-tosyl-7-substituted neopine (**A**); (ii): S_N_2 reaction: further nucleophile attack on the 6-tosyloxy group, resulting in 6-deoxy-6β,7β-disubstituted neopine (**B**); and (iii): 1,4-elimination reaction.

The azidolysis [156,157,158] of 14-chlorocodeine tosylate (**119a**, Figure 36) with 2.5 equiv. NaN_3_ resulted in 6-deoxy-6β,7α-diazido-neopine (**121**, 27%) as the main product and 6-deoxy-6β,7β-diazido-neopine (**122**, 7%) as a minor by-product. Carrying out the azidolysis of **119a** with 2.5 equiv. trimethylsilyl azide in DMF at 100 °C yielded 6-*O*-tosyl-6-demethoxythebaine (**120**) and 6-deoxy-6β,7β-diazido-neopine (**122**) in a ratio of approximately 1:1, with a significant amount of the starting material (**119a**) remaining in the product mixture after a 24 h reaction time. When 6-*O*-tosyl-14-bromocodeine (**119b**) was reacted with azido anions (2.5 equiv. NaN_3_), the major product was 6-*O*-tosyl-7β-azidoneopine (**123**, 25%), with a lesser amount of 6-*O*-tosyl-6-demethoxythebaine (**120**, 16%) in 16% yield [156,157,158]. The reactivity sequence of the leaving groups was as follows: allyl bromide > allyl tosylate > allyl chloride.

##### Substrates Containing Double Allyl Halide Sub-Structural Units

In 1988, Berényi et al. [159] reported the substitution reactions of 6β,14-dichlorodesoxycodeine (**125a**) and 6β-chloro-14-bromodesoxycodeine (**125b**, Figure 37), containing double allyl halide (C^7^=C^8^-C^14^-X and C^8^=C^7^-C^6^-Cl) sub-structural units with the nucleophiles Cl^⊖^, Br^⊖^ and N_3_^⊖^ applied as LiCl, LiBr and NaN_3_, respectively. They also studied reactions in hot DMF in the absence of a nucleophilic partner. The authors claimed that only leaving groups (Cl^⊖^, Br^⊖^) linked to the tertiary C^14^ carbon (C^7^=C^8^-C^14^-X) took part in nucleophilic substitution reactions among the investigated 6β-chloro-14-halodesoxycodeines (**125a**–**b**).

The starting materials **125a** and **125b** for these investigations were prepared from 14-chlorocodeine (**124a**) [155] and 14-bromocodeine (**124b**, thebaine (**4**) → 14-bromocodeinone (**109**) → **1b**), respectively [160]. The reaction of **124a** with 2 equiv. phosphorus pentachloride in chloroform gave **125a** in 66% yield, and the analogous reaction of 14-bromocodeine (**124b**) gave **125b** in 53% yield. The researchers also prepared **125a** from 14-chlorocodeine tosylate (**119a**) with LiCl.

6β-Chloro-14-bromodeoxycodeine (**125b**, C^7^=C^8^-C^14^-Br and C^8^=C^7^-C^6^-Cl) reacted readily with Cl^⊖^ anion (10 equiv. LiCl) in DMF, giving 6β-chloro-7β-bromodesoxyneopine (**126a**) as the main product in 42% yield, with the further isolation of 6α-chloro-7β-bromodesoxyneopine (**126b**) in 4% yield by the column chromatographic separation of the mother liquor. The application of a Br^⊖^ nucleophile (5 equiv. LiBr, DMF) for **125b** gave 6-β-chloro-7β-bromodesoxyneopine (**126c**) in 16% yield. Using N_3_^⊖^ anions (5 equiv. NaN_3_, DMF, 100 °C, 1 h) gave 6β-chloro-7α-azidodesoxyneopine (**126d**, 44%) and 6β-chloro-9α-azidodesoxyindolinocodeine (**127**, 8%). After the heating of **125b** in DMF at 100 °C for 30 min in the absence of Nu^⊖^ unreacted starting material (**125b**), 6β-chloro-7β-bromodesoxyneopine (**126c**) and 6-chloro-6-demethoxythebaine (**118a**) were separated from the product mixture by column chromatography. Extending the reaction time (1 h) gave 6-chloro-6-demethoxythebaine (**118a**) in 62% yield.

The reaction of 6β,14-dichlorodesoxycodeine (**125a**, C^7^=C^8^-C^14^-Cl and C^8^=C^7^-C^6^-Cl) with Br^⊖^ anions (5 equiv. LiBr, DMF, 100 °C, 8 h) led to 6-chloro-6-demethoxythebaine (**118a**) in 72% yield. The azidolysis of **125a** under identical conditions to those described above for **125b** resulted similarly in 6β-chloro-7β-bromodesoxyneopine (**126c**, 48%) and 6-chloro-6-demethoxythebaine (**118a**, 7%). After the heating of **125a** in DMF at 110 °C without Nu^⊖^ reagent for 30 min, 6-chloro-6-demethoxythebaine (**118a**, 60%), 6β-chloro-7β-chlorodesoxyneopine (**126a**, 4%) and 6β-chloro-7α-chlorodesoxyneopine (**126b**, 1%) were isolated. Upon continuing the heating for 24 h, 6-chloro-6-demethoxythebaine (**118a**) was obtained in 72% yield. The differences in reactivity between the allyl chloride (C^7^=C^8^-C^14^-Cl and C^8^=C^7^-C^6^-Cl) and allyl bromide (C^7^=C^8^-C^14^-Br) subunits of the molecules manifested in the differing times required for whole conversion via an elimination reaction (E_1,2_ type) in the absence of Nu^⊖^ reagents. Thus, the conversion of 6β-chloro-14-bromodesoxycodeine (**125b**) to 6-chloro-6-demethoxythebaine (**118a**) took one hour, whereas the conversion of 6β,14-dichlorodesoxycodeine (**125a**) required 24 h.

### 2.9. Azidomorphinans

Organic azides [161,162,163] are suitable building blocks for the synthesis of a broad range of nitrogen-containing scaffolds. Due to the high electrophilicity of the azido group (mesomeric structures: -N=N-N: ↔ -N=N^⊕^=N^⊖^ ↔ -N^⊖^-N^⊕^≡N ↔ -N^⊖^-N=N^⊕^), organic azides are prone to react with various nucleophiles [164,165] and, furthermore, serve as starting materials for a large variety of transformations (e.g., reactions with electrophiles, cycloaddition, 1,3-cycloaddition with alkynes = click reaction, reduction to amines, etc.). The incorporation of the azido pharmacophore into a bioactive scaffold can be advantageous in providing new molecules with desirable pharmacological profiles, including increased potency, and other biochemical and pharmacological characteristics [166].

Cocker et al. [167] and later Willner et al. [168] thoroughly investigated the displacement by N_3_^⊖^ and other nucleophiles of *N*-acetylated 7-aminocephalosporanic acid (7-ACA) derivatives. The prepared 3-azidomethyl-3-cephem analogs were effective in inhibiting the growth of Gram-negative bacteria [167,168].

Azidomazenil (8-azido-5,6-dihydro-5-methyl-6-oxo-4*H*-imidazo-1,4-benzodiazepine-3-carboxylic acid ethyl ester, Ro-15,4513, Hoffmann-La-Roche), the azido analog of the 1,4-benzodiazepine derivative flumazenil (Ro-15,1788, Hoffmann-La-Roche), a partial inverse GABA_A_ receptor (GABA_A_R) agonist, blocks the electrophysiological and behavioral effects of low-to-moderate ethanol doses [169]. Indeed, GABA_A_Rs may mediate the rewarding effects of ethanol, which require binding at the benzodiazepine-binding site (cBZR) of the α5-subunit of the GABA_A_R [170]. In 1992, the ^11^C-labeled version of Ro-15,4513 was developed by Halldin et al. [171] for PET imaging of benzodiazepine receptors in a living human brain [170].

The azido analog of levorphan and its tritiated version *N*^17^-(4-azidophenyl-CH_2_C(^3^H)_2_)-norlevorphan were synthesized by Winter et al. [172] in 1972. *N*^17^-Desmethyl-levorphan was reacted with β-(4-azidophenyl)ethyl bromide in methanol (K_2_CO_3_, 42 °C, 90 h) to yield the target compound, which was four-fold more effective (ED_50_ = 0.78 mg/kg) than levorphan as an analgesic in the mouse hot-plate test, perhaps due to its greater lipophilicity and brain penetration.

#### 2.9.1. Azidomorphine Analogs

6-Azido-substituted morphine derivatives have a conspicuous analgesic effect [173,174], having a 50–300-fold increase in analgesic potency relative to morphine (**1a**) in the case of a 6-azido substitution of dihydroisomorphine (**82**, Figure 21) [175,176,177]. Over several decades, the research group of Makleit paid particular attention to investigating the nucleophilic substitution reaction of the sulfonate esters of the morphine series (see Figure 18 and Figure 19). They studied extensively the azidolysis of 6-*O*-tosyl and 6-*O*-mesyl esters of different morphinans and the subsequent transformations to the corresponding amino derivatives. In 1968, Bognár and Makleit investigated the syntheses of aminocodide and aminomorphide derivatives (Figure 38) [54,55], at a time when only a few ring-C amino-substituted morphinan derivatives were known. Furthermore, the absolute stereochemistry of these compounds, 6-aminocodide, 8-aminocodide (**84**) and 8-aminodihydrocodide (**75a**), was uncertain at the time. As starting points in the Δ^7,8^-unsaturated series, they used the sulfonate esters of codeine (**2a**) and 3-*O*-acetylmorphine (**55**). These materials readily reacted with azide anion (N_3_^⊖^), giving the 8β-azides azidocodide (**83**) as well as 8-azidomorphide (**128**), i.e., products belonging to the pseudocodeine or γ-isomorphine series (pseudomorphine derivatives, according to the Makleit–Bognár nomenclature).

The nucleophile substitution of codeine tosylate (**63a**) with azide anion (N_3_^⊖^) can occur according to an S_N_2′ reaction mechanism (Figure 39), whereby the azide anion attacks the Δ^7,8^ double bond at the C-8 position to give the 8β-azide (**XV**, Nu = N_3_, **83**) directly. Another possible mechanism is an S_N_2 reaction followed by allylic rearrangement ([3,3]-suprafacial sigmatropic rearrangement, S_N_i’ mechanism), where the nucleophile (N_3_^⊖^) attacks the C-6 position. The hypothesized 6β-azide (**XVI**, Nu = N_3_^⊖^, allylic azide) intermediate undergoes an allylic rearrangement, which shifts the new substituent from C-6 into the C-8 position to give 8-azidocodide (**XV**, Nu = N_3_^⊖^, **83**).

The analogous reaction of 6-*O*-sulfonate esters belonging to the 7,8-dihydro series (e.g., **71a, 72a**) [117,118] with the N_3_^⊖^ nucleophile gave the corresponding 6β-azides: 6-desoxy-6-azidodihydroisocodeine (**74a**, “azidocodeine”) [119,120] and 6-desoxy-6-azidodihydroisomorpine (**130**, “azidomorphine”) [119,120], with the inversion of the configuration in a stereospecific nucleophile substitution reaction (S_N_2). The azides (**83**,**128** and **74a**,**130**) were reduced with LiAlH_4_ in THF to their corresponding 8β-amino (**84**,**129**) or 6β-amino (**75a**,**131**) derivatives.

In a rat hot-plate test of analgesia, azidomorphine (**130**, 6-AM, 6-desoxy-6-azidodihydroisomorphine; 3-hydroxy-4,5-α-epoxy-6β-azido-7,8-dihydro-17-methyl- morphinan) [119,120] was 270–300 times more potent than morphine (ED_50_ (**130**) = 0.016 mg/kg, versus ED_50_ (**1a**) = 4.7 mg/kg), whereas azidocodeine (**74a**, 6-desoxy-6-azidodihydroisocodeine, ED_50_ (**74a**) = 0.36 mg/kg;) was only 10 times more potent than morphine (**1a**) [178]. Nonetheless, mice showed lesser propensity for developing physical dependence on azidomorphine (**130**) compared to morphine (**1a**). Interestingly, there was a consistent dissociation between analgesic activity and physical dependence liability in monkeys, rats and mice [178]. Intensive investigations by Bulaev et al. of the pharmacological characteristics in vivo of azidomorphine (**130**) in different animal models (rats, rabbits and cats) consistently showed 20–100 times greater analgesic activity compared to morphine (**1a**) [179,180]. Knoll et al. [181,182,183] demonstrated that azidomorphine (**130**) is metabolized in rats by *N*-demethylation to norazidomorphine and by conjugation with glucuronic acid. In human studies, azidomorphine (**130**) was 40–300 times more potent than morphine (**1a**) [184], relieving severe pain in 0.5–1 mg doses (sc. or. iv.), but with less constipation, vomiting and euphoria than with equianalgesic doses of morphine (**1a**). When a composition containing azidomorphine (**130**, 0.5 mg) was given, the non-narcotic analgesic rimazolium mesylate (1,6-dimethyl-3-carbethoxy-4-oxo-6,7,8,9-tetrahydro-homopyrimidazol, probon, MZ-144; 150 mg) potentiated the analgesic effects while antagonizing the respiratory depressant effects of **130** [175,176,177]. In a human subject, azidomorphine (**130**) was metabolized by conjugation (6-AM-3G) and by reduction to 6-desoxy-6-aminoisomorphine [185].

In 1974, Sasvári et al. [186] reported the X-ray structural analysis of azidomorphine (**130**). When exploring the crystal structure of 14-hydroxyazidomorphine (**138**), Kálmán et al. [187] found that the substitution of the H-14 of azidomorphine (**130**) with a hydroxy group did not alter the C- and D-ring chair conformation, but did alter the orientation of the 6β-azido group. Tamás et al., studied the mass spectroscopic fragmentation pathway of azidomorphine (**130**) and its derivatives [188]. Under electron impact (70 eV, source temperature 100–150 °C), azidomorphine analogs gave a remarkably different fragmentation pattern compared to that of the parent morphine alkaloids; the main decomposition route involved the loss of a nitrogen (N_2_) molecule in the 6-desoxy-6-azidoisomorphine series (**74a**, **130**, **138**, **144**). Among 8-azido-substituted compounds containing a Δ^6,7^ double bond in ring-C (e.g., azidomorphide (**128**, 8-desoxy-8-azido-γ-isomorphine) or azidocodide (**83**)), there was a loss of the N_3_ radical (·N=N^⊕^=N^⊖^) due to the allylic effect. Dinya et al. [189,190,191] performed infrared spectroscopic investigations and quantum chemical analysis of azidomorphinans.

Aiming to improve upon the toxicity of azidomorphine (**130**, LD_50_ (**130**) = 8.1 mg/kg; LD_50_ (**1a**) = 320 mg/kg, i.v. in a rat) [178], Makleit et al. developed “14-hydroxyazidomorphine” (**138**, 6-desoxy-6-azido-14-hydroxydihydroisomorphine, 3,14β-dihydroxy-4,5α-epoxy-6β-azido-7,8-dihydro-17-methylmorphinan, Figure 40) in 1972 [121,122,192,193,194]. The synthesis of **138** from thebaine (**4**) followed seven consecutive steps. In brief, 14-hydroxycodeinone (**132**) was obtained from **4** with hydrogen peroxide and formic acid [151], and the Δ^7,8^ double bond of **132** was then saturated under heterogenous catalytic conditions (H_2_, 10% Pd-C, AcOH) to give oxycodone (**133**). Oxymorphone (**134**) was prepared from **133** by 3-*O*-demethylation with 48% hydrobromic acid. The C-6 keto group of **134** was reduced by sodium borohydride in a stereospecific reaction to 14-hydroxy-dihydromorphine (**135**). The phenolic 3-hydroxyl group of **135** was protected by partial acetylation using the method of Welsh [99] with acetic anhydride (NaHCO_3_, H_2_O). The 6α-hydroxyl group of compound **136** was transformed to sulfonate esters (**137a**–**b**) with the corresponding reagent (MsCl or TosCl, pyridine, 24 h, RT). Finally, the tosyl ester (**137a**) was subjected to azidolysis (sodium azide, DMF, H_2_O, 100 °C, 24 h) to give the desired azido derivative (**138**) [121,122,192,193]. 6-Desoxy-6-azido-14-hydroxydihydroisomorphine (**138**) had similar analgesic potency in the mouse hot-plate test to azidomorphine (**130**) (ED_50_ (**130**) = 0.024 mg/kg, ED_50_ (**138**) = 0.029 mg/kg) [184], although with lower toxicity. The introduction of the hydroxy group in position-14 of azidomorphine (**130**) conspicuously reduced toxicity; azidomorphine (**130**) was 6.5 times more toxic in rats and 11.6 times more toxic in mice than 14-hydroxyazidomorphine (**138**) [184]. Human studies indicated 40-fold greater analgesic activity for azidomorphine (**130**) and 14-hydroxyazidomorphine (**138**) compared to morphine (**1a**) [195,196].

The antitussive activity of **138** was determined and compared with that of the reference antitussive codeine (**2a**) using the Gosswald citric acid aerosol-induced cough test in Wistar rats [194]. 14-Hydroxyazidomorphine (**138**) had 1000-fold higher antitussive activity compared to codeine (**2a**, AtD_50_ (**2a**, s.c.) = 19.0 mg/kg, AtD_50_ (**138**, s.c.) = 0.021 mg/kg, ratio **2a**/**138** = 904) [194].

In a subsequent study, the same research group [197,198] performed structure optimization investigations, aiming to identify azidomorphine analogs with a more advantageous pharmacological profile. For the synthesis of 3-desoxyazidomorphine (**141**, Figure 41), they first produced the starting material 3-desoxydihydromorphine (**139**) by the removal of the 3-hydroxyl group of morphine via the hydrogenation of morphine-3-(1-phenyltetrazolyl)- or morphine-3-(pyrimidyl)-ether [199]. Under these conditions, the Δ^7,8^ double bond was also saturated. In the next steps, the tosyl ester (**140**) was prepared and subjected to azidolysis (sodium azide, DMF, H_2_O, 100 °C, 24 h) and transformed to 3-desoxyazidomorphine (**141**) in 20% yield.

Analogously, “azidoethylmorphine” (**144**, “3-*O*-ethylazidomorphine”, azidodionine, 6-azido-6-desoxy-ethyldihydroisomorphine) [197,198] was prepared from ethylmorphine (**33**) in three steps (Figure 41). Compound **33** was reduced to 3-*O*-ethyl-dihydromorphine (**142**) [200,201], which was converted to the esters ethyldihydromorphine tosylate (**143a**) or mesylate (**143b**) [200,201]. The subsequent azidolysis of the esters (**143a**–**b**) gave azidoethylmorphine (**144**). Antitussive activity measured by the Gosswald method in rats was 60-fold higher for azidodionine (**144**) than for codeine (**2a**) (AtD_50_ (**2a,** oral) = 100.0 mg/kg, AtD_50_ (**144,** oral) = 1.67 mg/kg, ratio **2a**/**144** = 59.88) [197,198,202,203].

3-*O*-Morpholinylethylazidomorphine (**145**, azidopholcodine, Figure 41) [197,198,202] has also been prepared from morphine (**1a**) in five steps (morphine (**1a**) → ethylmorphine (**33**) → ethyldihydromorphine → 3-*O*-morpholinylethyldihydromorphine → 3-*O*-morpholinylethyldihydromorphine mesylate → **145**), and directly from azidomorphine (**130**) by alkylation with morpholinylethylchloride thanks to the stability of the azido group of **130** against the strong base NaOEt. Azidopholcodine (**145**) had four-fold greater antitussive activity in rats (AtD_50_ (**145,** s.c.) = 4.5 mg/kg) as compared to codeine (**2a**) (AtD_50_ (**2a,** s.c.) = 19.0 mg/kg, ratio **2a**/**145** = 4.22) [203].

To study antagonistic properties in the azidomorphine series, Makleit et al. synthesized the *N*^17^-substituted analogs (**149a**–**b**, cyclopropylmethyl, allyl, Figure 42) of azidomorphine (**130**), azidocodeine (**74a**), 14-hydroxyazidomorphine (**138**) and 14-hydroxyazidocodeine [204,205]. *N*^17^-Cyclopropylmethyl-norazidomorphine (**149a**, CAM) and *N*^17^-allyl-norazidomorphine (**149b**, AAM) were synthesized via two different routes. In the first approach, the acetylation of the phenolic hydroxy group in position-3 of azidomorphine (**130**) resulted in 3-*O*-acetylazidomorphine (**146**). The von Braun reaction with cyanogen bromide in the presence of the 6-azido group gave high yields (e.g., **146** → **147** in 90% yield). The alkaline hydrolysis of the cyanamide (**147**) and subsequent acid hydrolysis gave azidonormorphine (**148**) in 47% yield. The *N*^17^-alkylation of **148** with allyl bromide or cyclopropylmethyl bromide resulted in the corresponding *N*^17^-substituted norazidomorphines (**149a**–**b**). On the other hand, the partial acetylation of *N*^17^-cyclopropylmethyl-nordihydromorphine (**150a**) or *N*^17^-allyl-nordihydromorphine (**150b**) resulted in the corresponding 3-*O*-acetyl-nordihydromorphines (**151a**–**b**). The latter compounds were converted into their sulfonate esters (tosyl, mesyl, **152a**–**d**) for subsequent azidolysis, resulting in the desired analogs (**149a**–**b**).

In 1977, Knoll et al. [206] reported on their pharmacological investigations of numerous *N*^17^-substituted-azidomorphine analogs. They compared the OR antagonistic properties of *N*^17^-cyclopropylmethyl-norazidomorphine (**149a**, CAM, Figure 42), *N*^17^-allyl-norazidomorphine (**149b**, NAM) and their 14-hydroxy analogs—*N*^17^-cyclopropylmethyl-14-hydroxynorazidomorphine (COAM) and *N*^17^-allyl-14-hydroxynorazidomorphine (NOAM)—relative to a reference antagonist (naltrexone) in different tissues. Surprisingly, *N*^17^-substituted-azidomorphine analogs were potent antagonists in the isolated cat nictitating membrane (CNM) assay but were extremely potent pure agonists in the guinea pig ileum (GPI) and mouse vas deferens (MVD) tests. CAM (**149a**) [204,205] stimulated A-type receptors, which are involved in the behavioral effects of OR ligands (GPI and MVD tests; cholinergic neurotransmission). On the other hand, CAM (**149a**) had greater antagonist potency than naloxone in the CNM test (B-type receptor: responsible for the analgesic, antitussive and respiratory depressant effects of OR ligands; catechol adrenergic neurotransmission) [206]. Subsequently, Fürst et al. tested the antagonistic potencies of **149a**–**b** in the oxymorphone righting test [207]. The *N*^17^-substituted azidomorphines (**149a**–**b**) were more potent antagonists than nalorphine or pentazocine but were equipotent with naloxone. *N*^17^-Cyclopropylmethyl-norazidomorphine (**149a**, CAM) was suggested to be a κOR-sub-type agonist based on its agonistic activity in the rabbit vas deferens (RVD) assay and given the low potency of naloxone in reversing its agonistic effect in GPI and MVS tests [207,208]. CAM (**149a**) has a different pharmacological profile from bremazocine or ethylketazocine (EK), which are agonists in the CNM assay.

Inspired by the novel pharmacological profile of *N*^17^-substituted norazidomorphines (**149a**–**b**) [206], the Alkaloida research group prepared *N*^17^-substituted-norazidoethylmorphine derivatives (**155a**–**c**, R = CPM, allyl, *n*-propyl, Figure 43) [209] through two independent routes, from azidoethylmorphine (**144**) as well as from ethyldihydromorphine (**142**).

In brief, in the first reaction sequence [209], azidoethylmorphine (**144**, Figure 43) was *N*-demethylated using the von Braun method (BrCN) via *N*-cyano-azidoethylmorphine (**153**). The cyanamide (**153**) was hydrolyzed with 6% hydrochloric acid; the *N*-demethylation of azidoethylmorphine (**144**) was also achieved easily and in good yields with methyl- or phenyl-chloroformate. The formed *N*^17^-phenyl- or *N*^17^-methyl-oxycarbonyl-*N*^17^-desmethyl-azidoethylmorphine was converted to norazido-ethylmorphine (**154**) with 25% KOH in ethanol. The subsequent alkylation of *N*^17^-desmethyl-azidoethylmorphine (**154**) with cyclopropylmethyl bromide or allyl bromide resulted in *N*^17^-cyclopropylmethyl-norazidoethylmorphine (**155a**, ECAM, Figure 43) and *N*^17^-allyl-norazidoethylmorphine (**155b**) in yields of 46 and 54%, respectively.

In a second approach [209], the synthesis of *N*^17^-substituted-norazido-ethylmorphines (**155a**–**c**, Figure 43) was performed in six consecutive transformations starting from ethyldihydromorphine (**142**). The alcoholic hydroxyl group in position-6 of **142** was protected with an acetyl group by treatment with acetic anhydride, resulting in 6-*O*-acetyl-ethyldihydromorphine (**156**, 94%). The von Braun *N*-demethylation of **156** with cyanogen bromide and subsequent hydrolysis of the cyanamide (**157**) gave ethyldihydronormorphine (**158**). Other routes to *N*^17^-desmethyl-dihydromorphine (**158**) entailed the treatment of **156** with diethyl azodicarboxylate (DEAD) or with methyl- or phenyl-chloroformate. *N*-demethylation without the protection of the 6-OH function starting directly from ethyldihydromorphine (**142**) was also feasible. Compound **158** was alkylated with the appropriate reagent (cyclopropylmethyl bromide, allyl bromide, *n*-propyl bromide) to *N*^17^-substituted-ethyldihydronormorphines (**159a**–**c**). The compounds **159a**–**c** were converted into their mesyl esters (**160a**–**c**), and the subsequent azidolysis of these compounds gave target *N*^17^-substituted-norazidoethylmorphines (**155a**–**c**).

The agonist–antagonist characteristics of the synthesized analogs were determined by the hot-plate method with rats. A ten-fold higher morphine (**1a**) dose was necessary to achieve the same analgesic effect in conjunction with 5 mg/kg s.c. ECAM (**155a**) compared to the controls without antagonist co-treatment [209]. With the oral application of **155a** (25–30 mg/kg), a 2–3-fold higher morphine (**1a**) dose was required for an equianalgesic effect. Similarly to ECAM (**155a**), the application of *N*^17^-allyl-norazidoethylmorphine (**155b**) at 50 mg/kg (s.c.) raised tenfold the necessary morphine (**1a**) dose. Interestingly, ECAM (**155a**) showed an anorexic effect in food-deprived rats.

#### 2.9.2. 14-Hydroxy-8-azido-8-desoxyallopseudocodeine Derivatives

In 1988, Gulyás et al. [210,211] synthesized a series of 6- and 7-halo-substituted 14-hydroxy-8-azido-8-desoxyallopseudocodeine derivatives (Figure 44). The starting materials for these investigations, 6-demethoxythebaine (**105**), 6-chloro- and 6-bromo-6-demethoxythebaine (**118a**–**b**) and 7-chloro- and 7-bromo-6-demethoxythebaine (**113a**–**b**), were prepared according to earlier protocols of the Makleit group [143,144,152,153,157,158] (see also Section 2.8.4, Section 2.8.5, Section 2.8.6). 6-Demethoxythebaine (**105**) was obtained from neopine mesylate (**100**) with tetrabutylammonium fluoride [143,144], 6-chloro- and 6-bromo-6-demethoxythebaine (**118a**–**b**) from 14-chloro-codeine tosylate (**119a**) with LiCl or LiBr [157,158] and, finally, 7-chloro- and 7-bromo-6-demethoxythebaine (**113a**–**b**) from 7α-halo-neopine mesylates (**112a**–**b**) by boiling with potassium *tert*-butylate in methanol [152,153]. 6-Demethoxythebaine derivatives (**105**, **113a**–**b**, **118a**–**b**) were converted to the corresponding 8β,14β-epoxides (**161a**–**e**, 8β,14β-epoxycodides) with formic acid–hydrogen peroxide (low yields: 22–42%) or, alternatively, with disuccinylperoxide or *m*-chloroperbenzoic acid in 10% formic acid. The epoxides (**161a**–**e**) were subjected to azidolysis (5 equiv. NaN_3_, dioxane, H_2_O, 100 °C, 10 h) to give 6- and 7-substituted 14-hydroxy-8-azido-8-desoxyallopseudocodeine derivatives (**162a**–**e**) [210,211] in 28–63% yields.

The latter compounds were transformed to the corresponding 8α-amines (**163a**–**e**) with a yield of 28–63%. The acetolysis of the epoxides (**161a**–**e**) was performed with 10% acetic acid at 70 °C (1.5 h), which gave 8α-acetoxy-14-hydroxy derivatives (**164a**–**e**, 36–83%). The hydrolysis of the acetates with 10% KOH in ethanol resulted in 6- and 7-substituted 14-hydroxyallopseudocodeine derivatives (**165a**–**e**) [210,211] in 36–90% yield.

In a subsequent study, Gulyás and Makleit [212,213] investigated the reactions of 8β,14β-epoxycodide [214,215] (**161a**, Figure 45) containing a vinyl-oxirane sub-structural unit. The starting material (**161a**) [210,211] was prepared from 6-demethoxytebaine (**105**). The reduction of **161a** with lithium aluminum hydride in diethyl ether gave 14-hydroxydesoxycodeine-C (**166**) in 50% yield. When the azidolysis of 8β,14β-epoxycodide (**161a**) with azidotrimethylsilane in the presence of zinc (II) iodide was performed, the sole isolated product was 14-trimethylsilyloxy-8-azido-8-desoxyallopseudocodeine (**167**, 60%). Removing the TMS protecting group with sodium fluoride in methanol yielded the same 14-hydroxy-8-azido-8-desoxyallopseudocodeine (**165a**, 67%), which was prepared earlier from **161a** with sodium azide (dioxane-H_2_O, 100 °C, 10 h) [212,213]. When **161a** was treated with Cl^⊖^ anions in aqueous–acidic acetonitrile (LiCl, 10% HCl–CH_3_CN) [212,213], there was the formation of 6-chloro-14-hydroxyisocodeine (**168**, 14-hydroxy-α-chlorocodide, 20%), 14-hydroxy-isocodeine (**169**) and 14-hydroxy-allopseudocodeine (**106**).

Due to the difficulties encountered during the separation of the solvolysis products, the 6β,14β-dihydroxy (**169**, iso)/8α,14β-dihydroxy (**106**, allopseudo) product ratio was determined by a proton NMR spectrum analysis of the mixture, which gave a ratio of 5:6. The analogous reaction of 8β,14β-epoxycodide (**161a**) with Br^⊖^ ions in aqueous-hydrobromic acid–acetonitrile mixture (LiBr, 10% HBr–CH_3_CN) resulted in 8-bromo-8-desoxy-14-hydroxy-pseudocodeine (**170**, 14-hydroxy-β-bromocodide, 31%) as the main product, and a mixture of by-products **169**/**106** (ratio: 3:4) [212,213]. The formation of the main products with different structures, i.e., 6β-chloro (**168**) and 8β-bromo (**170**), is explicable by steric, electronic and stability factors, and is in agreement with the earlier results of Bognár et al. [216,217] concerning the nucleophilic substitutions of 14-hydroxydihydrocodeine tosylate (**73a**). The reaction of **73a** with F^⊖^ or Cl^⊖^ nucleophiles gave the corresponding 6-substituted-6-desoxy-14-hydroxydihydroisocodeine derivatives, whereas using Br^⊖^ anions resulted in 8-bromo-8-desoxy-14-hydroxypseudocodeine (**170**) [216,217]. The treatment of 8β,14β-epoxycodide (**161a**) with the organocuprate reagent lithium dibutylcuprate (LiCu(C_4_H_9_)_2_, prepared in situ from copper (I) iodide and butyllithium in diethyl ether–hexane at −78 °C) resulted in the 6β-butyl derivative (**171**) in 21% yield [212,213].

#### 2.9.3. Azido Derivatives of 6,14-Ethenomorphinans

Berényi et al. investigated the azidolysis of tosyl esters (**174a**–**g**) derived from different in ring-C bridged morphinans (Bentley compounds, or 6,14-ethenomorphinans [218,219,220,221], **173a**–**g**, Figure 46) [222,223]. The morphinandienes **4**, **105** and **118a** were transformed with methyl vinyl ketone or acrolein in a diastereoselective Diels–Alder reaction to the corresponding ring-C bridged 7-acetyl (**172a**,**c**,**d**) or 7-formyl (**172b**) adducts. Next, the keto group (**172a**–**d**) was reduced with NaBH_4_ in methanol to a mixture of diastereomeric secondary alcohols (**173a**, **b**, **d**–**g**) [222] or to the appropriate primary alcohol (**173c**) [223], which was tosylated (TosCl, pyridine, room temperature, 1–6 days) to yield **174a**–**g**. After separation by column chromatography of the diasteromerically pure tosylates (**174a**–**b**, **174d**–**g**), they were subjected to azidolysis (NaN_3_, 100 °C, 1.5 h). A novel class of morphinans containing a 4-azatetracyclo-[4.4.0.0^2.4^.0^3.8^]decane ring system (**176a**–**g**) was prepared by heating the azides (**175a**–**g**) in *N*,*N*-dimethylformamide at 100 °C for 24 h.

When longer reaction times were used for the azidolysis of the tosylates (**174a**–**g**), substitution and subsequent cyclization by the intramolecular addition of the azide to the Δ^18,19^ double bond resulted directly in the aziridine derivatives **176a**–**g**. The authors found a significant difference in the course of the reactions for 20*S*- and 20*R*- tosyl esters. After azidolysis of the 20*S*-tosylates (**174**), 7-ethylidene derivatives were isolated as major products resulting from an elimination reaction. However, the azidolysis of the 20*R*-tosylates (**174**) under identical conditions led to the formation the 20*S*-azides (**175**), by the inversion of the configuration. Despite the detection of the products by TLC, their isolation and characterization were possible in only a few cases (e.g., 20*S*-**175g**, and **175c**) due to their generally low stability. When the tosyl ester (**174c**) of the primary alcohol (**173c**) was subjected to azidolysis (5 equiv., NaN_3_, DMF, H_2_O, 100 °C, 1.5 h), the 7-azidomethyl derivative (**175c**) and aziridine (**176c**) were formed. After separation by column chromatography, the product ratio **175c**/**176c** was 9:1. The azide **175c** was converted to aziridine (**176c**, 59%) by heating in *N*,*N*-dimethylformamide at 100 °C for 24 h. The structure of **175c** was also proven by chemical reaction; its reduction (hydrazine hydrate, Raney-nickel, EtOH, reflux, 30 min) resulted in the primary amine 7α-aminomethyl derivative in 69% yield. In 1992, Batta et al. [224] reported the detailed NMR analysis of the 20*R*- and 20*S* diastereomeric alcohols (**173a**–**b**, **173d**–**g**), their tosylates (**174a**–**b**, **174d**–**g**), and also the 4-azatetracyclo [4.4.0^2.4^.0^3.8^] derivatives (**176a**–**f**). The authors [222,223,224] further discussed the differing behaviors of the 20*S* and 20*R* diastereoisomers, explaining the greater predisposition of the 20*S*-tosylates for elimination by the more advantageous, nearly antiperiplanar position of the 20-OTos group to the 7β-H.

The primary and secondary azides (**175a**–**g**, Figure 46) developed by Berényi et al. [222,223,224] proved to be unstable due to the intramolecular addition of the azido group into the Δ^18,19^ double bond. They degraded into azatetracyclodecane derivatives (**176a**–**g**, Figure 46) with nitrogen loss via heating or spontaneously at room temperature, and were thus unsuitable as ligands for pharmacological investigations. Therefore, the Makleit research group extended their investigations to the 20-tertiary azides of 18,19-dihydro-6,14-ethenomorphinans. In 1993, Sepsi et al. [225] described the synthesis of a series of tertiary azides with a morphinan scaffold.

The first strategy, which was the synthesis of the target azides via nucleophilic substitution of aryl or alkylsulfonate esters, failed due to the conversion of the tertiary alcohols (e.g., 20-methylthevinol, **173a**, Figure 47) into the corresponding 20-*O*-tosylate or mesylate (e.g., **177**). However, the use of hydrazoic acid and BF_3_-etherate [226] turned out to be practicable. 20-Methylthevinol (**173a**, Figure 47) thereby converted into the tertiary azide (**178**) in 62% yield, and its reduction to the corresponding tertiary amine (**179**) was successfully performed (83%) with hydrazine hydrate/Raney-Ni in ethanol (reflux, 30 min). As expected, the thermal treatment of **178** (DMF, 100 °C, 48 h) gave aziridine (**180**) as a result of the addition of the azido group into the 6,14-etheno bridge.

To avoid the aforementioned ring closure reactions, representatives of the 6,14-ethano series, tertiary alcohols (**182a**–**e**, Figure 48) prepared from dihydrothevinone (**181**) in Grignard reactions (RMgX, R = Me, Et, *n*Pr, Ph, *t*Bu), were chosen as starting materials for the synthesis of the target tertiary azides (**183a**–**e**). The replacement of the 20-OH tertiary hydroxyl group of **182b**–**d** (20*R*-**182b**, 20*R*-**182c**, 20*S*-**182d**, and 20*S*-**182e**) with an azido group using hydrazoic acid and boron trifluoride etherate resulted in a diastereomeric mixture of 20*R* and 20*S* of azides (**183b**–**d**, 35–44%) as major products, but also gave elimination products (**184b**–**d**, 5–13%). The diastereomeric azide ratios were determined from the ^1^H-NMR spectra based on the integrals of characteristic signals (20-CH_3_, 6-OCH_3_, 5β-H): **183b** (4:1), **183c** (3:1) and **183d** (1:1).

Efforts to split up the diastereomeric azides (**183b**–**d**) were unsuccessful, but there was the separation of the diastereomeric mixture of amines (**185a**–**b**), prepared from 20*R*-propyldihyrothevinol (**183c**) by reduction with hydrazine hydrate and Raney-Ni in boiling ethanol. The conversion of 20-*tert*-butyldihydrothevinol (**182e**) to the corresponding tertiary azide (**183e**) failed. Instead of **183e**, the furanocodide derivative (**190**) was isolated as the major product and the anhydro/olefin compound (**191**) as a minor by-product. Furanocodides are known acid-catalyzed rearrangement products of thevinols [227,228]. For the preparation of the tertiary azide with an orvinol structure (**189**), 20-methydihydrothevinol (**182a**) was 3-*O*-demethylated to 20-methyldihydroorvinol (**186**). The phenolic hydroxyl group of **186** was protected by the application of the Welsh method [99]. 3-*O*-Acetyl-dihydroorvinol (**187**) was reacted with hydrazoic acid and BF_3_ · Et_2_O to give the azide **188** (60%), which was deprotected with hydroxylamine hydrochloride to yield **189** (79%). The *tert*-azido-orvinol derivative (**189**) [225] as a lead compound was tested for OR agonist properties in the GPI test in vitro and the mouse hot-plate test (HP) in vivo, proving it to be a μOR agonist. In the mouse HP tests, **189** and **186** showed 10–18 times higher potency relative to morphine (ED_50_ (mg/kg) = 0.59 (**189**), 0.32 (**186**), 6.0 (**1a**). However, in comparison with the native 20-methyldihydroorvinol (**186**), the azido compound **189** did not show higher relative activity [225].

#### 2.9.4. 6-Azido-6-demethoxythebaine

In 1997, Csutorás et al. [229] reported the synthesis of 6-azido-6-demethoxythebaine (**193**, Figure 49) in six consecutive transformations starting from thebaine (**4**). 6-Azido-6-desoxy-14-hydroxyisocodeine (**87**) was prepared from **4** as described earlier by Makleit et al. (**4** → **132** → **66** → **70a**–**b** → **87**, Figure 49) [121,122]. The sulfonate esters (**70a**–**b**) were converted to 6-azido-6-desoxy-14-hydroxyisocodeine (**87**) by treatment with NaN_3_ in DMF/H_2_O at 100 °C for 4 h. A more prolonged reaction time under the same conditions gave 8-azido-8-desoxy-14-hydroxypseudocodeine (**85**) through rearrangement. Otherwise, the reaction of **87** with two equivalents of PBr_3_ resulted in 7β-bromo-6β-azidodesoxyneopine (**192**) in 77% yield. In this product (**192**), the trans-diaxial orientation of the 7β-bromo substituent relative to the 6α-hydrogen favors hydrogen bromide elimination, which is readily obtained by treatment with potassium *tert*-butoxide in ethanol (room temperature, 30 min, 75%). The structure of 6-azido-6-demethoxythebaine (**193**) was confirmed by IR and NMR spectra and by chemical reaction. It was stable at −20 °C, but decomposed within a few days at room temperature due to its vinylazide structural element. In a search for derivatives showing higher stability, the reaction of the azidodiene (**193**) with carbon disulfide in the presence of triphenylphosphine (reflux, 2 h) gave 6-isothiocyanato-6-demethoxythebaine (**194**) in 51% yield. The hetero Diels–Alder reaction of 6-azido-6,8-morphinandiene (**193**) with 4-phenyl-4*H*-1,2,4-triazoline-3,5-dione (PTAD) at 0 °C readily gave a stable cycloadduct (**195**) in excellent yield.

### 2.10. Fluorinated Morphinans

Halogenated derivatives of morphine alkaloids have been a subject of continuous interest in organic chemistry [22,24]. Brominated morphinans (e.g., 1-bromo and 1,7-dibromo derivatives of dihydrothebainone and β-dihydrothebainone, 1-bromo-nordihydrothebainone) were valuable intermediates for the first morphine total synthesis accomplished by Gates and Tschudi [230], and were later successfully used for the preparation of B/C-trans-fused 1-bromodihydrocodeinone [231] and for the synthesis of codeine from dihydrothebainone [232,233]. Iodinated and brominated morphinans are precursors for the synthesis of tritiated radioligands [234,235], which are valuable research tools for OR pharmacological investigations. The effects of fluorine incorporation in biologically active derivatives have been thoroughly investigated [236] and are the subject of a very recent comprehensive survey of fluorine-containing morphinans by Sandulenko et al. [237].

#### 2.10.1. Ring-C Fluorinated Morphinans

The morphine alkaloid research group at the University of Debrecen has paid particular attention to the synthesis of ring-C halogenated morphinan derivatives [24,238]. 6-Fluoro-6-deoxyisocodeine (**59a**, Figure 50) was obtained via the S_N_2 reaction of codeine tosylate (**63a**) with tetrabutylammonium fluoride (TBAF) [111,112] in 48% yield, and separately from pseudocodeine tosylate (**97**) [129,130] in an S_N_1′ reaction with the same nucleophile in acetonitrile, albeit in poor yield (10%). In 1972, Bognár et al. [216,217] described the synthesis of 6-fluoro-6-deoxy-14-hydroxyisocodeine (**80**) from 14-hydroxycodeine tosylate (**70a**) with TBAF. In an extension of earlier studies of the group [117,118,121,122], Somogyi et al. [239,240] investigated the S_N_2-type reactions of dihydrocodeine mesylate (**71b**) with halide anions (F^⊖^, Cl^⊖^, Br^⊖^, I^⊖^).

6-Fluoro-6-deoxydihydroisocodeine (**196a**, 6-fluorodihydrocodide, α-fluorodihydrocodide) was prepared from dihydrocodeine mesylate (**71b**) [239,240] with tetrabutylammonium fluoride in acetonitrile in low yield. 6-Chloro-6-deoxydihydroisocodeine (**196b**, 6-chlorodihydrocodide, α-chlorodihydrocodide) was also prepared by the heating of **71b** for 1 h with ten equiv. of lithium chloride in 75% yield. When **71b** was treated with LiBr in DMF at 100 °C for 43 h, 6-bromo-6-deoxydihydroisocodeine (**196c**, 6-bromodihydrocodide, α-bromodihydrocodide) was isolated in 26% yield. When the latter reaction was performed at higher temperatures (reflux for 4 h), desoxycodeine-C (**197**, Δ^7^-desoxycodeine) was formed in 63% yield. When **71b** was treated with 10 equiv. of sodium iodide, the corresponding 6-iodo derivative could not be isolated; instead, **197** was obtained in 29% yield.

In 1980, Boswell and Henderson [241] reported the fluorination of morphinan-6-ones (e.g., dihydrocodeinone (**49**), and 14-hydroxydihydrocodeinone) with diethylaminosulfur trifluoride (DAST). 6,6-Difluoro compounds were isolated as main products and 6-monofluoro-Δ^6,7^ unsaturated derivatives as by-products. In 1984, Ganti [242] synthesized numerous 14-fluoro-analogs of oxymorphone (**198a**, Figure 51), naltrexone (**198b**) and nalbuphine (**198c**) from 6-*O*-protected 14-hydroxy-dihydromorphinones with DAST.

In 1984, the research group of Kenner C. Rice [243,244,245,246,247,248] developed 6β-fluoro-oxymorphol (**204a**, foxy) and 6β-fluoro-naltrexol (**204c**, cyclofoxy) analogs (Figure 52). They performed the synthesis of 6-fluoro-6-desoxy-14-hydroxydihydroisomorphine derivatives (**204a**–**d**) in two steps from the corresponding 3-*O*-acetyl-6-trifluromethansulfonyloxy-14-hydroxydihydromorphines (**203a**–**d**).

Specifically, the Rice group prepared cyclofoxy (**204c**, cyF) by introducing the fluorine substituent in the 6β-position via an S_N_2 reaction by the displacement of the 6α-*O*Tf leaving group of **205c** with fluoride ions (potassium fluoride) in the presence of 18-crown-6 in acetonitrile. They then removed the acetyl protecting group by treating the fluorinated intermediate (**203c**) with NH_4_OH in methanol. For labeled foxy and cyclofoxy derivatives, see Section 2.12.

Woudenberg and Maat prepared chlorine-containing etorphine analogs from 7-chloro-morphinan-6,8-dienes (7-chloro-6-demethoxythebaine and 7-chloro-5β-methyl-6-demethoxythebaine) in multistep-syntheses [249], and subsequently evaluated their pharmacological characteristics, showing their generally high affinity for μOR and variable selectivity for κOR sub-types [250]. The Makleit group investigated Diels–Alder (DA) cycloaddition of the highly reactive dienophile 4-phenyl-4*H*-1,2,4-triazoline-3,5-dione (PTAD) to 7-halomorphinandienes (Cl, Br) [251]. In 1990, studying the DA reaction of thebaine with 2-fluoroacrolein, Jeong et al. observed the formation of a diastereomeric mixture of 7-fluoro-7-formyl-DA adducts (**207a**–**b**, 7α/7β, Figure 53) and a hetero-DA adduct [252]. The same research group reported the reactions of thebaine with trifluoromethyl-substituted acetylenic dienophiles (trifluoropropyne, hexafluoro-2-butyne) and with perfluoroaldehydes [253]. The hetero Diels–Alder (HDA) reaction of thebaine and 5-methyl-thebaine with trifluoroacetaldehyde resulted in 14-(trifluoro-2-hydroxy-ethyl)-5-methylcodeinone derivatives [254].

In 1992, Berényi and colleagues elaborated a new methodology for the synthesis of 6-fluoro-6-demethoxythebaine derivatives (**208a**–**c**, *N*^17^-substituent: CH_3_, H, *n*-propyl) from 14-hydroxy-6-fluoro-6-deoxyisocodeine [255]. They used microwave-promoted synthesis for the acid-catalyzed rearrangement of morphinans and fluoromorphinans (**208a**–**c**) [256] to yield the corresponding aporphine derivatives. Subsequently, they reported the cycloaddition reaction of 6-fluoro-6-demethoxythebaine with methyl-vinyl ketone and the synthesis and biological evaluation of 6-fluoro-6-demethoxy-20-methylorvinol (**209**) [257,258]. In 2013, Magnus performed the synthesis of 7β-fluorodihydrocodeine (**210**, Figure 53) [259]. The six-step transformation from codeine (**2a**) involved the fluorination of a 6,7-epoxyde intermediate with a hydrogen-fluoride–pyridine complex.

2-Haloaporphines are dopamine agonists that can be synthesized from halomorphinans by acid-catalyzed rearrangement. 2-Fluoro-*N*-*n*-propylnorapomorphine (2F-NPA) [255,260] is the most important member of this class, with extremely high affinity (K_i_ = 12 pM) and D_2_-sub-type selectivity (D_2_/D_1_ > 50,000) [261]. In 1994, Hosztafi and Makleit [262] reported the synthesis of numerous C-1 halogenated (1-chloro and 1-bromo) derivatives of codeine, morphine and their 7,8 dihydro analogs and, subsequently, the rearrangement of 1-halogenated-codeines to the corresponding ring-D-substituted 8-halogen-apomorphines [263]. The synthesis of [8,9-^3^H]apomorphine was accomplished by the bromination of apomorphine hydrochloride in trifluoroacetic acid (TFA) with molecular bromine [264] to 8,9-dibromo-apomorphine and followed by tritium dehalogenation (^3^H_2_, 10% Pd/C, ethanol). Filer [265] presented the radiosynthesis of 8,9-di-[^18^F]fluoroapomorphine by the fluorodestannylation of 8,9-di-(trimethylstannyl)-apomorphine. 8,9-Dibromoapomorphine and/or 8,9-diiodoapomorphine were applied as intermediates for the preparation of the labelling tin precursor.

#### 2.10.2. 1-Fluoro-Substituted Morphinans

In 1974, Lousberg and Weiss [266] reported the first synthesis of 1-fluorocodeine (**216a**) in 1974. Subsequently, Makleit and Dubina [267] published an improved method for its preparation through the pyrolysis of diazonium fluoroborate (**215a**) (Balz–Schiemann reaction). 1-Fluorodihydrocodeine (**216b**) was prepared from 1-fluorocodeine (**216a**) by heterogenous catalytic reduction [267].

In 2016, Hosztafi and Marton [268] synthesized 1-fluoro-substituted codeine derivatives (**220a**–**b**, *N*^17^-allyl, *N*^17^-*n*-propyl, Figure 54) and their 7,8-dihydro derivatives (**220c**–**d**). They first prepared 1-fluoromorphine (**217a**) and 1-fluorodihydromorphine (**217b**) from their respective codeine precursors via 3-*O*-demethylation with boron tribromide in chloroform. 1-Fluorocodeine (**216a**) and *N*^17^-*n*-propyl-1-fluorocodeine (**220a**) were subjected to acid-catalyzed rearrangement using methansulfonic acid (100 °C, 45 min) to give 8-fluoroapocodeine (**222a**) and *N*^6^-propyl-8-fluoroapocodeine (**222b**), respectively, with low yields (20–24%). 3,6-di-*O*-Acetyl-1-fluorodihydromorphine (**221**) was synthesized by the acetylation of 1-fluorodihydromorphine (**218b**) with acetic anhydride (100 °C, 1 h, 55%). In 2018, Hosztafi et al. reported the synthesis of 1-iodo-substituted codeine and dihydrocodeine derivatives [269].

#### 2.10.3. Fluorinated 6,14-Ethenomorphinans

In the past decade, the Moiseev research group expended considerable effort toward the synthesis of fluorinated 6,14-ethenomorphinan derivatives [237]. In 2016, they reported the synthesis of the key starting material for their investigations of the 7α-trifluoroacetyl analog of thevinone (**172a**)—21,21,21-trifluorothevinone (**224**, Figure 55)—starting with thebaine (**4**) [270]. Due to the low stability of the dienophile trifluoromethyl vinyl ketone, the direct conversion of **4** to **224** in a Diels–Alder reaction failed. This led them to adopt a three-step procedure in which thebaine was first converted in a [4 + 2] cycloaddition reaction with acrolein to the 7α-formyl cycloadduct (**172b**, thevinal, Figure 55). This was reacted with the nucleophilic trifluoromethylating Ruppert–Prakash reagent (TMSCF_3_) to yield the diastereomeric mixture of the secondary alcohols 20*R*- and 20*S*-**223**, with the generation of trifluoromethyl anion (CF_3_^⊖^) in situ from TMSCF_3_ in the presence of tetrabutylammonium fluoride (TBAF). The mixture of the epimeric alcohols (**223**) was transformed to the desired 21,21,21-trifluorothevinone (**224**) by Swern oxidation. Subsequently, Zelentsova et al. [271] prepared 21,21-difluorothevinone (**225**) and 21-fluorothevinone (**226**) via the defluorination of 21,21,21-trifluorothevinone (**224**) with magnesium and trimethylsilyl chloride, followed by the acidic hydrolysis of the TMS intermediate. The Grignard reaction of **224** with phenylmagnesium bromide gave a mixture of fluorinated 20*R*- and 20*S*-phenylthevinols (**227**) [270,272,273].

In 2023, Sandulenko et al. [274] synthesized numerous 20-methyltrifluoro-6-*O*-desmethylorvinols (**229**, **232**, **235**, Figure 56) starting from thevinone (**172a**) and *N*^17^-allyl-dihydronorthevinone (**233**). Notably, structure–activity relationships have only been reported so far for a few 6-*O*-desmethylorvinols [275]. The addition of Me_3_SiCF_3_ to thevinone (**172a**) resulted in 20*R*-**228**. Boiling the latter compound with 48% hydrobromic acid for 30 min led to 20-trifluoromethyl-6-*O*-desmethyl-orvinol (**229**, 60%) via an unusual simultaneous 3-*O*- and 6-*O*-demethylation. For the preparation of 20-trifluoromethyl-6-*O*-desmethyl-cyprenorphine (**232**), 20*R*-**228** was *N*-demethylated with ethyl azodicarboxylate to 20-trifluoromethyl-northevinol (**230**). Next, the secondary amine was acylated with cyclopropanecarbonyl chloride, whereupon the resulting *N*^17^-acyl compound reduced with lithium aluminum hydride to the *N*^17^-cyclopropylmethyl derivative (**231**). The *O*-Demethylation of **231** with boron tribromide in dichloromethane at −78 °C gave **229** in 76% yield. Analogously, *N*^17^-allyl-dihydronothevione (**233**) [276] with Me_3_SiCF_3_ gave 20*R*-**234**, which was *O*-demethylated to *N*^17^-allyl-20-trifluormethyl-6-*O*-desmethyl-dihydronororvinol (**235**, 53%). In general, thevinols with tertiary hydroxyl groups in position-20 are acid-sensitive compounds and thus prone to undergo, by acid-catalyzed dehydration, enol ether hydrolysis and rearrangement to 14-alkenylcodeinones, anhydro-20-alkylthevinols or 5,14-bridged thebainones. The authors attributed the suitability of BBr_3_ for the *O*-demethylation of fluorinated thevinols to the presence of the EWG CF_3_ group in the molecule [272]. The 20-CF_3_ group can prevent the formation of a carbocation from the C-20 tertiary alcohols and, accordingly, avoid the intramolecular rearrangements mentioned above. The target compounds (**229**, **232**, and **235**) administered s.c. were evaluated for analgesic activity in rodent tail-flick tests in comparison to morphine (**1a**). The results showed that the introduction of a fluorine substituent in position-20 of the orvinol scaffold did not eliminate the analgesic properties; 20-trifluoromethyl-6-*O*-desmethyl-orvinol (**229**) showed analgesic activity comparable to that of morphine (**1a**). 20-Trifluoromethyl-6-*O*-desmethyl-cyprenorophine (**232**) with *N*^17^-cyclopropylmethyl substituent proved to be a partial agonist with weak analgesic activity, and *N*^17^-allyl-20-trifluoromethyl-6-*O*-desmethyl-dihydronororvinol (**235**) displayed no analgesic activity.

In a subsequent study, the same research group placed their attention on the stereochemistry of the C-20 chiral center of fluorine-containing thevinols [277]. They thoroughly investigated the reactions of 21,21,21-trifluorothevinone (**224**, 7α-trifluoroacetyl, Figure 57) with various alkylmagnesium halides (MeMgI, EtMgBr, *n*PrMgBr, iPrMgBr, *t*BuMgCl) and alkyllithium reagents (*t*BuLi, *i*PrLi) and also the addition of TMSCF_3_ to thevinone (**172a**) and its analogs (7α-acetyl). These studies proved to be of fundamental importance in elucidating the structure–activity relationships of thevinols and orvinols with different C-20 configurations. The reaction of 7-acyl-type 6,14-ethenomorphinans with Grignard reagents results in complex product mixtures [57,278,279] due to various processes: Grignard addition, the reduction of the carbonyl function, and base-catalyzed rearrangement. Sandulenko et al. [277] found that the reaction of thevinone (**172a**) and its 7α-acyl analogs (7α-propionyl, 7α-butanoyl) with trimethyl-(trifluoromethyl)-silane in the presence of CsF in THF at room temperature gave a mixture of the tertiary alcohols 20*R*-**236** and 20*S*-**236**, with a predominance of the 20*R* isomer. The reaction of 21,21,21-trifluorothevinone (**224**) with two equivalents of methylmagnesium iodide in diethyl ether at room temperature resulted in a 100:67 mixture of the 20*R*-**236** and 20*S*-**236** epimeric tertiary alcohols. In the reaction of **224** with MeMgI, the researchers also studied the effects of the addition of metal salts (MgX_2_, X = Cl, I; ZnCl_2_, MeOMgI, *t*BuOK) on the product ratio, based on an earlier approach of Zelentsova et al. using one-dimensional ^19^F NMR spectra [273]. When **224** (Figure 57) was reacted with 1.2 equivalents of methyllithium in THF, a mixture of isomeric tertiary alcohols 20*R*-**236** and 20*S*-**236** (R = Me) was formed, with a predominance of the 20*S* epimer (e.g., at −78 °C, ratio 20*R*/20*S* = 25:100). Reacting 21,21,21-trifluorothevinone (**224**) with RMgX reagents other than MeMgI (R = Et, *n*Bu, *i*Pr, *t*Bu) gave a predominant formation of the secondary alcohols 20*R*-**237** and 20*S*-**237**.

In 2023, Zelentsova et al. [271] reported the synthesis of 21,21-difluorothevinone (**225**) and 21-fluorothevinone (**226**) by the defluorination of 21,21,21-trifluorothevinone (**224**, Figure 55 and Figure 57). Prominent members of the 6,14-ethenomorphinan series, buprenorphine, diprenorphine and dihydroetorphine, contain a saturated 6,14-ethano bridged C-ring [57,221,280]. Therefore, the next logical step was to try to synthesize the 18,19-dihydro analogs of **224**, **225** and **226**. The direct conversion of **224** to 18,19-dihydro-21,21,21-trifluorothevinone (**238**) by the catalytic hydrogenation of the Δ^19,19^ double bond failed. Even harsh reaction conditions at 60 Bar hydrogen pressure (AcOH, 10% Pd-C, 55–60 °C, 45 h) gave only a 15% conversion rate, with the isolation of the rearranged product **239** in 11% yield instead of the desired compound (**238**), which is the 7α-trifluoacetyl analog of dihydrothevinone (**181**). As an alternative route for the synthesis of *N*^17^-substituted-20-difluoromethyl-tevinols (**242a**–**c**), dihydrothevinone (**181**) was refluxed with the pronucleophile (difluoromethyl)trimethylsilane (Me_3_SiCHF_2_) in HMPA/THF in the presence of CsF to give 20*R*-**240** in 35% yield. The secondary amine 20*R*-**241** was prepared from 20*R*-**240** by *N*-demethylation with diethyl azodicarboxylate (DEAD) in acetonitrile. Subsequently, the target tertiary amines (**242a**–**c**) were prepared by functionalization with the corresponding alky or acyl halides with yields in the range of 48–52%. 20*R*-**242c** was also prepared directly from *N*^17^-allyl-dihydronorthevinone (**233**, Figure 58) with Me_3_SiCHF_2_ in 23% yield.

Aiming to synthesize the conjugates of 6,14-ethenomorphinans with bioactive molecules, the same research group prepared 7α- and 7β-carboxylic acid esters of thevinone, so-called thevinoic acid fluoroalkyl esters, from thebaine (**4**) with β-fluoroalkyl acrylates [281].

Very recently, Finke et al. [282] reported a procedure for the preparation of 6-trifluoromethyl-substituted morphinans from morphinan-6-ones. 6-Ketomorphinans (e.g., 14-hydroxycodeinone (**132**), 4-*O*-methylsinomenine, 1-iodo-4-*O*-methylsinomenine) were converted to the corresponding 6-trifluoromethyl compounds with (trifluoromethyl)trimethylsilane (Me_3_SiCF_3_, Ruppert–Prakash reagent) in the presence of Bu_4_NF or *t*BuOK.

### 2.11. Application of the Mitsunobu Reaction in the Morphine Series

The Mitsunobu reaction [283,284] continues to find wide use for the functionalization of alcoholic hydroxyl groups [285] and in the field of alkaloid synthesis [286]. During this reaction, a primary or secondary alcohol (R-OH, e.g., **2a**, Figure 59) substrate reacts with a pronucleophile (H-Nu (**243**), a compound containing a dissociable proton) in the presence of an azodicarboxylic acid ester (e.g., DEAD, DIAD, ADDP or DTBAD, **244**) and triaryl or trialkylphosphine (**245**, e.g., TPP or TBP, Figure 59). Following a complex reaction sequence, the alcohol (R-OH, **2a**) alkylates the conjugated base (Nu^⊖^) of the pronucleophile (H-Nu, **243**), forming R-Nu (**246**) as main product and the side products hydrazinecarboxylic acid alkyl ester and the corresponding phosphine oxide (e.g., Ph_3_P=O, **248**).

The Mitsunobu reaction is an indispensable alternative to classical S_N_2-type reactions. Taking place under mild (0–25 °C) and neutral conditions, the reaction operates in dipolar aprotic solvents (e.g., THF, benzene, DMF, acetonitrile). The reaction time (τ) is commonly between 0.5 and 12 h, with rare instances of a more prolonged τ-s. The Mitsunobu reaction is chemoselective: only the alcoholic hydroxyl groups react under such mild conditions. Typically, primary alcohols react more quickly than secondary alcohols, which enables the selective acylation of substrates when primary and secondary hydroxyl groups are present in the same molecule. The Mitsunobu reaction is usually stereoselective and yields products with opposite configurations (Walden inversion). Side reactions (elimination or allylic rearrangement) rarely occur. As pronucleophiles, molecules with dissociable protons (pK_a_ < 11) can be used, and examples of weak-acidity nucleophiles with higher pK_a_ values (<15) are also known. The mechanism of the Mitsunobu reaction is depicted in Figure 60.

(1)In the first elementary step of the process, a reactive betaine (**XVII**) is formed from the azodicarboxylic acid ester (**244**) and trialkyl or triarylphosphine (**245**) in an irreversible reaction;(2)In the second step, the reactive zwitterionic adduct (**XVII**) reacts with the acidic pronucleophile (H-Nu, **243**) to produce the nucleophile (Nu^⊖^);(3)The alcoholic hydroxyl group (**2a**) attacks the protonated betaine (**XVIII**) to form an alkoxyphosphonium intermediate (**XIX**). During this process, 1,2-hydrazinecarboxylic acid dialkyl ester (**247**) arises;(4)The conjugated base form (Nu^⊖^) of the pronucleophile (H-Nu, **243**) reacts with the alkylphosphonium intermediate (**XIX**), from which emerges the product (R-Nu, **246**) as well as triaryl or trialkylphosphine oxide (**248**). When a chiral secondary alcohol is applied as the starting substrate, a change in the configuration of the chiral center is to be expected (Walden inversion).

#### 2.11.1. The First Application

Robson and Kosterlitz [287] postulated that the denaturation of the OR binding sites in guinea pig brain homogenates could be achieved by the alkylation of the protein with phenoxybenzamine (PBZ). Indeed, the incubation of brain preparations with PBZ reduced the specific binding of [^3^H]dihydromorphine (μ-ORs) and [^3^H]*D*-Ala^2^-*D*-Leu^5^]enkephalin (δ-ORs). Conversely, the presence of OR ligands (peptides or opiates) protected against inactivation due to alkylating agents. Smith and Simon [288] found that treatment with the thiol (SH) alkylating reagent *N*-ethylmaleimide (NEM) inhibited the stereospecific binding of tritiated OR ligands (e.g., [^3^H]*D*-Ala^2^-*D*-Leu^5^]enkephalin). Bowen et al. [289] postulated that disulfide bonds in ORs are essential for ligand binding, proposing a three-state allosteric model (μ-agonist-, μ-antagonist- and δ-agonist-preferring states) under regulation by an “SH”–“S-S” (thiol–disulfide) exchange mechanism.

Based on this hypothesis, in 1988, Fujii et al. [290] investigated the possibility of introducing an SH group into the C-6 position of the morphine (**1a**) skeleton. In a first attempt, morphine (**1a**) was reacted with thioacetic acid in the presence of 1,1-dineopentyloxytriethylamine (toluene, 80 °C) to yield the 3-*O*-acetyl-6β-thioester (**250**, Figure 61, 59%) and the 6β-thioester (**249**, 30%). In a second method, morphine (**1a**) was reacted with thioacetic acid under Mitsunobu conditions (diisopropyl azodicarboxylate (DIAD), triphenylphosphine (TPP)). This approach gave the predominant formation of 6-*S*-acetyl-6-desoxy-isomorphine (**249**, 73%) and its 3-*O*-acetyl derivative (**250**, 22%). The absolute stereochemistry of the C-6 stereo center was proven by NMR spectroscopic methods. In the ^1^H-NMR spectra of compounds **249** and **250**, the coupling constant ^3^*J*_5β,6α_ was 0.5 Hz, which implies a 6β-orientation of the thioacetyl group [291]. Fujii et al. [290] went on to hydrolyze *S*-acetyl-6-desoxy-isomorphine (**249**) by treatment with 0.2 M potassium hydroxide in ethanol to give 6β-thiomorphine (**251**, 66%). Interestingly, the analogous reaction of dihydromorphine (**35**) with thioacetic acid (DIAD, TPP) resulted in 3-*O*-acetylmorphine (**254**) as the main product (93%) and the corresponding 3-*O*-acetyl-6β-thioacetyl derivative (**253**) as a minor product (7%). When dihydrocodeine (**67**) was subjected to the Mitsunobu reaction with thioacetic acid under identical conditions, there was the formation of the 6β-thioester (**255**) in almost quantitative yield. 3-*O*-Demethylation with boron tribromide in CHCl_3_ gave **256** in 86% yield. The latter compound was hydrolyzed with 0.2 M KOH solution in ethanol to afford 6βthio-7,8-dihydro-morphine (**257**). The reaction of 6β-thiomorphine (**251**) and 6β-7,8-dihydro-thiomorphine (**257**) with 2-nitrobenzenesulfenyl chloride (acetonitrile, 0 °C) gave the corresponding disulfides (**252**,**258**).

The analgesic activities of 6-*S*-acetyl-6-desoxy-isomorphine (**249**, Figure 61) and the disulfide (**252**) derivative were tested in guinea pig ileum (GPI) [290]. Compound **249** proved to be twice as potent as morphine (**1a**), whereas the disulfide (**252**) was quasi-equipotent with morphine (**1a**).

In 1990, Kanematsu et al. [292] synthesized *S*-activated 6-sulfhydryl-dihydoisomorphine (**253**) starting from dihydromorphine (**35**) using Mitsunobu conditions by a modification of the Fujii et al. [290,293] approach (5 eq. AcSH, 5 eq. DIAD and 5 eq. TPP). 6β-7,8-Dihydro-thiomorphine (**257**) was reacted with 5-nitro-2-pyridinesulfenyl chloride (CH_2_Cl_2_, 0 °C), giving the corresponding disulfide (**259**, 84%). The analgesic activity of the synthesized 6β-(5′-nitro-2′-pyridyldithio)deoxydihydromorphine (**259**) was tested in guinea pig ileum (GPI) and mouse vas deferens (MVD) preparations, giving IC_50_ values of 9.3 and 76 nM, respectively. In mice (MVD), compound **259** had approximately three times higher potency than morphine. The researchers also investigated the thiol–disulfide exchange reaction of the 5′-nitro-2′-pyridyldithio derivative (**259**) with *L*-cysteine methyl ester in DMF.

#### 2.11.2. Preparation of Isomorphine and Isocodeine Derivatives

At the beginning of the 1990s, there were only a few known morphine derivatives with 6-beta configuration (iso series, 6β-OH). Their preparation in low-yield procedures was only possible by the application of complicated multistep syntheses, with the further disadvantage of numerous side products. In 1969, Makleit and Bognár [123,124] prepared isocodeine (**2b**) in low yield (16%) by the acetolysis of codeine tosylate (**63a**) with 10% acetic acid. Subsequently, there was a report of the synthesis of isomorphine (**1b**) from 3-*O*-acetylmorphine tosylate (**69a**), and the preparation of dihydroisomorphine (**82**) from isomorphine (**1b**) by catalytic reduction [131,132]. Fleischhacker [294] found higher yields when performing solvolysis with 70% acetic acid. In 1971, Kirby and Massey [295] reported the synthesis of isocodeine (**2b**) from codeine (**2a**) in a three-step procedure: codeine (**2a**) → codeine tosylate (**63a**) → isocodeine acetate → isocodeine (**2b**). Codeine tosylate (**63a**) was converted into isocodeine acetate with hexadecyltrimethylammonium acetate, and the alkaline hydrolysis of the resulting 6β-acetate led to isocodeine (**2b**, 68% overall yield from **2a**). Simon [296] was the first from the Makleit group to investigate extensively the application of the Mitsunobu reaction in the field of morphine alkaloids.

In 1991, Simon et al. [297] thereby synthesized numerous isocodeine (**262a**–**c**, Figure 62) and isomorphine (**262g**–**j**) derivatives via the Mitsunobu reaction for the preparation of 6-benzoate esters (**261a**–**j**) and their subsequent hydrolysis.

This elegant two-step method [297] made available for the first time novel isocodeine isomorphine derivatives and previously known compounds of the iso series with acceptable yields. In the first step, codeine (**2a**), *N*^17^-substituted codeine derivatives (**260b**, **260c**, *n*Pr, allyl), 3-*O*-ethylmorphine (**260d**), 3-*O*-benzylmorphine (**260e**), 3-*O*-morpholinylethyl-morphine (**260f**), 3-*O*-acetylmorphine (**55**) or *N*^17^-substituted-3-*O*-acetylmorphine derivatives (**260h**–**j**, *n*Pr, allyl, cyclopropylmethyl) were reacted with benzoic acid as the acidic pronucleophile (H-Nu, **243**). The Mitsunobu reactions were conducted in the presence of diethyl azodicarboxylate (DEAD) and triphenylphosphine (TPP, **245**), resulting in the corresponding 6-*O*-benzoyl-isocodeine (**261a**–**j**) derivatives. The latter compounds were treated with a 10% aqueous potassium hydroxide solution to give the isomorphine and isocodeine derivatives (**262a**–**j**) with a free 6β-hydroxyl group.

In 1994, Friedmann et al. [298] investigated the influence of C-6 configuration on the OR affinity of a large number of *N*^17^-substituted-morphine derivatives (C-6α-hydroxy, *N*^17^-substituent: CH_3_ (**1a**), *n*-propyl (**260h**), allyl (**260i**), cyclopropylmethyl (**260j**) *N*^17^-substituted-isomorphine (C-6β-hydroxy, **1b**, **262h**–**j**)) and their 7,8-dihydro analogs [297]. They used in vivo rat tail-flick, mouse hot-plate and in vitro isolated GPI assays for pharmacological characterization [298]. The OR agonist activity of the *N*^17^-methyl derivatives (morphine (**1a**), isomorphine (**1b**), dihydromorphine (**35**) and dihydroisomorphine (**82**)) was determined by mouse hot-plate, rat hot-plate and rat tail-flick tests, in comparison with morphine (**1a**). The antinociceptive actions of the C-6α and C-6β derivatives did not differ significantly, having relative potencies 0.6–1.9-fold that of morphine (**1a**). In the isolated electrically stimulated GPI preparation, isomorphine (**1b**) was twice as potent as morphine (**1a**), but dihydroisomorphine (**82**) showed a potency only 1.18-fold that of dihydromorphine (**35**) relative to normorphine. The potencies relative to normorphine were in the range of 1.1–4.4. The epimerization of C-6α to C-6β derivatives had only slight effects on OR activities in vitro but influenced the antagonistic effect of the *N*^17^-cyclopropylmethyl and *N*^17^-allyl compounds. *N*^17^-Allyl-dihydronorisomorphine (AD_50_ = 4.0 mg/kg, (s.c) (3.2–7.3), showed a 10-fold increase in antagonist potency compared to the parent *N*^17^-allyl-normorphine (**260i**, 0.48 (0.4–0.57)) in the rat tail-flick test vs. morphine.

#### 2.11.3. Reactions of Codeine Isomers and Neopine

In the previously presented experiments, the starting secondary alcohols carried the hydroxyl group at position 6α- of the C-ring of the morphinan skeleton. In 1993, Simon [296] investigated the Mitsunobu reaction of codeine isomers (isocodeine (**2b**), allopseudocodeine (**2c**) and pseudocodeine (**2d**)) with various pronucleophiles (H-Nu compounds) by the application of diethyl azodicarboxylate (DEAD) and triphenylphosphine (TPP). These reactions were performed under exactly the same conditions as those described above for 6α-hydroxyl derivatives [297].

Upon subjecting isocodeine (**2b**, 6β-OH, Figure 63) to the Mitsunobu reaction with benzoic acid or phthalimide, there was no conversion. A previous investigation of the nucleophile substitution reactions of isocodeine tosylate (**81**) by Makleit [115] found the formation of pseudocodeine derivatives due to allylic rearrangement. In the Mitsunobu reaction of pseudocodeine (**2d**, 8β-OH) with 4-nitrobenzoic acid or phthalimide, 6β-substituted derivatives (isocodeine series) were formed due to [3,3-sigmatropic] rearrangement. Presumably, the attack of the corresponding nucleophile from the α-side is sterically hindered in an S_N_2-type reaction. Makleit, Somogyi and Bognár [129,130] had similar experiences during investigations of the nucleophile substitution reactions of pseudocodeine tosylate (**97**). They found the formation of 6-deoxy-6-substituted-isocodeines (**99**) or 8-deoxy-8-substituted-pseudocodeine derivatives (**98**), depending on the nature of the nucleophilic anion [129,130].

There was no observable reaction when allopseudocodeine (**2c**, 8α-OH) was treated with benzoic acid or 4-nitrobenzoic acid under Mitsunobu conditions [296]. Interestingly, when phthalimide was applied as pronucleophile, there was the formation of the 8β-phthalimido derivative (**98**, Nu = NPht, pseudocodeine series), a result corresponding to the Mitsunobu reaction mechanism (8α-hydroxyl (**2c**) → 8-phthalimido-pseudocodeine (**98**)). The authors proved the structure of the synthesized phthalimido compound (**98**) by NMR spectroscopy and by chemical reaction. The reaction of the 8β-phtalimido-compound (**98**, Nu = NPht) with hydrazine hydrate gave the same known 8β-amine (8-amino-pseudocodeine (**84**)) as that prepared earlier by Bognár and Makleit [117,118] via another route (codeine (**2a**) → codeine tosylate (**63a**) (or mesylate, **63b**) → 8-azidocodide (**83)** → 8-aminocodide (**84**)).

When neopine (**3a**) was chosen as a starting material, the replacement of the C-6α secondary alcohol using benzoic acid or phthalimide as a Mitsunobu pronucleophile failed [296], although in both cases, there was 6-demethoxythebaine (**105**) present in the product mixture. Likewise, when the reaction was performed in the absence of the pronucleophile (H-Nu), only the presence of the elimination product, 6-demethoxythebaine (**105**), could be detected. In the neopine (**3a**) molecule, the 6α-hydroxyl group occupies a position pseudo-axial to the 7β-proton, which represents a favorable steric condition for elimination. The elimination reactions of 6-*O*-mesyl-neopine (**100**) derivatives are well known from the literature [143,144,152,153,299].

#### 2.11.4. Synthesis of 6β-aminomorphinans

The conversion of codeine 6-sulfonesters (**63a**, **63b**) to 6β-halo-substituted derivatives by nucleophilic substitution (S_N_2) was achievable only for F^⊖^ and Cl^⊖^ anions [110,111,112,113,114]. The reaction of the sulfonesters with Br^⊖^ or I^⊖^ nucleophiles occurred via a complex mechanism (S_N_2 + S_N_i’), resulting in 8β-substituted products due to a [3,3] sigmatropic rearrangement.

We note that for the same reasons, the application of the N_3_^⊖^ nucleophile and the subsequent reduction of the formed azide gave 8β-amino derivatives. Morphinan-6-ones were employed earlier for the production of 6-amino-4,5-epoxymorphinans via reductive amination, which resulted in 6α/6β-amino epimer mixtures [300,301,302]. The Makleit research group developed a stereoselective method for the preparation of 6β-aminomorphinans [303]. In 1992, they extended their investigations [303] to the synthesis of numerous *N*^17^-substituted-6β-aminocodeine (**266a**–**d**) and *N*^17^-substituted-6β-aminomorphine (**266e**–**h**, Figure 64) derivatives. The carefully selected starting morphinan derivatives (**263a**–**h**) all contained a Δ^7,8^ double bond in ring-C. By the treatment of codeine (**2a**), *N*^17^-substituted-codeine (**263b**–**d**), 3-*O*-acetylmorphine (**55**) and 3-*O*-acetyl-*N*^17^-substituted-morphine (**263f**–**h**) derivatives with phthalimide under Mitsunobu conditions (DEAD, TPP, benzene), the corresponding 6β-phthalimido-codeines (**264a**–**d**) (68–92%) and 6β-phthalimido-3-*O*-acetylmorphines (**264e**–**h**) (39–56%) were formed with an inversion of the configuration. Side reactions, e.g., allylic shifts, were not expected. 6β-Phthalimido-morphines (**265a**–**d**) were prepared by the 3-*O*-deacetylation of the compounds **264e**–**h** with hydroxylamine hydrochloride in aqueous ethanol (50 °C, 10 min). The 6β-phthalimido compounds (**264a**–**h**) were subjected to hydrazinolysis (see also Gabriel synthesis) to give the primary amine 6β-amino derivatives (**266a**–**h**) in 50–98% yield. For the preparation of 6β-amino-7,8-dihydro derivatives (**267a**–**f**), the selected Δ^7,8^ compounds (**266a**,**b**,**d**,**e**,**f**,**h**) were saturated under heterogenous catalytic conditions (H_2_, 10% Pd-C, EtOH). 6β-Amino-dihydrocodeines (**267a**–**c**) and 6β-amino-dihydromorphines (**267d**–**f**) were obtained in 37–75% yield.

The Makleit group [304] accomplished the Mitsunobu reaction of *N*^17^-demethyl-*N*^17^-substituted-14-hydroxycodeines (**268a**–**d**, CH_3_, *n*Pr, allyl, cyclopropylmethyl) as well as *N*^17^-demethyl-*N*^17^-substituted-3-*O*-acetyl-14-hydroxymorphines (**268e**–**h**, Figure 65) with phthalimide.

The 6β-phthalimido derivatives (**269a**–**h**) were prepared with yields in the range of 20–65%, without any observed allyl-migration. The treatment of these compounds with hydrazine hydrate in ethanol yielded 6β-amino-14-hydroxycodeine (**271a**–**d**) and 14-hydroxymorphine (**271e**–**h**) derivatives. Most of these 6β-amino derivatives were previously unavailable via the sulfonate ester → azide → amine route [54,55] or by the reductive amination of morphinan-6-ones [300,301].

Continuing their systematic investigations, the Makleit group extended the epimerization studies to compounds saturated in ring-C (7,8-dihydro compounds): dihydromorphines, dihydrocodeines and their 14-hydroxy analogs [305]. When benzoic acid was applied as a pronucleophile in the Mitsunobu reaction of dihydrocodeine (**67**), there was incomplete conversion. Interestingly, by the application of *p*-nitrobenzoic acid instead, the corresponding dihydroisocodeine *p*-nitrobenzoate was isolated in a yield of 90%. Accordingly, 14-hydroxydihydrocodeine (**68**) and *N*^17^-substituted-14-hydroxydihydrocodeine derivatives (**273b**–**c**, Figure 66) were reacted under Mitsunobu conditions (TPP, DEAD) with 4-nitrobenzoic acid in anhydrous benzene to yield the corresponding *p*-nitrobenzoic esters (**274a**–**h**). Ester cleavage of these compounds was performed by alkaline hydrolysis, giving the desired 6β epimers (**275a**–**h**) with yields in the range of 37–90%. In this manner, the authors achieved a new stereoselective synthesis of the human metabolites of naloxone (β-naloxol (**275g**)) and naltrexone (**199**, β-naltrexol (**275h**)).

In a second series of experiments [305], dihydrocodeine- (**67**), 3-*O*-acetyldihydromorphine-(**54**), 14-hydroxydihydrocodeine-(**68**, **276b**–**d**, X = OH) and 3-*O*-acetyl-14-hydroxydihydromorphine derivatives (**276e**–**h**, X = OH, Figure 67) were treated with phthalimide in benzene in the presence of triphenylphosphine and diethyl azodicarboxylate (DEAD) to yield the corresponding 6β-phthalimido compounds (**277a**–**h**). For the hydrolysis of the phenol esters (**277e**–**h**), the researchers used hydroxylamine hydrochloride in aqueous ethanol, giving the corresponding 6β-phthalimido-14-hydroxydihydomorphine derivatives (**278a**–**d**) with a free phenolic hydroxyl group in 58–70% yield. The treatment of the 6β-phthalimido derivatives (**277a**–**h**) with hydrazine hydrate led stereoselectively to the 6β-amino compounds (**279a**–**h**) in 51–90% yield. At the time, these investigations of stereoselective syntheses of 6β-oxymorphamine (**279e**, R_2_ = CH_3_), 6β-naloxamine (**279g**, R_2_ = allyl) and 6β-naltrexamine (**279h**, R_2_ = cyclopropylmethyl) were considered novel. Szilágyi et al. presented a detailed NMR analysis of the prepared new compounds [306], and Fürst et al. subsequently undertook their pharmacological characterization [298,307].

#### 2.11.5. Synthesis of 6β-Succinimido Derivatives

Upon reacting codeine (**2a**) with the succinimide pronucleophile (HSu) [303] under identical Mitsunobu conditions to those described above (DEAD, TPP, benzene), 6β-succinimido-codeine (**280a**, Figure 68) was obtained in 80% yield. In the case of 3-*O*-acetylmorphine (**55**), only the isolation of the deacetylated 6β-succinimido-morphine (**281b**) was feasible. The saturation of the Δ^7,8^ double bond under heterogenous catalytic conditions (H_2_, 10% Pd-C, RT, atmospheric pressure) was only achievable for 6β-succinimido derivatives (**280a**, 73%, **280b**, 55%). In the case of the appropriate 6β-phthalimido compounds (**264a**–**h**), the reduction of the Δ^7,8^ double bond was not realizable under atmospheric pressure.

#### 2.11.6. Reaction of 14-Halogenocodeines

14-Halogenocodeines (**282a**–**b**, Figure 69) contain both an allyl alcohol (C^8^=C^7^-C^6^-OH) and allyl halide (C^7^=C^8^-C^14^-X) sub-structural unit within the same molecule. Thebaine (**4**) reacted with *N*-halosuccinimides (NCS, NBS) in acetic acid or acetone–water (2:1 (*v*/*v*)) gave 14-halocodeinones [155].

The reduction of codeinone derivatives (e.g., codeinone (**24**), 14-hydroxycodeinone (**132**), 1-bromocodeinone and 14-bromocodeinone (**109**)) with sodium borohydride led stereoselectively to the corresponding codeines (e.g., **282a**–**b**, 6α-hydroxy compounds). 14-Chlorocodeine (**282a**) [138,139] and 14-bromocodeine (**282b**) [155] are useful starting materials for the synthesis of neopine derivatives. The reduction of neopinone (**23**) with NaBH_4_ results in a mixture of neopine (**3a**) and isoneopine (**3b**). Okuda et al. [160,308] found that the NaBH_4_ reduction of 14-bromocodeinone (**109**) in methanol–water gave three isolated products: the main product neopine (**3a**, 46%), isoneopine (**3b**, 6%) and indolinocodeine (28%).

In 1993, Simon et al. [309] investigated the Mitsunobu reaction of 14-halogenocodeines (**282a**–**b**, Figure 69) with phthalimide as a pronucleophile in the presence of triphenylphosphine (TPP) and dialkyl azodicarboxylates (DEAD or DIAD) or diphenyl azidophosphate (DPPA). The reaction of 14-chlorocodeine (**282a**) with phthalimide (TPP, DEAD, benzene) resulted in three products. The main product, a Δ^6,8^-conjugated diene (**285a**) containing a hydrazine dicarboxylic diethyl ester substituent in position-6, was obtained in 36% yield. Further products, namely 6β-phthalimido-14-chloro-6-deoxycodeine (**283**, 12%) formed by inversion and 6β,14-dichloro-6-deoxycodeine (**284a**, 13%), were isolated by column chromatography in a ratio of about 1:1. Interestingly, when 14-bromocodeine (**282b**) was reacted with phthalimide under identical conditions (TPP, DEAD, benzene, RT, 1 h), there was no observed formation of a 6β-phthalimido derivative, presumably due to steric hindrance by the bulky 14-bromo substituent. In this case, the main product was the thebaine analog (**285a**, 34%), followed by 6β,14-dibromo-6-deoxycodeine (**284b**, X = Br, 25%). To prove the structure of the Δ^6,8^-diene (**285a**) with the hydrazine dicarboxylic diethyl ester substituent in position-6, it was boiled with methyl vinyl ketone to yield the corresponding DA adduct thevinone analog (**286a**). In a second approach, 14-halogenocodeines (**282a**–**b**) were reacted under the above-mentioned conditions (TPP, DEAD) in the absence of a pronucleophile (H-Nu). The formation of 6β,14-dichloro-6-deoxycodeine (**284a**, 32%) and 6β,14-dibromo-6-deoxycodeine (**284b**, 25%) as well the Δ^6,8^-diene (**285a**; 32%, from **282a** and 25% from **282b,** respectively) was observed. The authors [309] proved the structure of 6β,14-dihalo-6-deoxycodeines (**284a**–**b**) by spectroscopy and chemical reactions. When these derivatives (**284a**–**b**) were allowed to react in *N*,*N*-dimethylformamide in the absence of a nucleophile [159], the same 6-halogeno-6-demethoxythebaine (**118a**–**b**) derivatives were obtained as had been prepared earlier in the nucleophilic substitution reaction of 14-chlorocodeine tosylate (**119a**) with Cl^⊖^ and Br^⊖^ ions, respectively (see Section 2.8.6) [156,158]. The formation of the 6β,14-dihalo-6-deoxycodeines (**284a**–**b**) and Δ^6,8^-dienes (**118a**–**b**) in this reaction was explained by the in situ liberation of hydrogen halides (HCl, HBr), which could then act as pronucleophiles in the Mitsunobu reaction. To substantiate the presence of the in situ-generated H-Nu compound, the hydrochloride salt of codeine (**2a**) was allowed to react under Mitsunobu conditions (TPP, DEAD), which gave α-chlorocodide (**59b**, Figure 69) in 70% yield. In another approach, 14-halogenocodeines (**282a**–**b**) were reacted with diphenylphosphorylazide (DPPA) under Mitsunobu conditions, giving the 6β-azido-14-halogeno-6-deoxycodeine derivatives (**287a**, X = Cl, 37%, **287b**, X = Br, 25%).

The reaction of morphine (**1a**) or codeine (**2a**) with thionyl chloride resulted in α-chloromorphide (**288a**) and α-chlorocodide (**59b**), respectively [110]. Notably, the reactions of codeine (**2a**) with thionyl bromide [110] or codeine tosylate (**63a**) with lithium bromide [110] both resulted in β-bromocodide (**65b**, Figure 17), the product of thermodynamic control. When hydrogen halide salts (hydrochloride or hydrobromide) of codeine (**2a**) or morphine (**1a**) were allowed to react under Mitsunobu conditions (DEAD, TPP, toluene) in the absence of other nucleophiles [310], 6β-halogen-substituted derivatives were isolated (e.g., from codeine hydrochloride → α-chlorocodide (**59b**), from codeine hydrobromide → α-bromocodide (**288c**), Figure 70). The importance of the above-mentioned results of Simon et al. [310] have to be emphasized, as this was the first isolation of α-bromocodide (**288c**), the product of kinetic control. Analog reactions starting from the hydrogen halide (HCl, HBr) salts of 14-hydroxy derivatives (14-hydroxycodeine (**66**), 14-hydroxymorphine (**289a**), 14-hydroxydihydrocodeine (**68**) and 14-hydroxydihydromorphine (**289b**)) led to the corresponding 6β-halogen (Cl, Br)-substituted compounds (**290a**–**d**). The reactions of Δ^7,8^ unsaturated derivatives (**1a**, **2a**, **66**, **289a**) were complete within 1 h, but the conversion proceeded much slower (3–4 h) in the 7,8-dihydro series (**68**, **289b**).

#### 2.11.7. Novel Applications

In this section, we present a few novel examples highlighting potential applications of Mitsunobu reactions in the field of semisynthetic morphinans, as informed by the results of the Makleit group in the 1990s. In 2014, Fujimura et al. [311] reported the pharmacological characteristics of the OR ligand TRK-130 (**278d**, naltalimide, *N*-[(5*R*,6*R*,14*S*)-17-cyclopropylmethyl-4,5-epoxy-3,14-dihydroxymorphinan-6-yl]phthalimide, Figure 71). The compound TRK-130 (**278d**) proved to be a selective partial μOR agonist (K_i_ for μOR = 0.268, δOR = 121 and κOR = 8.97 nM, respectively, δ/μ = 451, κ/μ = 33.5) [311] in radioligand-binding assays with human ORs. Naltalimide (**278d**) is used as a prophylactic agent against urinary incontinence, i.e., overactive bladder (OAB) [311]. The synthesis of naltalimide (**278d**) was achieved by the Simon et al. method [305] from 3-*O*-acetyl-*N*^17^-cyclopropylmethyl-14-hydroxydihydromorphine (3-*O*-acetyl-α-naltrexol, **276h**, route: **276h** → **277h** → **278d**, X = OH, see also Figure 68) in a Mitsunobu reaction with phthalimide. Izumimoto et al. [312] developed an alternative synthesis route for naltalimide (**278d**) and its *N*^17^-allyl analog (**278c**) from β-naltrexamine (**279h**) [305] and β-naloxamine (**279g**) [305], respectively. The corresponding 6β-amine (**279g**–**h**) was reacted with phthalic anhydride in DMF in the presence of triethylamine at 140 °C [312] to yield the 6β-phthalimido derivatives (**278c**–**d**) with yields of 34–58%.

To potentially treat substance dependence, the research group of Kenner C. Rice developed heroin vaccine haptens [313,314] (Heroin Hapten (MorHap, **291**, Figure 72)). One of the chosen hapten scaffolds was 6β-amino-6-desoxymorphine (**266e**). For its preparation, they applied the method described earlier by the Makleit group [303]. In brief, they selectively acetylated morphine to 3-*O*-acetylmorphine (**55**) by the Welsh method [99]. They treated this product with phthalimide under Mitsunobu conditions (TPP, DIAD, toluene, RT, 2 h) to yield 6β-phtalimido-3-*O*-acetylmorphine (**264e**, 88%). The stereochemistry of the phthalimido derivative (**264e**) was substantiated by X-ray crystallographic analysis. The cleavage of the phthalimido protecting group with hydrazine hydrate in ethanol (55 °C, 90min) led to 6β-amino-6-desoxymorphine (**266e**) in 81% yield. Subsequently, the free 6β-amine was acylated with the activated ester 2,5-dioxopyrrolidin-1-yl 3-(tritylthio)propanoate to the desired hapten (**291**, MorHap). Finally, the hapten (**291**, MorHap) was conjugated to the carrier protein (tetanus toxoid (TT) or cross-reactive material 197 (CRM_197_)). Antibodies formed due to the immunization effect of the TT-MorHap conjugate led to nearly complete protection against heroin dependence in mouse experiments.

Scammels et al. [315] prepared fluorescently labeled morphinan-sulfo-Cy5 conjugates for the visualization of ORs in living cells. For these bioimaging investigations, they developed a fluorescent partial OR agonist ligand. For the synthesis of C-6 amide pre-congeners, first, 3-*O*-(4-methoxybenzyl)-6-amino-6-desoxymorphine (**296**, Figure 73) was synthesized from morphine (**1a**) in five steps. The phenolic hydroxyl group of **1a** was protected as PMB (*p*-methoxy-benzyl) ether. Next, 3-*O*-PMB-morphine (**292**) was reacted with benzoic acid under Mitsunobu conditions (DIAD, TPP, toluene) to form the benzoate ester (**293**). The latter compound (**293**) was converted into the 3-*O*-PMB-isomorphine (**294**) by saponification with 1M potassium hydroxide in aqueous ethanol solution. The 6-beta-hydroxyl function of **294** was transformed to a 6α-amino group via a second Mitsunobu reaction and subsequent phthalimide cleavage. 3-*O*-PMB-isomorphine (**294**) was reacted with phthalimide (DIAD, TPP, toluene) to give 6α-phthalimido-3-*O*-PMB-morphine (**295**). The treatment of this latter compound with hydrazine hydrate in ethanol resulted in 3-*O*-PMB-6-amino-6-desoxymorphine (**296**).

In 1998, the research group of Nagase [316] developed nalfurafine (**300**, TRK-820, 17-cyclopropylmethyl-3,14-dihydroxy-4,5-epoxy-6β-[*N*-methyl-*trans*-3-(3-furyl)-acrylamido]morphinan, Remith^®®^, Figure 74), a 4,5-epoxymorphinan-type OR ligand. TRK-820 (**300**) was a selective, very-high-affinity agonist for κ-ORs, and a partial agonist for μ-ORs and δ-ORs (K_i_ (κ-OR) = 75 pM, K_i_ (μ-OR) = 5.2 nM, K_i_ (δ-OR) = 161 nM, μ/κ = 69.3, δ/κ = 2146) [317]. TRK-820 (**300**) is an approved pharmaceutical for the treatment of hemodialysis-related uremic pruritus and for patients with chronic liver diseases. Nalfurafine (**300**) was prepared from naltrexone (**199**) via *N*-methyl-β-naltrexamide (**298**). The secondary amine (**298**) was acylated with 3-(3-furyl)acryloylchloride (**299**) to yield **300**. Recently, Suzuki et al. [318] synthesized nalfurafine (**300**) and its 10α-hydroxy analogs from 3-*O*-acetyl-α-naltrexol (**276h**) via a modified synthesis route by the application of the Mitsunobu reaction.

In the previous decade, 6β-acylaminomorphinans [319] and 6β-pyridinyl amidomorphinans were synthesized [320] starting with precursors arising from earlier work by the Makleit group [303].

### 2.12. Poppy Alkaloids as Starting Materials for Molecular Imaging

Molecular imaging by positron emission tomography (PET) or single-photon emission computed tomography (SPECT) enables the detection of one or more sub-types of ORs in healthy brains and in pathologies involving opioid neurotransmission [35,321,322,323]. OR PET radiochemistry development has benefited from the foundational work on opioid chemistry described above, which presented important lead compounds for radiolabeling. As with the lead compounds, some classes of OR PET ligands have incomplete sub-type selectivity and differ with respect to their agonist–antagonist binding profile. Furthermore, the extreme potency of some opioid agonists calls for particular attention to the molar activity of OR agonist tracers.

OR PET imaging began with development of the μ-OR-selective agonist 4-anilidopiperidine derivative carfentanil ([^11^C]caf) [324] and the μ/κ OR-selective antagonist morphinan foxy ([^18^F]**204a**, [^18^F]foxy) and cyclofoxy derivatives ([^18^F]**203a**, acetyl-[^18^F]cyclofoxy; [^18^F]**204c**, [^18^F]cyclofoxy ([^18^F]FcyF); [^11^C]**204c**, [^11^C]cyclofoxy ([^11^C]cyF)) [243,244,245,246,247,248] (Figure 75). The synthesis of the [^18^F]foxy and [^18^F]cyclofoxy derivatives starts from the poppy alkaloid thebaine (**4**) [325] and proceeds with conversion to 14-hydroxycodeinone and the further synthetic manipulation of the C-ring to introduce the 6β-fluoro group by nucleophilic substitution [244] (see Figure 52).

Following the synthesis of the δ-OR-selective antagonist *N*1′-([^11^C]methyl)naltrindole ([^11^C]MeNTI, (K_i_ (μ) = 14 nM, K_i_ (δ) = 0.02 nM, K_i_ (κ) = 65 nM) [326], and *N*1′-(2-[^18^F]fluoroethyl)naltrindole ([^18^F]FE-NTI, BU97001, affinities) was prepared [327].

Researchers have developed a range of ^11^C- and ^18^F-labeled cyclofoxy derivatives (Figure 75), namely [^11^C]cyclofoxy ([^11^C]cyF [247]), [^18^F]foxy [243], [^18^F]acetylcyclofoxy [245,246] and [^18^F]cyclofoxy ([^18^F]FcyF) [243,248]. Among these, the μOR/κOR antagonist [^18^F]FcyF has served in a study of OR availability in methadone-treated former opioid addicts [248], revealing lower OR availability in the striatum as compared to healthy volunteers, in measurements of occupancy by methadone.

Notably, Pasternak et al. reported on the synthesis of radioiodinated iodobenzoyl derivatives of **279e**, **279g** and **279h**, 6β-oxymorphamine (^125^I-BOxyA), 6β-naloxamine (^125^IBNalA) and 6β-naltrexamine (^125^IBNtxA) in 2011 [328], with potential application for SPECT imaging.

6,14-Ethenomorphinans (orvinols/Bentley compounds) [57,280,329] are ring-C bridged semisynthetic derivatives of the poppy alkaloid thebaine [325] and/or oripavine [330,331,332]. Their radiolabeled derivatives are the most frequently used tracers for OR molecular imaging, e.g., the nonselective OR antagonist ^11^C-diprenorphine ([^11^C]DPN, **301a**, (K_i_ (μ) = 0.07 nM, K_i_ (δ) = 0.23 nM, K_i_ (κ) = 0.02 nM, Figure 75) [333,334] and the partial μOR agonist and kappa antagonist ^11^C-buprenorphine ([^11^C]BPN, **301b**) [334,335] (K_i_ (μ) = 1.5 nM, K_i_ (δ) = 6.1 nM, K_i_ (κ) = 2.5 nM, K_i_ (NOP) = 77.4).

Following the development of the agonist ligand ^11^C-phenethyl-orvinol (**302a**, [^11^C]PEO) (K_i_ (μ) = 0.18 nM, K_i_ (δ) = 5.1 nM, K_i_ (κ) = 0.12 nM) [336], Schoultz et al. reported its ^18^F-fluorine-labeled version, ^18^F-phenethyl-orvinol (**302b**, [^18^F]FE-PEO, (K_i_ (μ) = 0.10 nM, K_i_ (δ) = 0.49 nM, K_i_ (κ) = 0.08 nM) [337,338,339]. In 2000, Wester et al. reported the radiosynthesis of the antagonist ligand 6-*O*-(2-[^18^F]fluoroethyl)-6-*O*-desmethyl-diprenorphine ([^18^F]FE-DPN, **301c**, K_i_ (μ) = 0.24 nM, K_i_ (δ) = 8.00 nM, K_i_ (κ) = 0.20 nM, Figure 75) [340] thorough an indirect [^18^F]fluoroethylation procedure. Subsequently, Schoultz et al. [341] elaborated a more efficient method for the synthesis of [^18^F]FE-DPN (**301c**) and [^18^F]FEOTos, and they also extended their method to the radiosynthesis of 6-*O*-(2-[^18^F]-fluoroethyl)-6-*O*-desmethyl-buprenorphine (**301d**, [^18^F]FE-BPN, K_i_ (μ) = 0.24 nM, K_i_ (δ) = 2.10 nM, K_i_ (κ) = 0.12 nM) [341]. More recently, Marton et al. [342] developed a novel precursor molecule, 6-*O*-(2-tosyloxyethyl)-6-*O*-desmethyl-3-*O*-trityl-diprenorphine (TE-TDDPN, «Henriksen precursor»), for the one-pot, two-step nucleophilic radiosynthesis of [^18^F]FE-DPN ([^18^F]**301c**). In 2023, the research group of Mikecz [343] optimized the radiosynthesis of [^18^F]FE-DPN ([^18^F]**301c**) from the new precursor. Despite its incomplete OR sub-type selectivity in vitro, 6-*O*-(2-fluoroethyl)-6-*O*-desmethyl-diprenorphine (FE-DPN, **301c**) preferentially binds to μ-ORs in living brains [344].

### 2.13. Other Semisynthetic Derivatives

In the main sections of this review, we presented selected research topics from a huge body of work entailing more than five decades of scientific activity of the alkaloid research groups of the University of Debrecen and the Alkaloida Chemical Company. In this final section, we seek to highlight some additional important studies. The investigation of Horváth and Makleit [345,346] is noteworthy due to its importance in the field of the morphine total synthesis. The authors developed an efficient method for the conversion of 6-*O*-demethyl-salutaridine to salutaridine. In the early 1990s, the Makleit research group turned with great interest toward the Bentley compounds [221,280]. The main driving force was the finding of Lewis et al. [347] that buprenorphine, a 6,14-ethanomorphinan partial μOR agonist and κOR antagonist, serves as an alternative to methadone for the treatment of heroin/diamorphine dependence. Makleit et al. developed numerous novel synthesis routes for the efficient preparation of buprenorphine, diprenorphine and their *N*^17^-substituted analogs [276,348,349,350]. Very recently, we provided an overview of their research in the field of C-ring bridged morphinans together with the Diels–Alder adducts of morphinan-6,8-dienes and their transformations dating back almost a century [57]. We cannot omit mentioning the investigations of Berényi, who established aporphine chemistry as a new direction for the Debrecen alkaloid research group. They expended enormous effort over the years in the synthesis of morphinan-6,8-dienes [57] and their rearrangements to potent dopaminergic aporphine derivatives [325,351]. We emphasize the efforts extending over many years of Hosztafi in methodological research for the *N*-demethylation of morphine alkaloids [352,353,354,355]. We also note the thorough investigations carried out by Seller et al. [356] for the refinement of poppy alkaloid extraction technology with the help of modern technical equipment and separation methods.

## 3. Summary and Conclusions

In considering the cause-and-effect relationships in the current opioid crisis [357,358,359,360], it is important to appreciate the enormous efforts in medicinal chemistry driven by the dream of generations of scientists of finding the ideal analgesic with a reduced risk of dependence and overdose.

Here, the concept of biased pharmacology for several OR sub-types presents a model for obtaining therapeutic responses, with a lesser risk of dependence and side effects [51,52]. Comprehending the broad scope of this concerted search calls for consideration of the historical background of these investigations and its implications for the new generation of researchers and scholars.

In the present survey, we provide an overview of the organic chemical investigations of morphine alkaloids arising from a historical collaboration between Hungarian research groups in industry (Alkaloida Chemical Company) and academia (University of Debrecen), dating back to the 1950s. We emphasize the historical continuity of this endeavor, building continuously upon the pioneering work by Kabay, now recognized as having been one of the most important morphine alkaloid producers in the world [14]. The privatization of the Alkaloida Chemical Company following the political changes in Hungary in the 1990s brought to an abrupt end the five-decade-long close and fruitful collaboration between the Department of Organic Chemistry at the University of Debrecen and the Alkaloida Chemical Company in Tiszavasvári. Our review intends to enshrine and memorialize this exemplary history of industrial/academic collaboration. For list of abbreviations used in this survey and the members of the «University of Debrecen—Alkaloida» research team mentioned by name [361], see Appendix A.

## Figures and Tables

**Figure 1 ijms-26-02736-f001:**
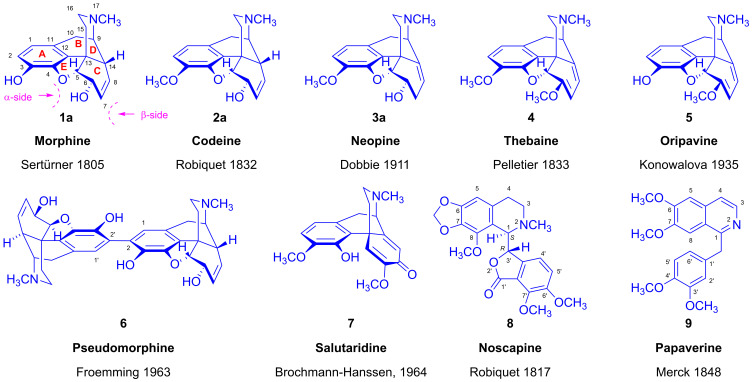
Chemical structures of selected poppy alkaloids, with scientist’s name and year of first isolation.

**Figure 2 ijms-26-02736-f002:**
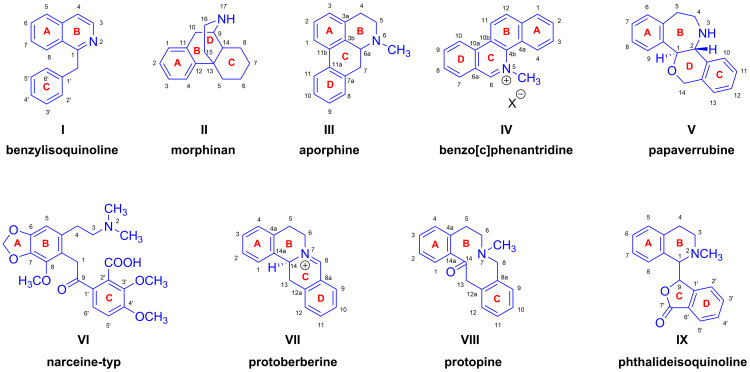
Ground scaffolds of poppy alkaloids.

**Figure 3 ijms-26-02736-f003:**
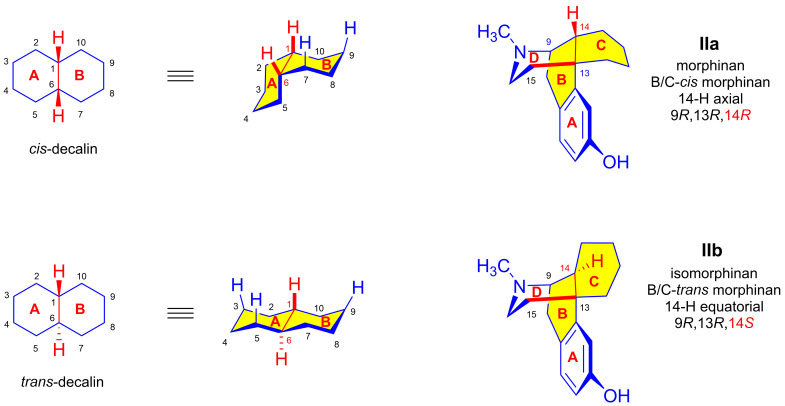
Structures of B/C-*cis* and B/C-*trans* morphinans.

**Figure 4 ijms-26-02736-f004:**
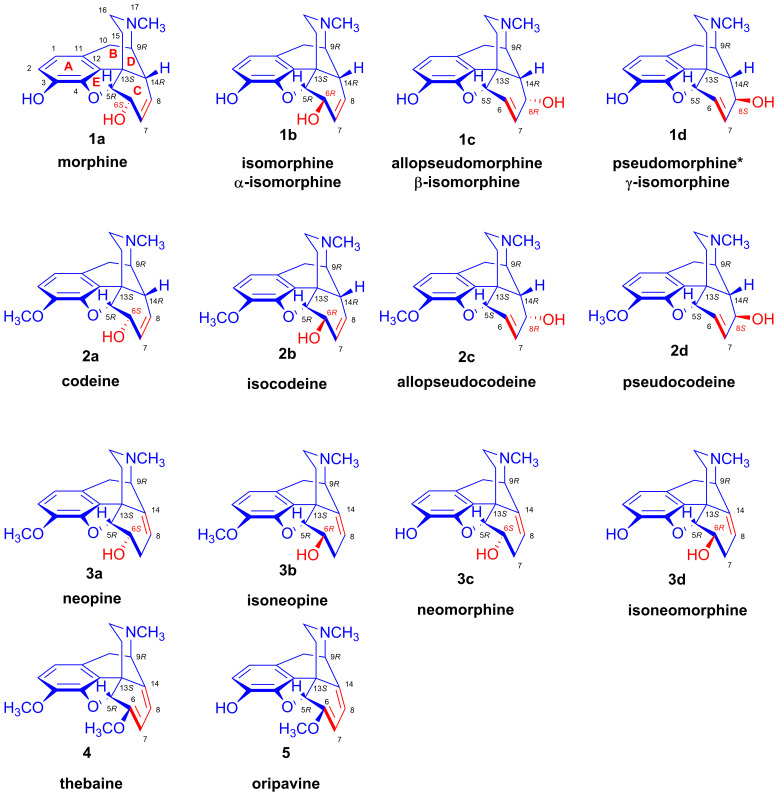
Chemical structures of selected morphinan alkaloids and their synthetic derivatives. * Makleit–Bognár nomenclature.

**Figure 5 ijms-26-02736-f005:**
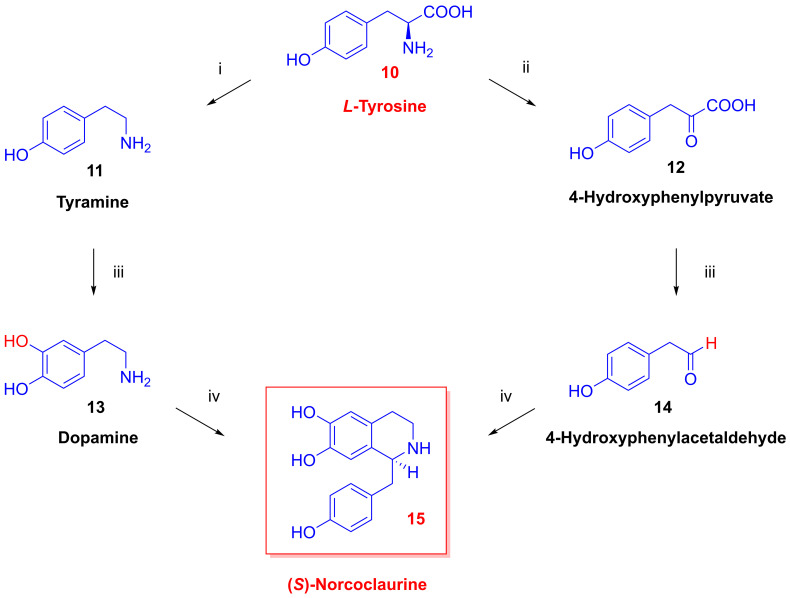
Biosynthesis of (*S*)-norcoclaurine. *Reactions catalyzed by enzymes*: (i): *L*-tyrosine decarboxylase (TyrDC); (ii): *L*-tyrosine transaminase (TyrAT); (iii): phenolase; (iv): 4-hydroxy-phenylpyruvate decarboxylase (4HPPDC).

**Figure 6 ijms-26-02736-f006:**
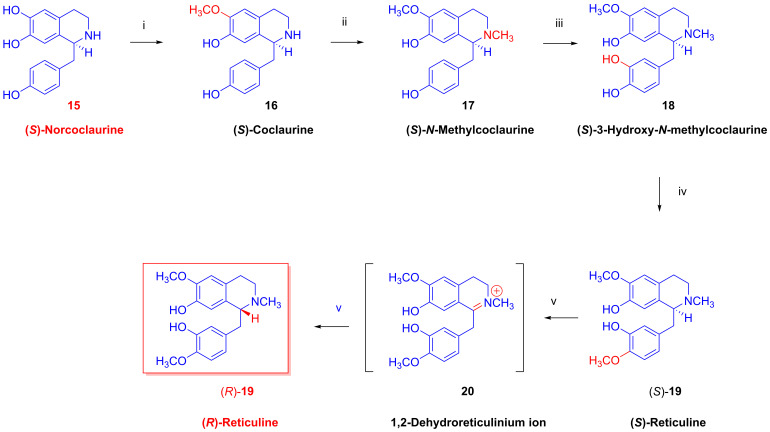
Biosynthesis of (*R*)-reticuline. *Reagents and conditions*: (i): norcoclaurine-6-*O*-methyltransferase; (ii): tetrahydrobenzylisoquinoline-*N*-methyltransferase; (iii): phenolase; (iv): (*S*)-3′-hydroxy-*N*-methylcoclaurine-4′-*O*-methyltransferase; (v): 1,2-dehydroreticuline synthase, NADPH-dependent reductase.

**Figure 7 ijms-26-02736-f007:**
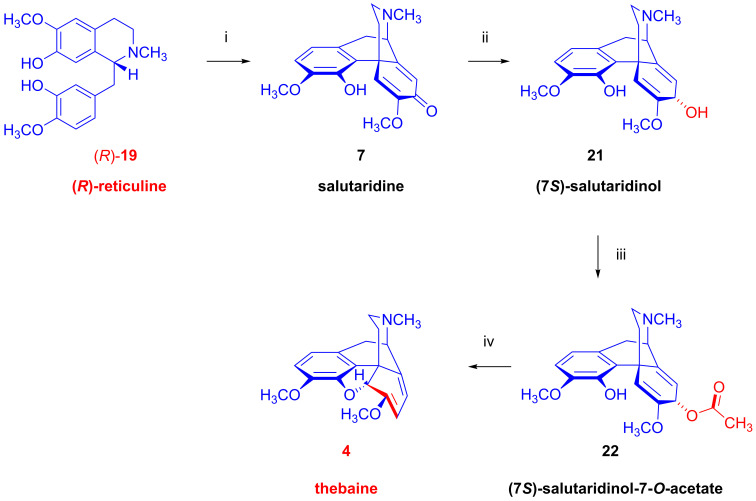
Biosynthesis of thebaine from (*R*)-reticuline. *Reagents and conditions*: (i): salutaridine synthase (NADPH, O_2_); (ii): NADPH-dependent salutaridine oxydoreductase (NADPH, NADP^⊕^); (iii): salutaridinol-7-*O*-acetyltransferase (AcCoA, CoA); (iv): spontaneous (pH 8–9, -AcOH).

**Figure 8 ijms-26-02736-f008:**
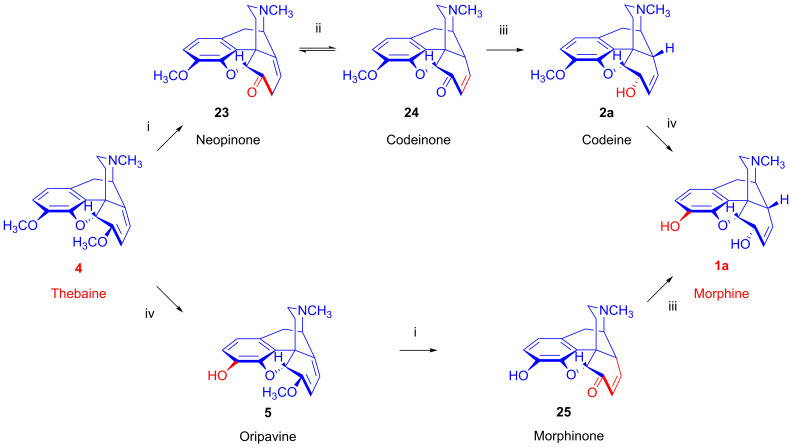
Biosynthesis of morphine from thebaine. *Reagents and conditions*: (i): enolether hydrolysis; (ii): chemical equilibrium; (iii): codeinone reductase; (iv): 3-*O*-demethylation.

**Figure 9 ijms-26-02736-f009:**
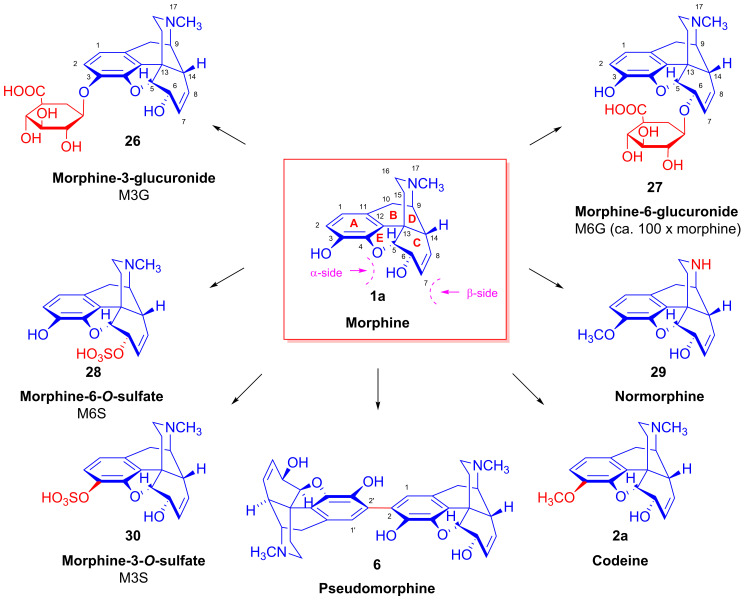
Structures of selected morphine metabolites.

**Figure 10 ijms-26-02736-f010:**
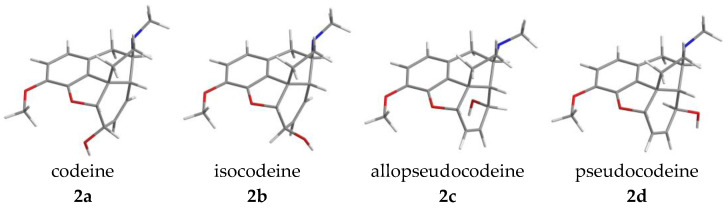
Three-dimensional structures of codeine isomers.

**Figure 11 ijms-26-02736-f011:**
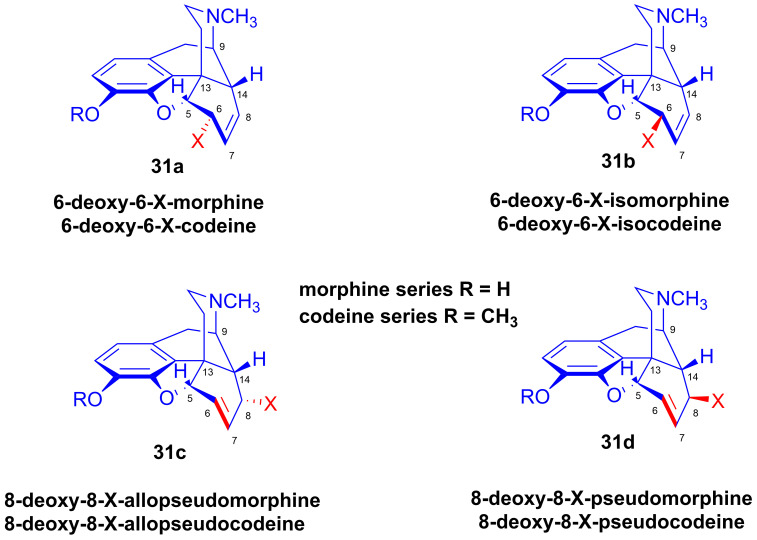
Examples for the application of the Makleit–Bognár nomenclature.

**Figure 12 ijms-26-02736-f012:**
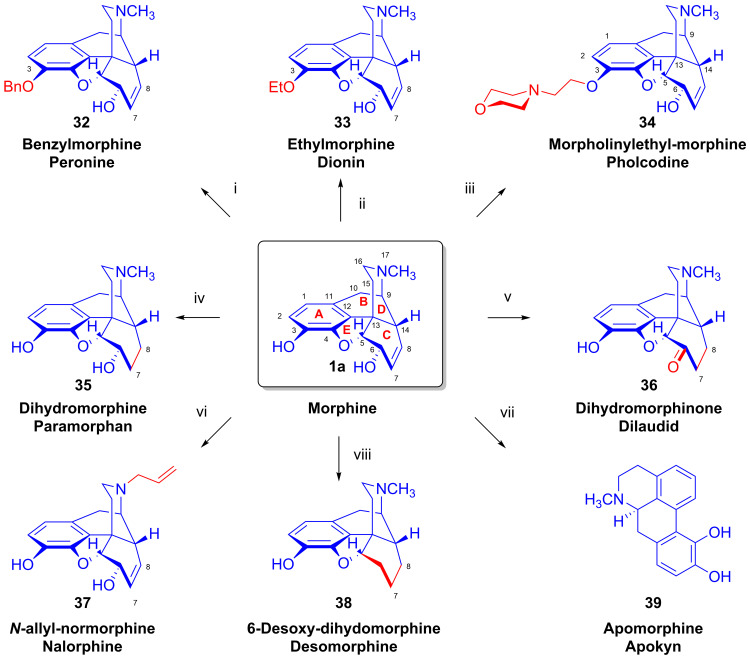
Industrial-scale syntheses of some morphine derivatives. *Reagents and conditions*: (i): benzyl bromide, NaOEt, EtOH, heating; (ii): ethyl bromide, NaOEt, EtOH, 100 °C, 2 h; (iii): morpholinylethylchloride hydrochloride, NaOEt, ethanol, reflux, 1 h, 72%; (iv): H_2_, Pd-C, 1 N HCl or diluted AcOH; (v): catalytic rearrangement; (vi): (1) BrCN, CHCl_3_, reflux; (2) A: KOH, H_2_O, EtOH, 50 °C, B: diluted HCl, heating; (3) allyl bromide, NaHCO_3_, EtOH, reflux, 10 h; (vii): cc. HCl, sealed tube; (viii) see details in Section 2.6.

**Figure 13 ijms-26-02736-f013:**
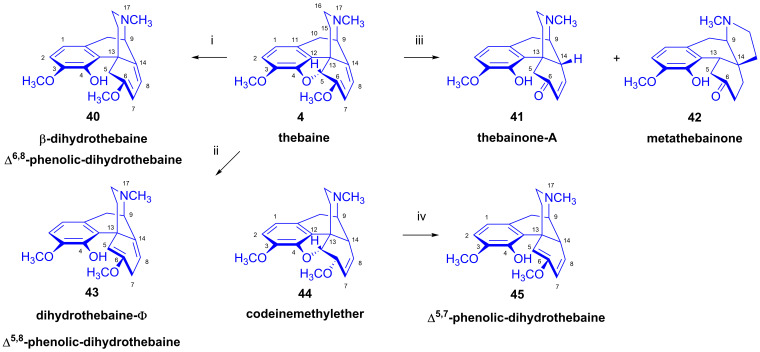
Reduction of thebaine with chemical reducing agents. *Reagents and conditions*: (i): LiAlH_4_, benzene, diethyl ether or THF, reflux, 8 h, 22–27%; (ii): Na, NH_3_ (liq.), 90%; (iii): 5 equiv. SnCl_2_, cc HCl, sealed tube, 100 °C, 20 min, 53% (**42**); (iv): NaOEt, EtOH.

**Figure 14 ijms-26-02736-f014:**
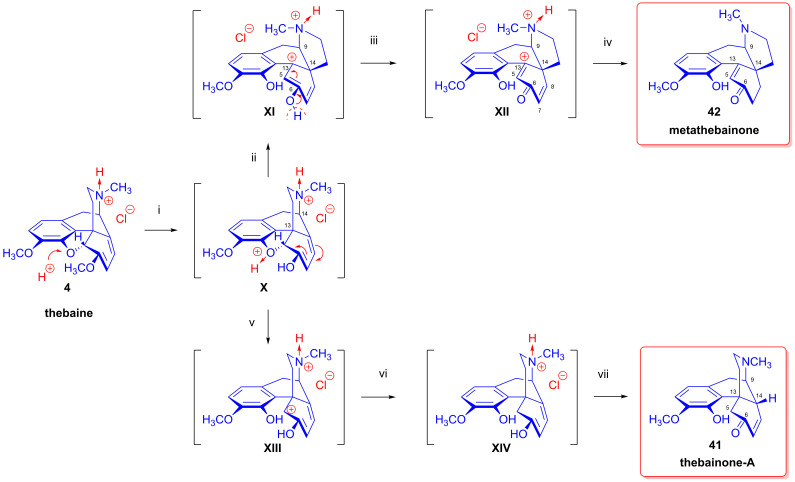
Mechanism of the formation of metathebainone and thebainone-A. *Reagent and elementary step of the reaction*: (i): **A**: + HOH, **B**:−CH_3_OH, **C**: + H^⊕^; (ii): rearrangement; (iii): −H^⊕^; (iv): H_2_/catalyst; (v): heterolysis of C^5^-O bond; (vi): + 2e^⊖^, + H^⊕^; (vii): ketonization with 1,4-mechanism.

**Figure 15 ijms-26-02736-f015:**
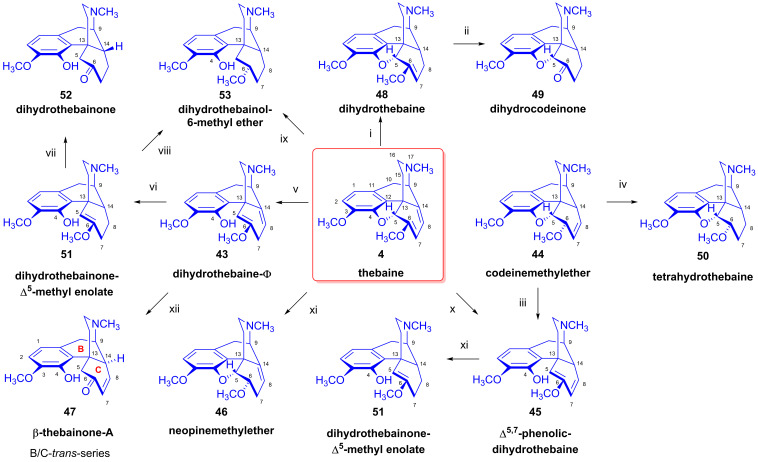
Products of the reduction of thebaine [84,85]. *Reagents and conditions*: (i): from **4** hydrochloride salt, H_2_, atmospheric pressure, 0.08 M hydrochloric acid (2.8-fold), 10% Pd-C (0.1-fold), 62%; (ii): 5 equiv. 5 M hydrochloric acid, RT, 24 h, quant.; (iii): NaOEt, EtOH; (iv): H_2_, 10% Pd-C (4 m/m %), 1 M hydrochloric acid, quant.; (v); Na, NH_3_ (liq.), 90% [73,74,80,81,82]; (vi): H_2_, PtO_2_, NaHCO_3_, EtOH, 45% or H_2_, Pd-SrCO_3_, EtOH, 50%; (vii): **A**: AcOH or 1 M HCl, RT, **B**: H_2_, 10% Pd-C; (viii): 10% Pd-C, EtOH, or directly from **4** base by catalytic reduction in neutral solution (5% Pd-BaSO_4_, EtOH), or alternatively by catalytic reduction (H_2_, PtO_2_) of β-dihydrothebaine (**40**) [79]; (ix): H_2_, 5% Pd-BaSO_4_, EtOH [67] or H_2_, Pt black, NaHCO_3_, EtOH; (x): H_2_, 1,6-hydrogen addition [C^5^-O; C^14^] on the allyl ether group combined with the 6,8-diene system [86]; alternatively from codeine (**2a**), codeine → codeine-*N*-oxide → codeine-*N*-oxide-6-*O*-methylether → codeine-6-methylether → phenolic Δ^5,7^-dihydrothebaine (**45**) [73,74,85]; (xi): H_2_, 10% Pd-C, NaHCO_3_, EtOH, RT, 7 h [85]; by application of H_2_, 10% Pd-C, 1 M hydrochloric acid, thebainone-A and dihydrothebainone (**52**) formed [85]; (xii): warm aqueous KHSO_4_ solution [74].

**Figure 16 ijms-26-02736-f016:**
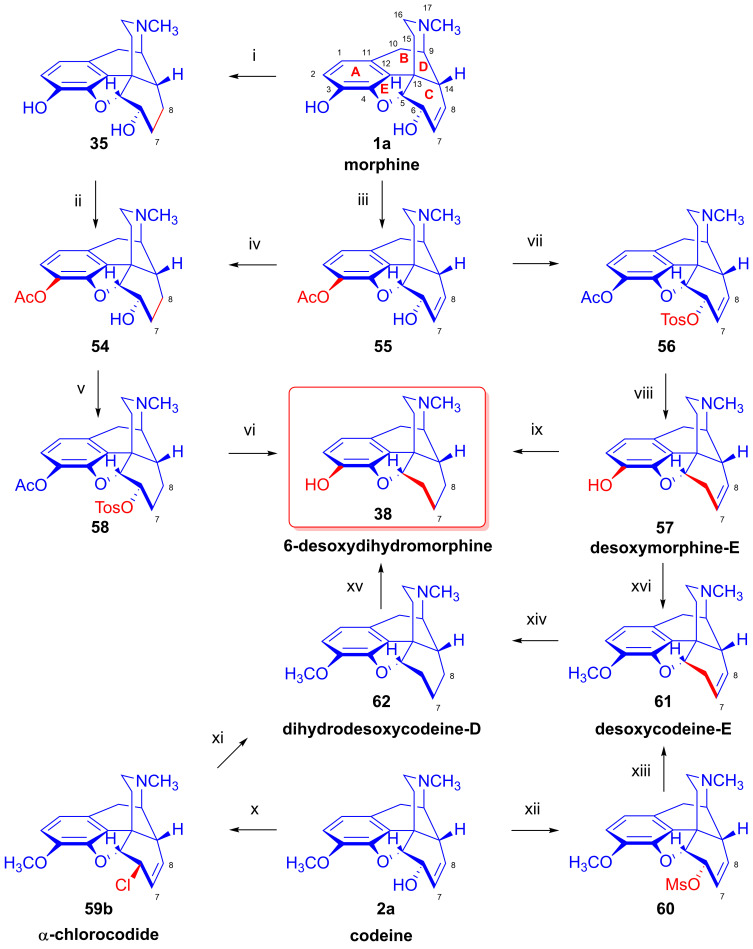
Synthesis of 6-desoxydihydromorphine. *Reagents and conditions*: (i): H_2_, Pd-C, 1 N HCl or diluted AcOH; (ii): acetic anhydride, aqueous NaHCO_3_ solution, 63%; (iii): H_2_, 10% Pd-C, aqueous HCl (pH 6–6.5), 96%; (iv): acetic anhydride, aqueous NaHCO_3_ solution, 92%; (v): TosCl, pyridine; **A**: 0 °C, 2 h, **B**: RT, 16 h, 51%; (vi): LiAlH_4_, THF; **A**: reflux, 3 h, **B**: RT, 16 h, 25%; (vii): TosCl, pyridine; **A**: 0 °C, 2 h, **B**: RT, overnight; (viii): LiAlH_4_, THF, reflux, 3 h, nitrogen stream; (ix): H_2_, PtO_2_, MeOH, 1 h; (x): SOCl_2_; reflux, 1.5 h; (xi): H_2_, Pd-BaSO_4_, MeOH or 1 N HCl, 30–80%; (xii): MsCl, Et_3_N, CH_2_Cl_2_, 0 °C, 1 h; (xiii); LiAlH_4_, THF, RT, 1 h; (xiv): H_2_, PtO_2_, MeOH, 4 Bar, 1 h; (xv): BBr_3_, CH_2_Cl_2_, RT, 30 min; (xvi): CH_2_N_2_, Et_2_O, 48 h, 0–4 °C.

**Figure 17 ijms-26-02736-f017:**
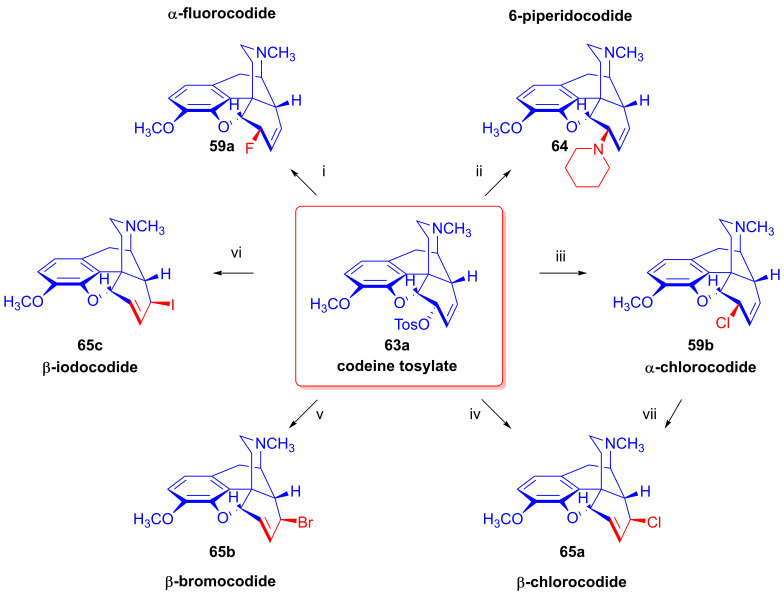
Reactions of codeine tosylate with nucleophiles. *Reagents and conditions*: (i): Bu_4_NF, acetonitrile, reflux, 4 h, 48%; (ii): piperidine, benzene, reflux, 36 h, 87%; (iii): A. LiCl, acetone, reflux, 4 h, quant. or B. LiCl, DMF, 40 °C, 5 h, 76%; (iv): LiCl, DMF, 120 °C, 5 h, 33%; (v): LiBr, acetone, reflux, 2.5 h, 98%; (vi): NaI, acetone, reflux, 2.5 h, 43%; (vii): DMF, 120 °C, 10 h, 41%.

**Figure 18 ijms-26-02736-f018:**
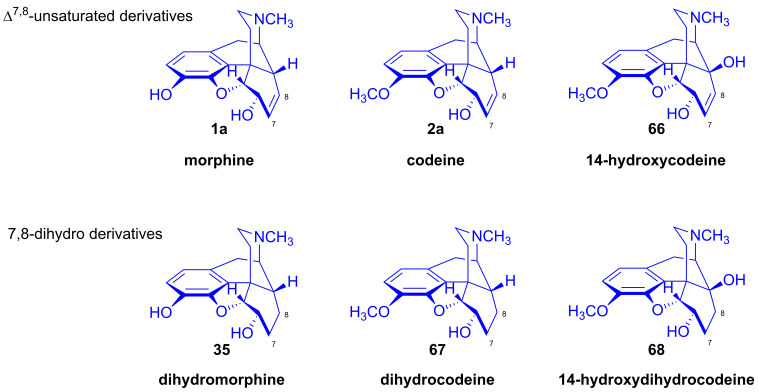
Structures of derivatives belonging to the “*morphine series*” defined by Makleit and Bognár.

**Figure 19 ijms-26-02736-f019:**
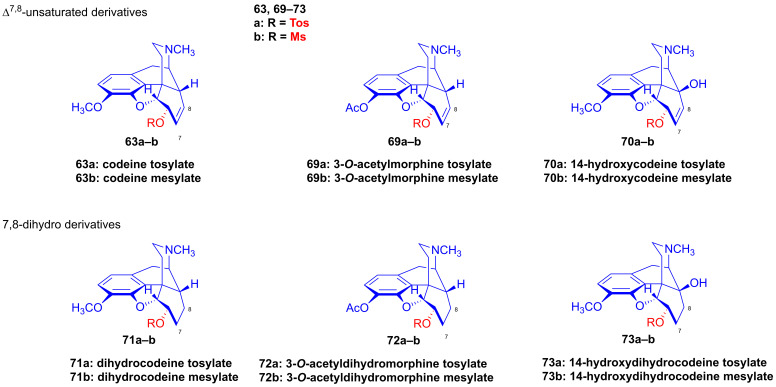
Sulfonate esters of selected morphine derivatives.

**Figure 20 ijms-26-02736-f020:**
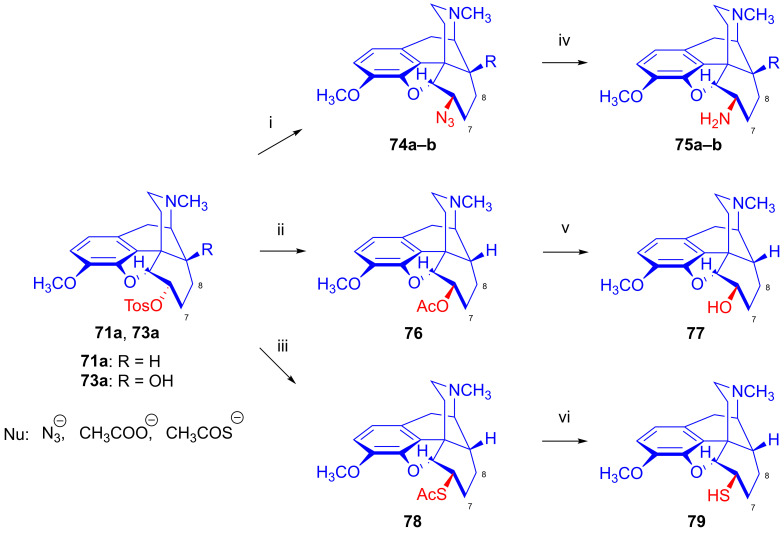
Reaction of the sulfonate esters of 7,8-dihydro derivatives with nucleophiles. *Reagents and conditions*: (i): NaN_3_, DMF, H_2_O, 100 °C, 24 h, R = H, 83%, R = OH, 67%; (ii): from **71b**, AcOH, Ac_2_O, NaOAc, reflux, 31 h; (iii): KSCOCH_3_, DMF, 100 °C, 24 h; under N_2_ atmosphere; (iv): LiAlH_4_, Et_2_O, reflux, 3 h, 60% or H_2_, 10% Pd-C, MeOH; (v): alkaline hydrolysis, (ii + iv) 21%; (vi): NaOMe, MeOH, room temperature, 24 h, 91%.

**Figure 21 ijms-26-02736-f021:**
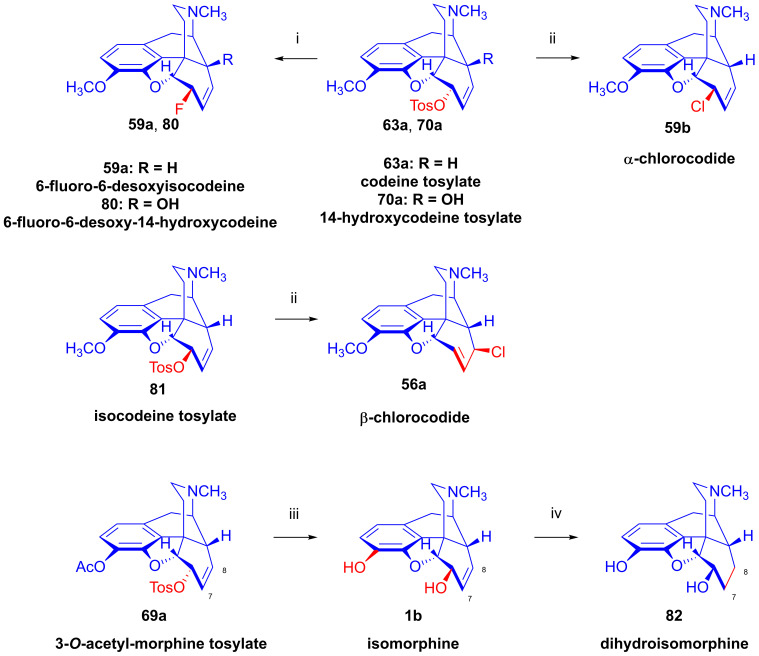
Reaction of selected 6-*O*-tosylates of Δ^7,8^-unsaturated morphinans. *Reagents and conditions*: (i): 1.6 equiv. Bu_4_NF, CH_3_CN, reflux, 4 h, 48% [111,112]; (ii): LiCl, acetone, reflux, 4 h; (iii): 10% AcOH (aq.), reflux, 4 h; (iv): H_2_, BaSO_4_/Pd-C, EtOH, 90%.

**Figure 22 ijms-26-02736-f022:**
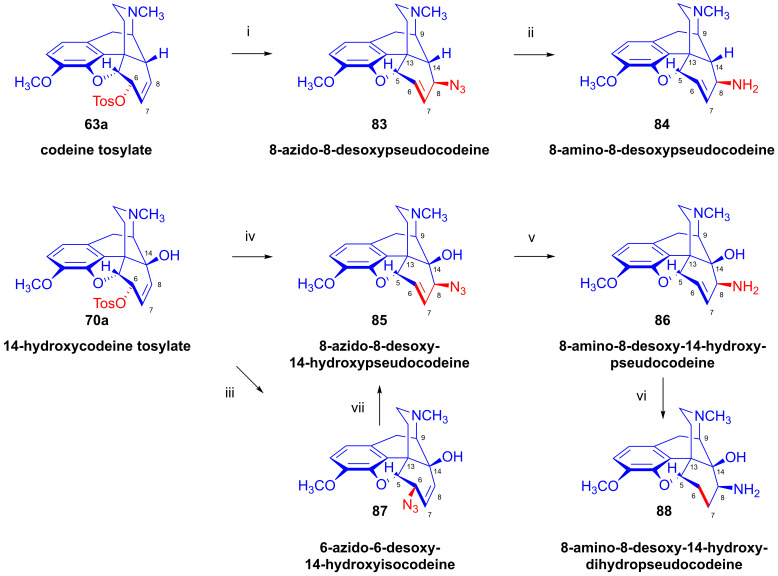
Synthesis of 8-substituted-8-desoxypseudocodeine derivatives. *Reagents and conditions*: (i): 10 equiv. NaN_3_, DMF, H_2_O, 100 °C, 4 h, 65%; (ii): LiAlH_4_, Et_2_O, reflux, 3 h, 46%; (iii): 1.25 equiv. NaN_3_, DMF, H_2_O, 100 °C, 4 h, 38%; (iv): 1.25 equiv. NaN_3_, DMF, H_2_O, 100 °C, 8 h, 46%; (v): LiAlH_4_, Et_2_O, reflux, 3 h, 25%; (vi): H_2_, Pd-C, MeOH, 75%; (vii): DMF, 100 °C, 6 h, 83%.

**Figure 23 ijms-26-02736-f023:**
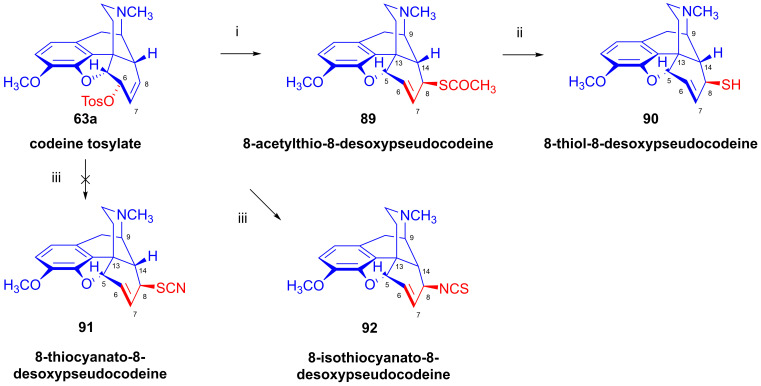
Reaction of codeine tosylate (**63a**) with CH_3_COS^⊖^ and ^⊖^SCN nucleophiles. *Reagents and conditions*: (i): 2 equiv. KSCOCH_3_, DMF, 100 °C, 5 h, N_2_ atmosphere, 57%; (ii): NaOMe, MeOH, room temperature, N_2_ atmosphere, dark, 24 h, 38%; (iii): 2 equiv. KSCN, acetone, reflux, 2.5 h, 48%.

**Figure 24 ijms-26-02736-f024:**
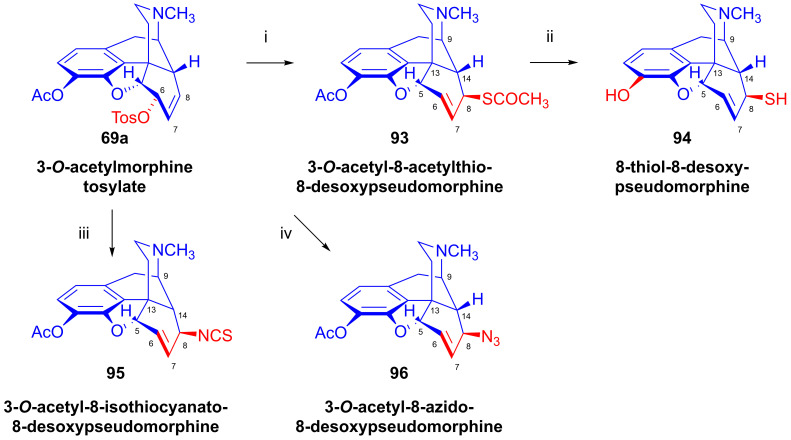
Reaction of 3-*O*-acetylmorphine tosylate with nucleophiles (following the Makleit–Bognár nomenclature [54,55]; γ-isomorphine derivatives). *Reagents and conditions*: (i): 2 equiv. KSCOCH_3_, DMF, 100 °C, 5 h, N_2_ atmosphere, 43%; (ii): NaOMe, MeOH, room temperature, N_2_ atmosphere, dark, 24 h, 42%; (iii): 2 equiv. KSCN, reflux, 2.5 h, 90%; (iv): NaN_3_, DMF, H_2_O, 100 °C, 4 h (**96** → **128**).

**Figure 25 ijms-26-02736-f025:**
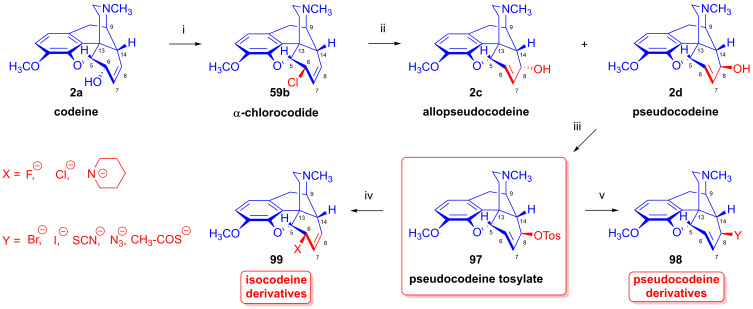
Reactions of pseudocodeine tosylate with various nucleophiles. *Reagents and conditions*: (i): PCl_5_, chloroform, room temperature, 2 h, 90–95%; (ii): water, acetic acid, reflux, 3 h, **2c**: 15%, **2d**: 38%; (iii): TosCl, pyridine, 0–5 °C, 2 h, and then room temperature, 24 h, 76%; (iv): **X** = **A**. Bu_4_NF, acetonitrile reflux, 19 h, 10%; **B**. LiCl, acetone, reflux, 25 h, 16%; **C**. piperidine, benzene, reflux, 127 h, 65%; (v): **Y** = **A**: LiBr, acetone, reflux, 19 h, 52%; **B**. NaI, acetone, reflux, 19 h, 26%; **C**. KSCN, acetone, reflux, 16.6 h, 15%; **D**. NaN_3_, DMF, 100 °C, 8 h, 30%; **E**. KSCOCH_3_, DMF, 100 °C, 8 h, 10%.

**Figure 26 ijms-26-02736-f026:**
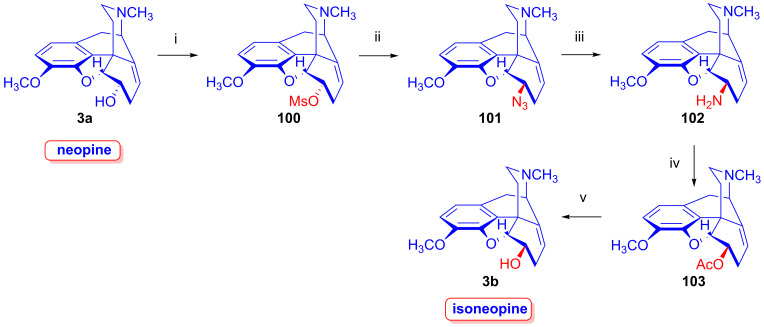
Synthesis of isoneopine. *Reagents and conditions*: (i): MsCl, pyridine, room temperature, 24 h; (ii): NaN_3_, DMF, H_2_O, 75%; (iii): Zn, NaI, DMF, reflux, 6 h, 65%; (iv): NaNO_2_, AcOH, 20 °C, 24 h, 90%; (v): KOH, EtOH, reflux, 10 min, quant.

**Figure 27 ijms-26-02736-f027:**
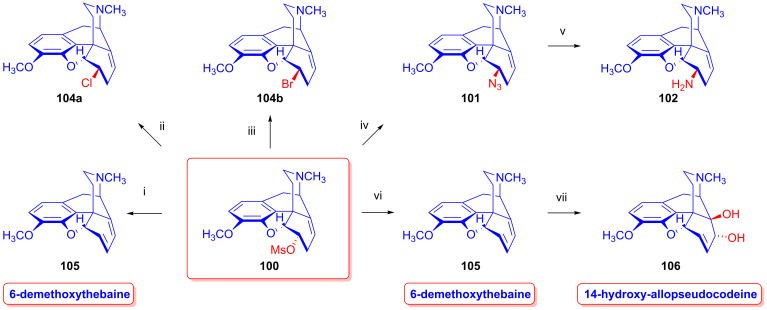
Reactions of 6-*O*-mesylneopine with nucleophiles. *Reagents and conditions*: (i): 5 equiv. Bu_4_NF, CH_3_CN, reflux, 2.5 h, 77%; (ii): 10 equiv. LiCl, DMF, 120 °C, 6 h, 38%; (iii): 10 equiv. LiBr, DMF, 120 °C, 11 h, 32%; (iv): 10 equiv. NaN_3_, DMF, 100 °C, 24 h, 55%; (v): LiAlH_4_, Et_2_O, reflux, 1 h, 32%; (vi): 10 equiv. NaI, DMF, 120 °C, 30 h, 21% (**105**), and 36% recovered neopine mesylate (**100**); (vii): 85% formic acid, 30% H_2_O_2_, 40 °C, 2 h, 71%.

**Figure 28 ijms-26-02736-f028:**
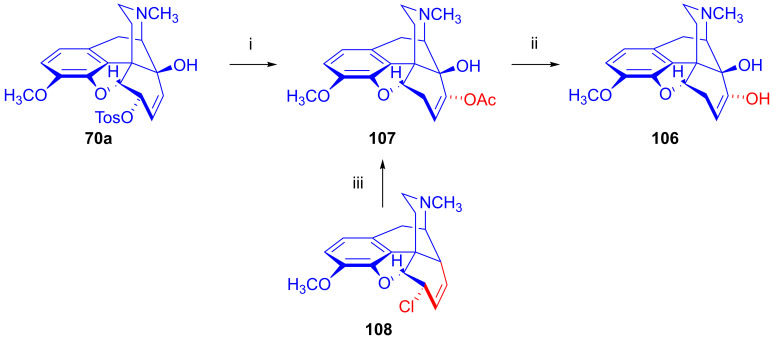
Synthesis of 14-hydroxy-allopseudocodeine. *Reagents and conditions*: (i): AcOH-H_2_O 7:1 (*v*/*v*), reflux, 4 h, 39% [150]; (ii): KOH, MeOH-H_2_O 3:1 (*v*/*v*), reflux, 1 h; (iii): **A**. 10% AcOH, reflux, 4 h, 45% [151], **B**. KOH, MeOH-H_2_O 4:1 (*v*/*v*), 40–50 °C, 1 h.

**Figure 29 ijms-26-02736-f029:**
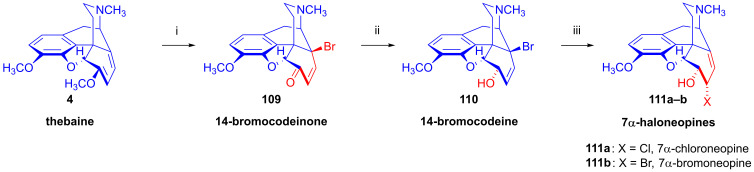
Synthesis of 7α-haloneopines from thebaine. *Reagents and conditions*: (i): *N*-bromosuccinimide, acetone–water 2:1 (*v*/*v*), 71%; (ii): NaBH_4_, benzene, MeOH, 5–10 °C, 1 h, 94%; (iii): cc. HCl, 5–10 °C, overnight, 90%, or 48% HBr, 0–5 °C, overnight.

**Figure 30 ijms-26-02736-f030:**
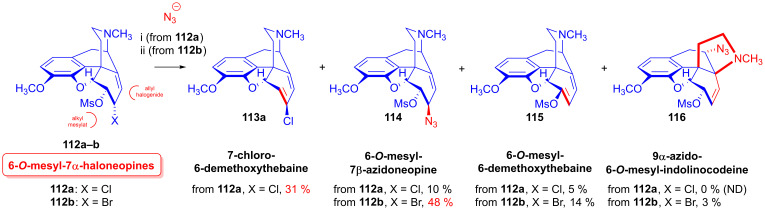
Azidolysis of 6-*O*-mesyl-7α-haloneopines. *Reagents and conditions*: (i): 5 equiv. NaN_3_, DMF, H_2_O, 100 °C, 8 h; (ii) 5 equiv. NaN_3_, DMF, H_2_O, 100 °C, 20 min.

**Figure 31 ijms-26-02736-f031:**
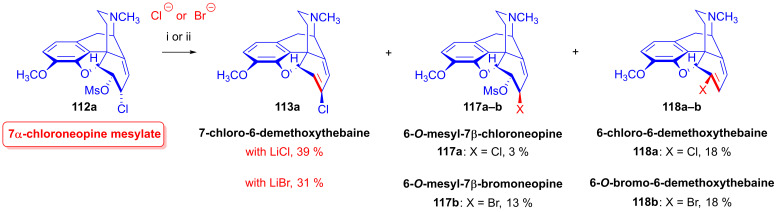
Reaction of 6-*O*-mesyl-7α-chloroneopine with Cl^⊖^ and Br^⊖^ nucleophiles. *Reagents and conditions*: (i): 5 equiv. LiCl, DMF, 100 °C, 24 h; (ii): 5 equiv. LiBr, DMF, 100 °C, 24 h.

**Figure 32 ijms-26-02736-f032:**
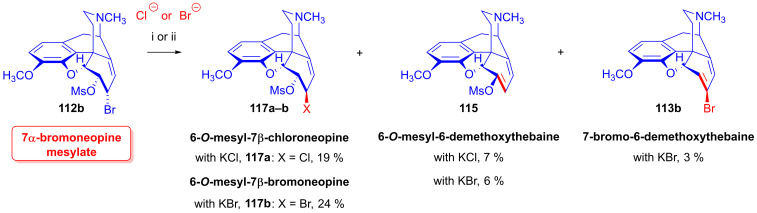
Reaction of 6-*O*-mesyl-7α-bromoneopine with nucleophiles. *Reagents and conditions*: (i): 5 equiv. LiCl, DMF, 100 °C, 30 min; (ii): 5 equiv. LiBr, DMF, 100 °C, 20 min.

**Figure 33 ijms-26-02736-f033:**
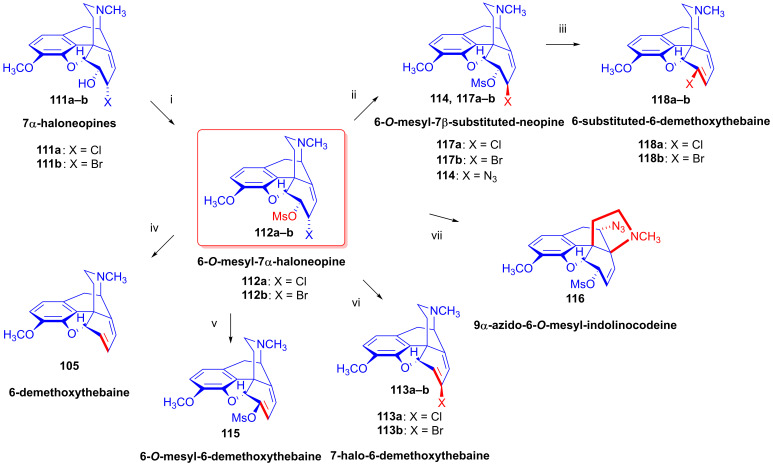
Nucleophilic substitution reactions of 7α-haloneopine mesylates. *Reagents and conditions*: (i): MsCl, Et_3_N, CH_2_Cl_2_, 15 min, from **111a**, X = Cl, 61%, from **111b**, X = Br, 51%; (ii): Nu^⊖^ = Cl^⊖^, Br^⊖^, N_3_^⊖^, S_N_2; (iii): S_N_i + -H; (iv): Nu^⊖^ = I^⊖^, S_N_2 + E_1,2_; (v): Nu^⊖^ = N_3_^⊖^, E_1,2_ (-HCl) or Nu = Cl^⊖^, Br^⊖^, N_3_^⊖^, E_1,2_ (- HBr); (vi): Nu = Cl^⊖^, Br^⊖^, I^⊖^, N_3_^⊖^, E_1,2_ (- CH_3_SO_2_OH); (vii): Nu^⊖^ = N_3_^⊖^.

**Figure 34 ijms-26-02736-f034:**
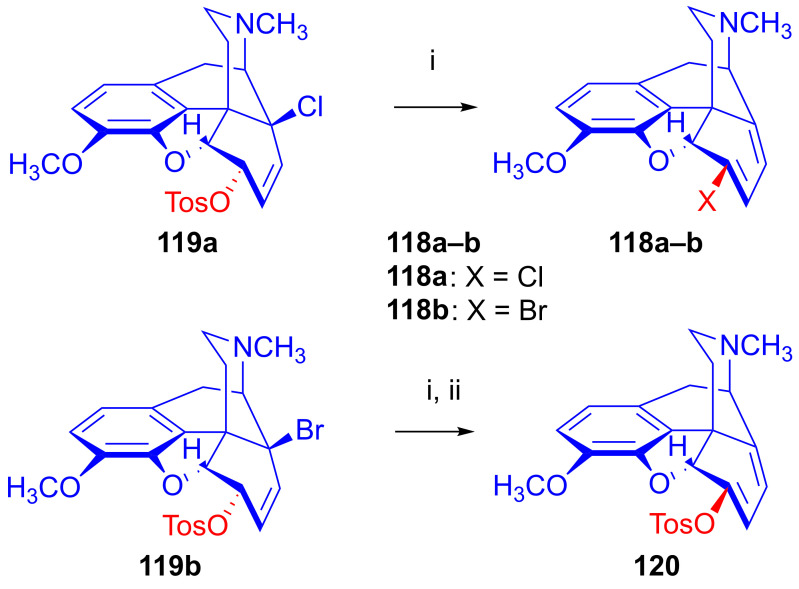
Reaction of 6-*O*-tosyl-14-halocodeines with nucleophiles. *Reagents and conditions*: (i): 6 equiv. LiCl, or 5 equiv. LiBr, DMF, 100 °C, 24 h; (ii): without any nucleophilic partner, DMF, 100 °C, 2 h.

**Figure 35 ijms-26-02736-f035:**
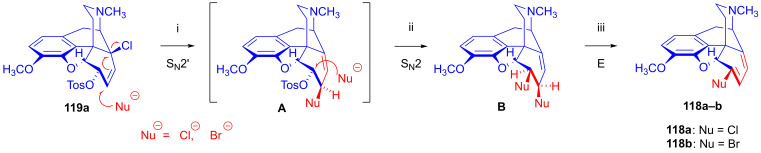
Supposed elementary steps of the formation of 6-halo-6-demethoxythebaines.

**Figure 36 ijms-26-02736-f036:**
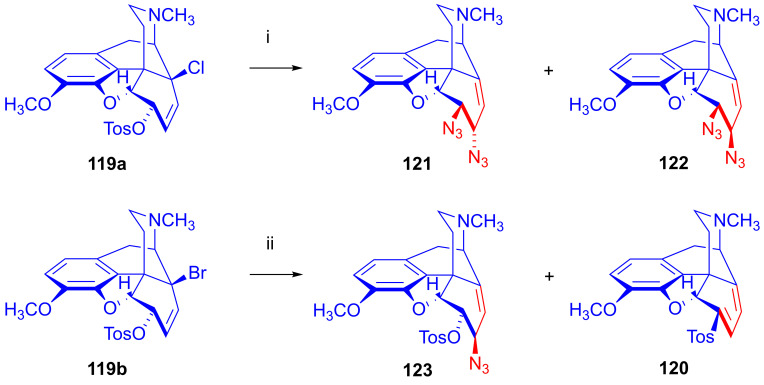
Azidolysis of 14-chlorocodeine tosylate and 14-bromocodeine tosylate. *Reagents and conditions*: (i): NaN_3_, DMF, H_2_O, 100 °C, 24 h; (ii): NaN_3_, DMF, 100 °C, 2 h.

**Figure 37 ijms-26-02736-f037:**
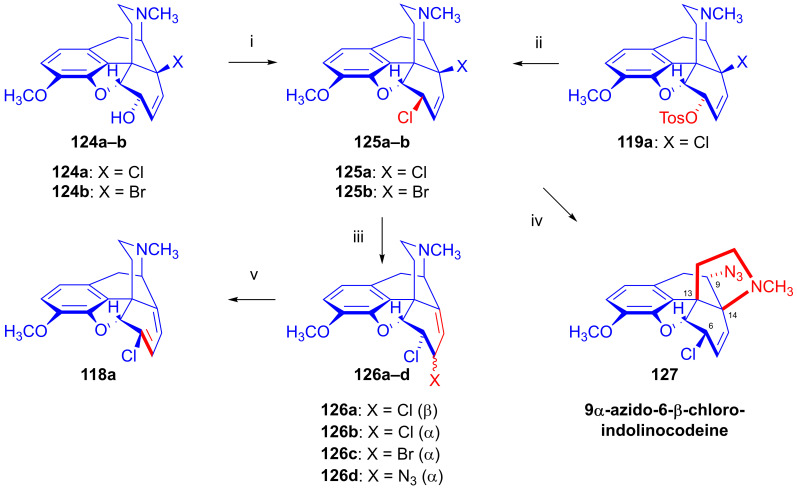
Nucleophilic substitution reactions of derivatives containing double allyl halide system. *Reagents and conditions*: (i): 2 equiv. PCl_5_, CHCl_3_, 0 °C, 2 h, **125a** (66%), **125b** (53%); (ii): 6 equiv. LiCl, DMF, 100 °C, 8 h; (iii): **A**: 10 equiv. LiCl, DMF, 100 °C, 30 min, **126a** (42%) and **126b** (4%); **B**: 5 equiv. LiBr, DMF, reflux, 4 h, **126c** (16%); **C**: 5 equiv. NaN_3_, DMF, H_2_O, 100 ° C, 1 h, **126d** (44%) and **127** (8%); (iv): 5 equiv. NaN_3_, DMF, H_2_O, 100 °C, 1 h, **126d** (48%) and **127** (7%); (v): from **126c**, standing at RT for a few days.

**Figure 38 ijms-26-02736-f038:**
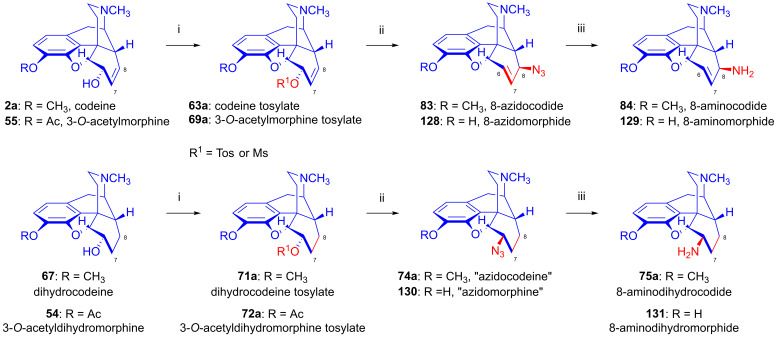
Synthesis of aminocodide and aminomorphide derivatives [54,55]. *Reagents and conditions*: (i): TosCl or MsCl, pyridine, A. 0 °C, 2 h, B. room temperature 24 h; (ii): 1.25 equiv. NaN_3_, DMF, H_2_O, 100 °C, 4 h (**83**,**128**) 24 h (**74a**,**130**); (iii): 3–8 equiv. LiAlH_4_, Et_2_O, reflux, 3 h.

**Figure 39 ijms-26-02736-f039:**
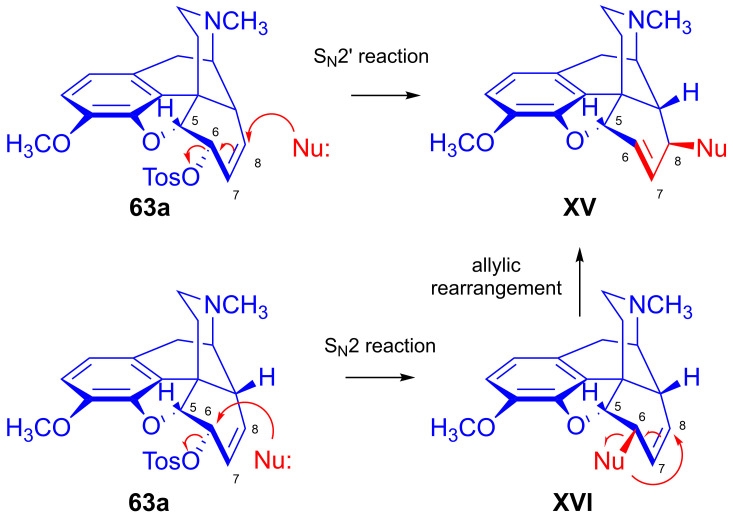
Supposed mechanism of the synthesis of 8-azidocodide. Nu = N_3_^⊖^.

**Figure 40 ijms-26-02736-f040:**
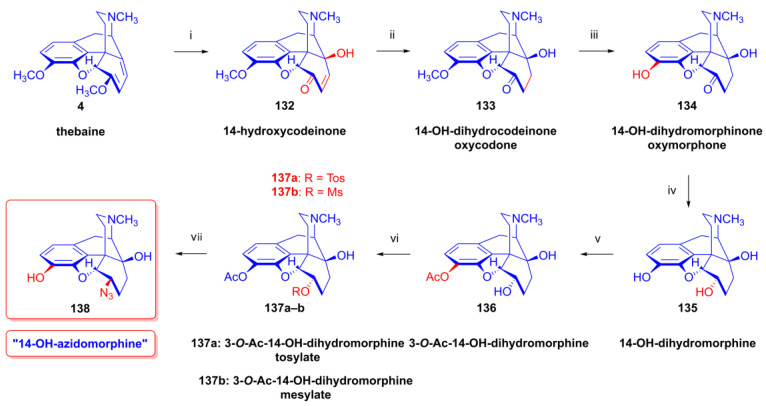
Synthesis of 14-hydroxyazidomorphine. *Reagents and conditions*: (i): formic acid, H_2_O_2_, 40 °C, 2 h; (ii): H_2_, 10% Pd-C, acetic acid; (iii): 48% hydrobromic acid, 120 °C, 20 min; (iv): NaBH_4_, EtOH, 20 °C, 20 h; (v): Ac_2_O, NaHCO_3_, H_2_O, 20 °C; (vi): MsCl or TosCl, pyridine, 20 °C, 24 h; (vii): NaN_3_, DMF, 100 °C, 24 h, 73%.

**Figure 41 ijms-26-02736-f041:**
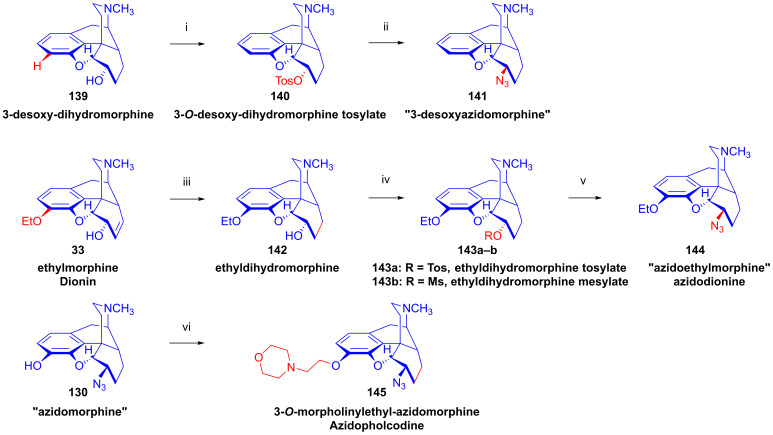
Synthesis of azidomorphine analogs. *Reagents and conditions*: (i): 1 equiv. TosCl, pyridine, room temperature, 24 h, 50%; (ii): 13 equiv. NaN_3_, DMF, H_2_O, 100 °C, 24 h, 50%; (iii): H_2_, Raney-Ni, MeOH or EtOH, RT, atmospheric pressure [200,201]; (iv): **A**. 1.2 equiv. TosCl, pyridine, 0 °C, 2 h, than RT for 24 h, 78%, or **B**. 1.1 equiv. MsCl, pyridine, 0 °C, 2 h, then room temperature for 24 h, 64%; (v): 10 equiv. NaN_3_, DMF, H_2_O, 100 °C, 24 h, 65%; (vi): 1.4 equiv. morpholinylethylchloride HCl, 2.3 equiv. Na, EtOH, reflux, 1 h, 40%.

**Figure 42 ijms-26-02736-f042:**
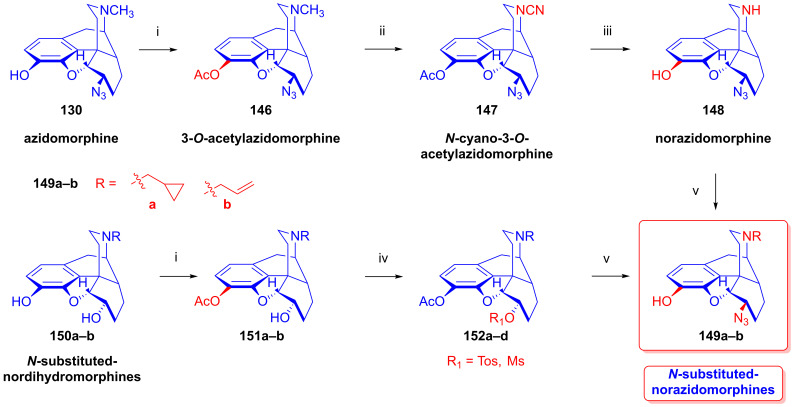
Synthesis of *N*^17^-substituted azidomorphine analogs. *Reagents and conditions*: (i): based on Welsh’s [99] method: Ac_2_O, NaHCO_3_, H_2_O; (ii): 2 equiv. BrCN, 60 °C, 2.5–6 h, 90%; (iii): 5–10% aqueous HCl reflux, 5–10 h; (iv): 1.2 equiv. TosCl or MsCl, 20 °C, 24 h; (v): 10 equiv. NaN_3_, DMF, 100 °C, 24 h.

**Figure 43 ijms-26-02736-f043:**
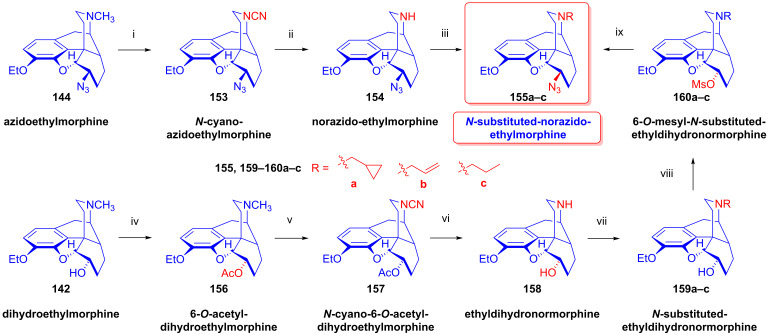
Synthesis of *N*-substituted-norazidoethylmorphine analogs. *Reagents and conditions*: (i): 2.1 equiv. BrCN, CHCl_3_, reflux, 4 h, 75%; (ii): 6% HCl, reflux, 6 h, 95%; (iii): **A**. cyclopropylmethyl bromide, NaHCO_3_, EtOH, reflux, 15 h, 46%; **B**. allyl bromide, NaHCO_3_, EtOH, reflux, 8 h, 54%; (iv): Ac_2_O, 100 °C, 1 h, 94%; (v): BrCN, CHCl_3_, reflux, 4 h, 95%; (vi): 10% HCl, reflux, 8 h, 94%; (vii): **A**. cyclopropylmethyl bromide, NaHCO_3_, EtOH, reflux, 15 h, 75%; **B**. allyl bromide, NaHCO_3_, EtOH, 80 °C, 20 h, 73%; **C**. *n*-propyl bromide, NaHCO_3_, 80 °C, 20 h, 58%; (viii): 1.2 equiv. MsCl, pyridine, room temperature, 24 h, 83% (R = CPM); (ix): 10 equiv. NaN_3_, DMF, H_2_O, 100 °C, 24 h, 62%, R = CPM.

**Figure 44 ijms-26-02736-f044:**
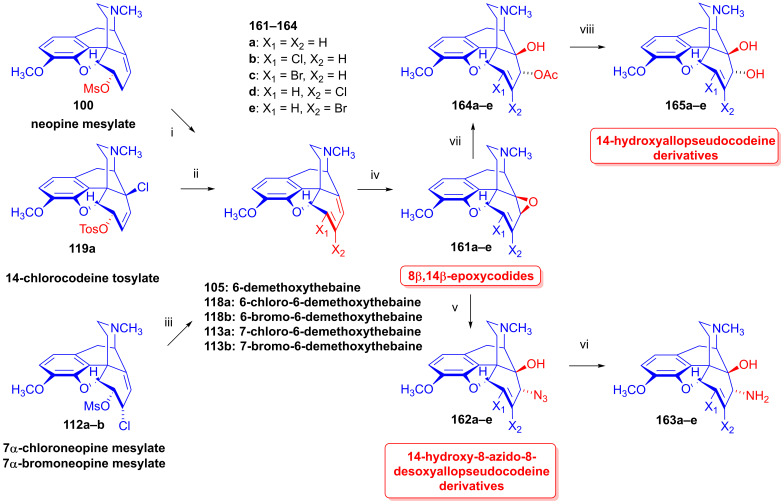
Synthesis of 14-hydroxy-8-azido-8-desoxyallopseudocodeine derivatives. *Reagents and conditions*: (i): 5 equiv. Bu_4_NF, CH_3_CN, reflux, 2.5 h, 77%; (ii): 6 equiv. LiCl or 5 equiv. LiBr, DMF, 100 °C, 24 h; (iii): 2.35 equiv. KO*t*-Bu, MeOH, EtOH, reflux, 10 min. from **112a** 95%, **112b**, 50%; (iv): A. from **105**, 85% formic acid, 30% H_2_O_2_, 40 °C, 25 min, 25%; B. from **105**, 10% formic acid, 90% *m*-chloroperbenzoic acid, EtOH, 0 °C, 105 min, 53%; C. from **105**, 2 equiv. disuccinyl peroxide, 10% formic acid, room temperature, 15 min., 32%; (v): 5 equiv. NaN_3_, dioxane, H_2_O, 100 °C, 10 h, 38–77%; (vi): 85% hydrazine hydrate, EtOH, reflux, 10 min, 28–63%; (vii): 10% AcOH, 70 °C, 1.5 h, 36–83%; (viii): 10% KOH solution (aq.), EtOH, reflux, 10 min, 36–90%.

**Figure 45 ijms-26-02736-f045:**
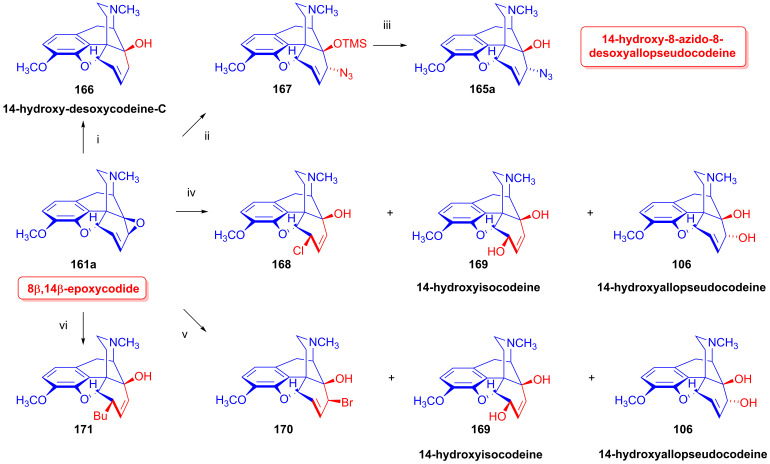
Investigations of the reactions of 8β,14β-epoxycodide. *Reagents and conditions*: (i): 1 equiv. LiAlH_4_, Et_2_O, reflux, 30 min, 50%; (ii): 3 equiv. trimethylsilyl azide, ZnI_2_, CH_2_Cl_2_, reflux, 90 min, 60%; (iii): 3 equiv. NaF, MeOH, reflux, 2 h, 67%; (iv): LiCl, 10% HCl, acetonitrile, reflux, 2.5 h, **168** (20%), Σ (**169** + **106**) (12%), recovered **161a** (22%); (v): LiBr, 10% HBr, acetonitrile, 100 °C, 2 h, **170** (31%), Σ (**169** + **106**) (26%), recovered **161a** (15%); (vi): copper (I) iodide, BuLi, diethyl ether, hexane, N_2_ atmosphere, −78 °C, 21%.

**Figure 46 ijms-26-02736-f046:**
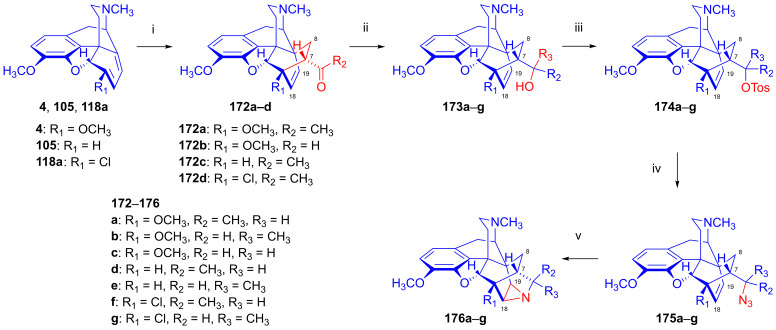
Synthesis of morphinans with an azatetracyclodecane ring system. *Reagents and conditions*: (i): methyl vinyl ketone or acrolein, toluene, reflux, 24 h; (ii): 4.5 equiv. NaBH_4_, MeOH, 0 °C, 30 min; (iii): 1.5 equiv. TosCl, pyridine room temperature, 1 day for **173d**,**e** and 6 days for **173f**,**g**; (iv): 5 equiv. NaN_3_, *N*,*N*-dimethylformamide, H_2_O, 100 °C, 1.5 h; (iv and v): 5 equiv. NaN_3_, *N*,*N*-dimethylformamide, H_2_O, 100 °C, 24 h.

**Figure 47 ijms-26-02736-f047:**
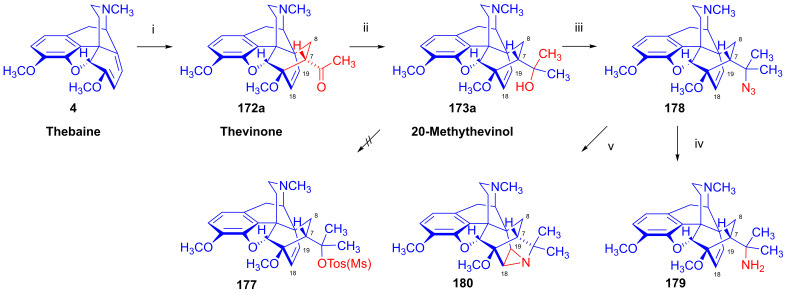
Synthesis of azides in the 6,14-ethenomorphinan series. *Reagents and conditions*: (i): methyl vinyl ketone, reflux, 1 h; (ii): MeMgI, toluene, THF, reflux, 1 h; (iii): 2 equiv. hydrazoic acid, 2 equiv. BF_3_-etherate, room temperature, 4 h, 62%; (iv): 98% hydrazine hydrate, Raney-Ni, EtOH, reflux, 30 min, 83%; (v): Δ (-N_2_), DMF, 100 °C, 40 h, 64%.

**Figure 48 ijms-26-02736-f048:**
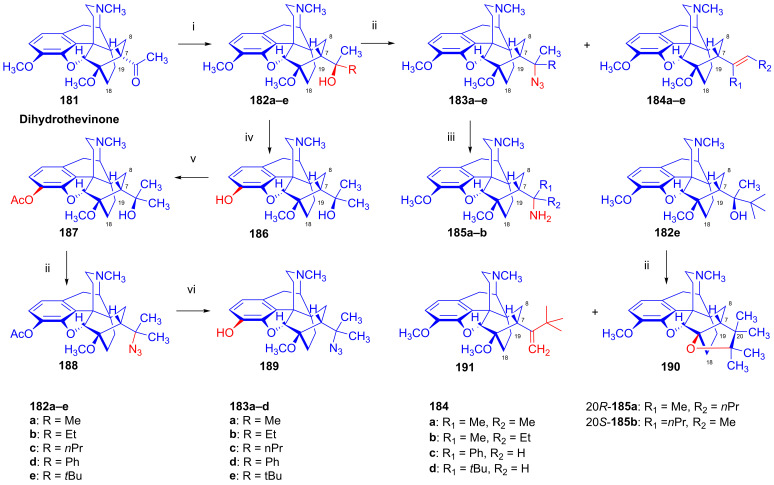
Synthesis azides of 18,19-dihydro derivatives in the 6,14-ethenomorphinan series. *Reagents and conditions*: (i): Grignard reagents (RMgX: MeMgI, EtMgBr, *n*PrMgBr, PhMgBr, *t*BuMgCl), Et_2_O–toluene or THF–toluene mixtures, reflux, 1–2 h; (ii): 2 equiv. HN_3_, 2 equiv. BF_3_-etherate, toluene, room temperature, 4 h, **183b**–**d**: 35–44% and **184b**–**d**: 5–13%; (iii): from **183c**, 8 equiv. hydrazine hydrate, Raney-Ni, ethanol, reflux, 30 min, 20*S*-**185a**: 58%, 20*R*-**185b**: 18%; (iv): KOH, diethylene glycol, 210 °C, 90 min; (v): based on Welsh’s [99] method: Ac_2_O, NaHCO_3_, H_2_O; (vi): 3 equiv. hydroxylamine hydrochloride, 50 °C, 30 min, 79%.

**Figure 49 ijms-26-02736-f049:**
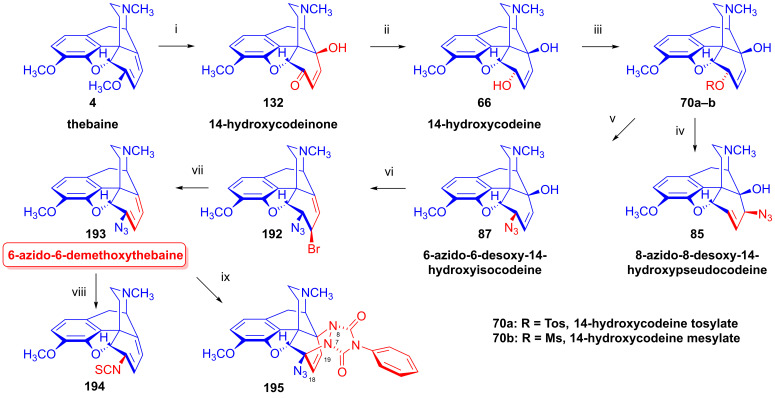
Synthesis of 6-azido-6-demethoxythebaine. *Reagents and conditions*: (i): formic aid, H_2_O_2_, 40 °C, 2 h; (ii): NaBH_4_, H_2_O, dioxane; (iii): A. TosCl, pyridine, 0 °C, [152,153], or MsCl, pyridine, 66% [121,122]; (iv): 1.25 equiv. NaN_3_, DMF, H_2_O, 100 °C, 4 h, 38% [121,122]; (v): 1.25 equiv. NaN_3_, DMF, H_2_O, 100 °C, 8 h, 46% [121,122]; (vi): 2 equiv. phosphorus tribromide, CHCl_3_, 50 °C, 2 h, 77%; (vii): 3.5 equiv. potassium *tert*-butoxide, EtOH, room temperature, 0.5 h, 75%; (viii): carbon disulfide, triphenylphosphine, reflux, 2 h, 51%; (ix): 1.2 equiv. 4-phenyl-4*H*-1,2,4-triazoline-3,5-dione (PTAD), acetone, room temperature, 20 min, 71%.

**Figure 50 ijms-26-02736-f050:**
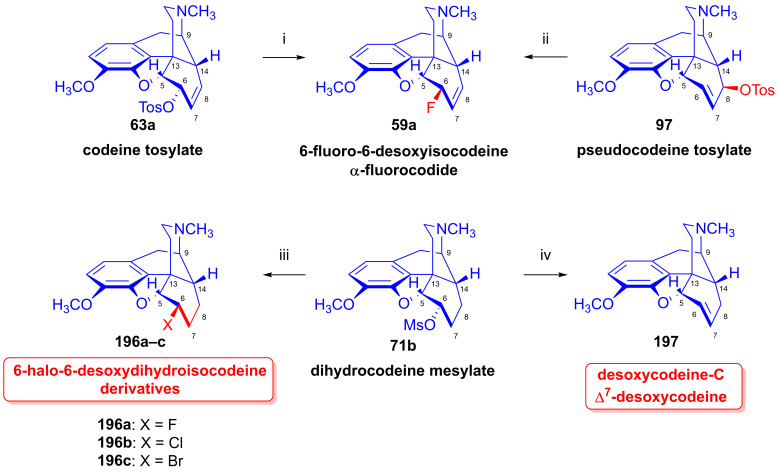
Synthesis of 6-halogene-substituted morphinans. *Reagents and conditions*: (i): 1.6 equiv. Bu_4_NF, acetonitrile, reflux, 4 h, 48% [111,112]; 5.4 equiv. Bu_4_NF, CH_3_CN, reflux, 29 h, 10% [129,130]; (iii) [239,240]: **A** (Nu = F^⊖^): 13 equiv. Bu_4_NF, acetonitrile, reflux, 12 h, **196a**, 8%; **B** (Nu = Cl^⊖^): 10 equiv. LiCl, DMF, reflux, 1 h, **196b**, 75%; **C** (Nu = Br^⊖^): 4 equiv. LiBr, 100 °C, 43 h, **196c**, 26%; (iv) [239,240]: **A** (Nu = Br^⊖^): 8 equiv. LiBr, DMF, reflux, 4 h, **197**, 63%; **B** (Nu = I^⊖^): 10 equiv. NaI, DMF, reflux, 7 h, **197**, 29%.

**Figure 51 ijms-26-02736-f051:**
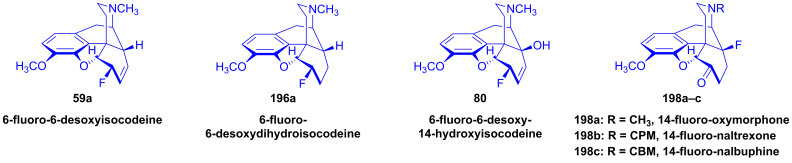
Ring-C fluorinated morphinan derivatives.

**Figure 52 ijms-26-02736-f052:**
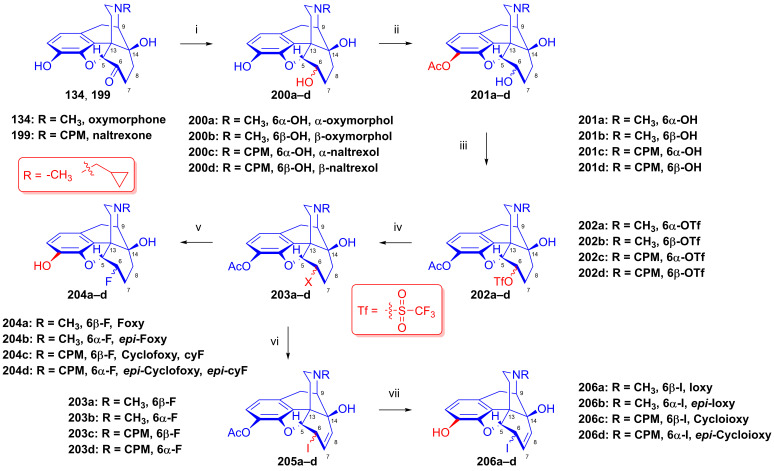
Synthesis of foxy, cyclofoxy and their iodinated derivatives. *Reagents and conditions*: (i): 0.5 equiv. NaBH_4_, THF, 0 °C, 1 h, 37%; (ii): Ac_2_O, NaHCO_3_, H_2_O, quant.; (iii): 2 equiv. Tf_2_O, CHCl_3_, pyridine, 20 °C, 20 min, 72%; (iv): 7.2 equiv. KF, 18-crown-6 ether, acetonitrile, reflux, 2 min, 62%. (v): NH_4_OH, MeOH, 80 °C, 30 min; (vi): Et_4_NI, acetonitrile, 80 °C, 4 h, argon atmosphere; (vii) NH_4_OH, acetonitrile, 25 °C, 20 min.

**Figure 53 ijms-26-02736-f053:**
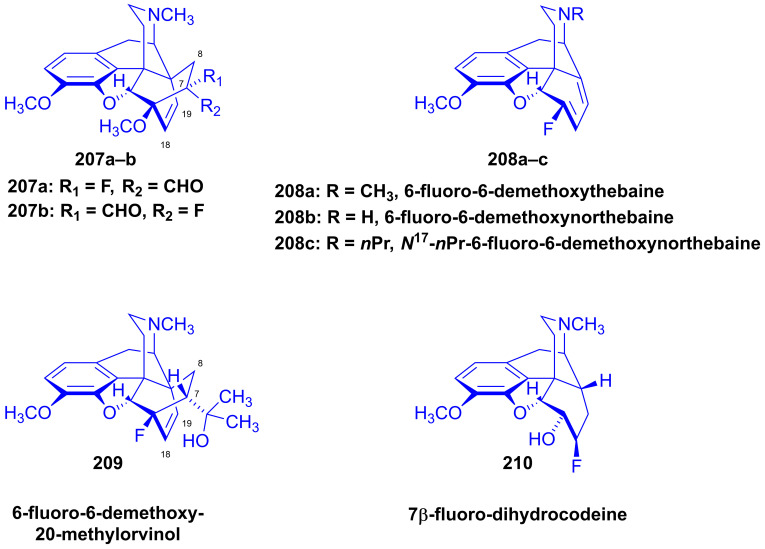
Structures of fluorinated morphinans.

**Figure 54 ijms-26-02736-f054:**
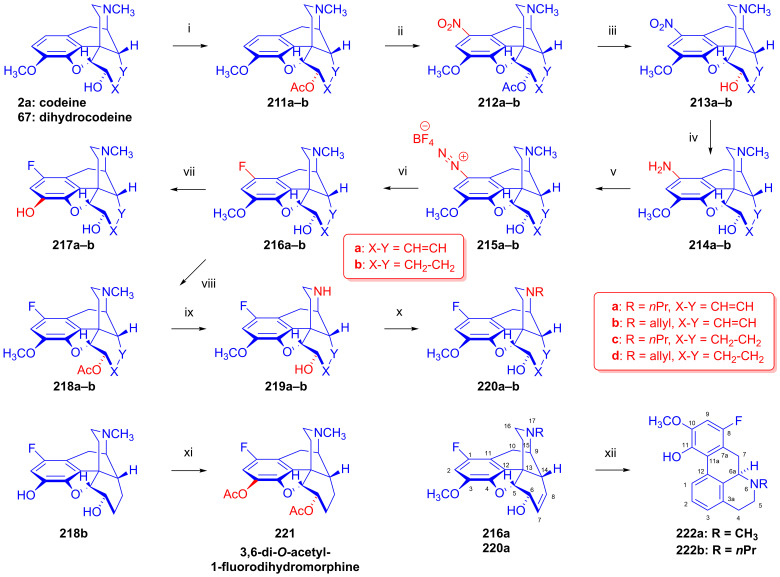
Synthesis of 1-fluoro-substituted codeine derivatives. *Reagents and conditions*: (i): Ac_2_O, reflux; (ii): 65% nitric acid, glacial acetic acid, **A**. 5 °C, 1 h, **B**. room temperature, 1 h; (iii): 10% NaOH, EtOH, reflux, 1 h; (iv): SnCl_2_, HCl, room temperature, 2 h; (v): 48% aqueous HBF_4_, EtOH, 2 M NaNO_2_, −15 °C 10 min; (vi): MgO, 170 °C, 1 h, in vacuo; (vii): BBr_3_, CHCl_3_, room temperature, 1 h; (viii): Ac_2_O, reflux, 4 h; (ix): **A**. ACE-Cl, NaHCO_3_, 1,2-dichloroethane, reflux, 8 h; **B**. MeOH, 50 °C, 1 h; (x): *n*-propyl bromide or allyl bromide, NaHCO_3_, DMF, 95 °C, 18 h; (xi): Ac_2_O, 100 °C, 1 h; (xii): methane sulfonic acid, 95 °C, 45 min.

**Figure 55 ijms-26-02736-f055:**
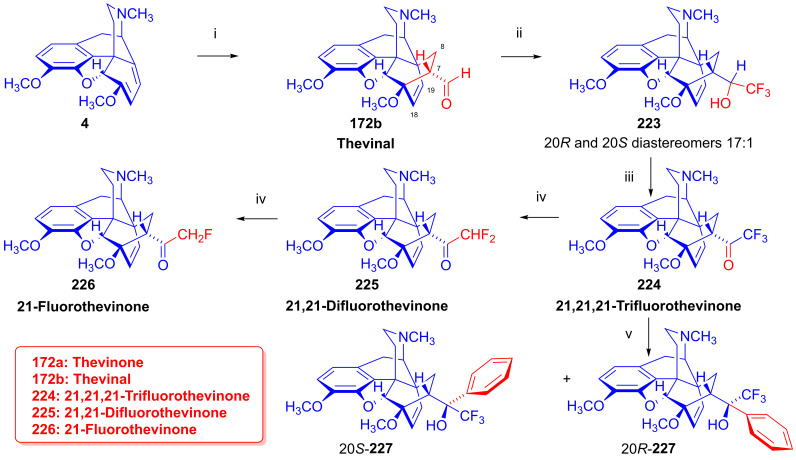
Synthesis of fluorinated 6,14-ethenomorphinan derivatives. *Reagents and conditions*: (i): acrolein, benzene, reflux; (ii): **A**. Me_3_SiCF_3_, TBAF, THF, **B**. HCl, H_2_O; (iii): **A**. oxalyl chloride, DMSO, CH_2_Cl_2_, −70 °C, 90 min, **B**. Et_3_N, 20 °C, 10 min; (iv): **A**. Mg, Me_3_SiCl, -5 °C, 4 h, B. 15% HCl, 1 h; (v): PhMgPr, THF, 20 °C, 24 h.

**Figure 56 ijms-26-02736-f056:**
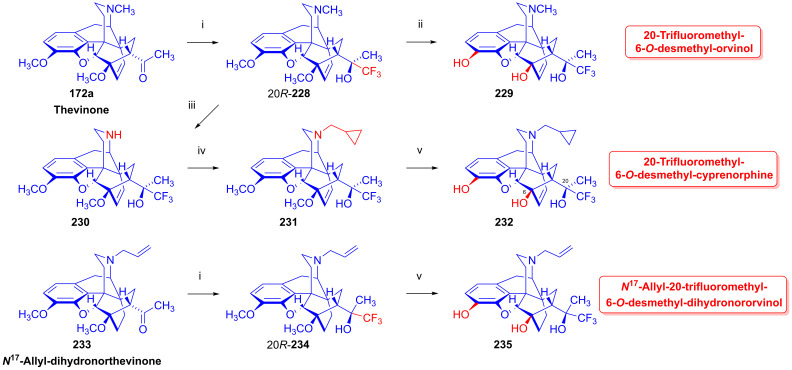
Synthesis of 21,21,21-trifluoro-6-*O*-desmethyl-orvinols. *Reagents and conditions*: (i): 2 equiv. MeSiCF_3_, (ii): 48% HBr, reflux, 30 min, 60%; (iii): **A**. 1.5 equiv. DEAD, benzene, reflux, 24 h, **B**. 1.6 equiv. pyridinium hydrochloride, MeOH, 30 min, 31%; (iv): **A**. 2.5 equiv. cyclopropanecarbonyl chloride, K_2_CO_3_, CH_2_Cl_2_, 20 °C, 6 h, **B**. 4.9 equiv. LiAlH_4_, THF, reflux, 1 h, 65%; (v): BBr_3_, CH_2_Cl_2_, -78 °C, 76%;.

**Figure 57 ijms-26-02736-f057:**
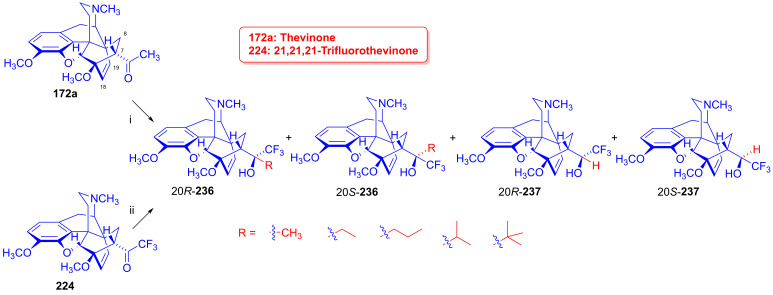
Reaction of thevinones with organometallic reagents. *Reagents and conditions*: (i): Me_3_SiCF_3_, HCl, H_2_O; (ii): **A**. Grignard reagents (MeMgI, EtMgBr, *n*BuMgBr, *i*PrMgBr, *t*BuMgCl), Et_2_O or THF, 20 °C, 15 min–18 h; **B**. RLi (MeLi, *i*PrLi, *t*BuLi), Et_2_O or THF, −78–20 °C, 15 min–60 h.

**Figure 58 ijms-26-02736-f058:**
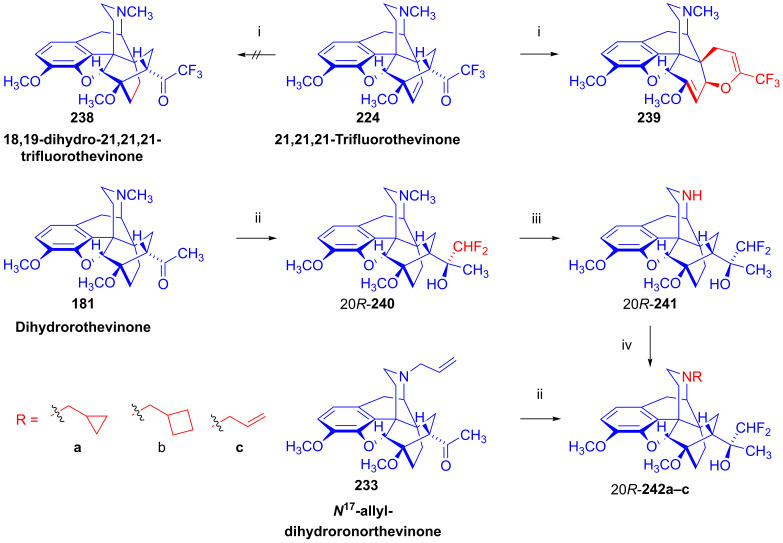
Synthesis of 21,21-difluorothevinol derivatives. *Reagents and conditions*: (i): H_2_, 10% Pd-C, 60 Bar, 55–60 °C, 45 h, 11%; (ii): **A**. Me_3_SiCHF_2_, CsF, HMPA, DMPU, THF, reflux. 10 h, **B**. 15% HCl (aq.), room temperature, 1 h, 20*R*-**240**, (35%) and 20*R*-**242c** from **233** (23%); (iii) **A**. 2 equiv. DEAD, acetonitrile, reflux, 5 h, **B**. 3 equiv. pyridinium hydrochloride, 77%; (iv): **A**. allyl bromide, NaHCO_3_, DMF, 90–95 °C, 20 h, 48%; **B**. (1) cyclopropanecarbonyl chloride, K_2_CO_3_, CH_2_Cl_2_, 20 °C, 2 h, (2) LiAlH_4_, THF, reflux, 1 h, 52%, **C**. (1) cyclobutanecarbonyl chloride, K_2_CO_3_, CH_2_Cl_2_, 20 °C, 2 h, (2) LiAlH_4_, THF reflux, 2 h, 50%.

**Figure 59 ijms-26-02736-f059:**
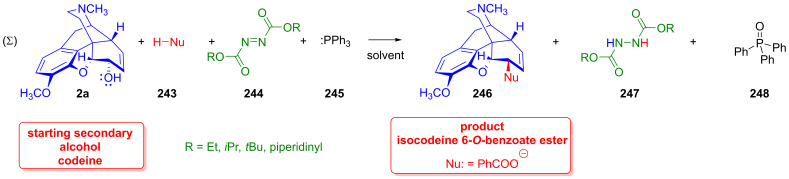
Overall reaction equation of the Mitsunobu reaction of codeine with H-Nu.

**Figure 60 ijms-26-02736-f060:**
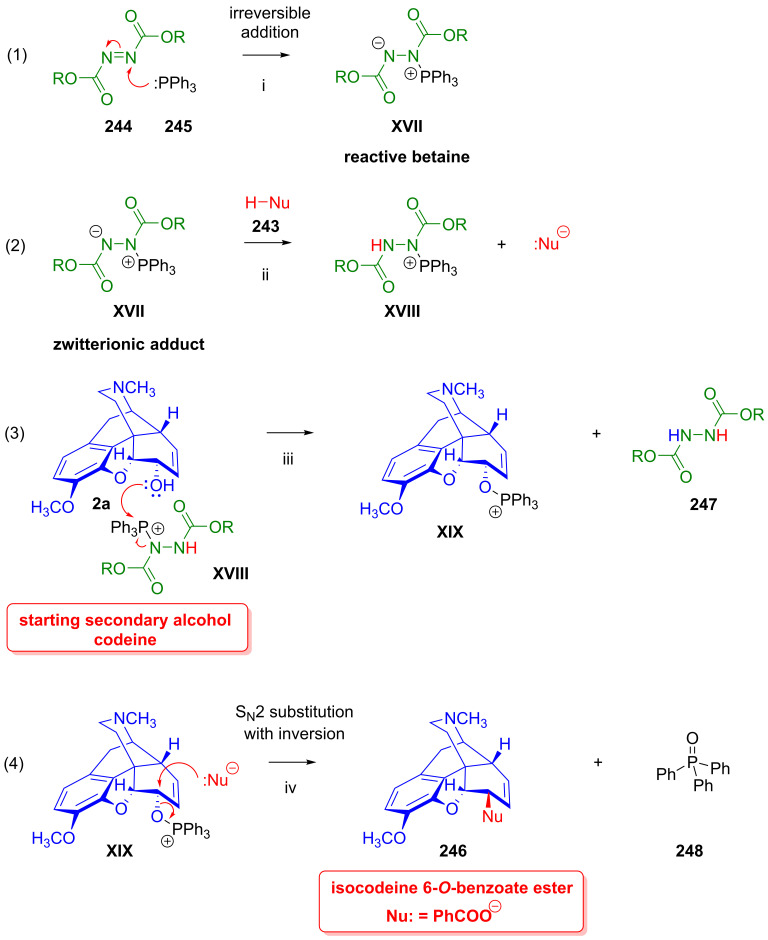
Elementary steps of the Mitsunobu reaction as exemplified by the interaction of codeine R-OH, **2a**) with benzoic acid (H-Nu, **243**).

**Figure 61 ijms-26-02736-f061:**
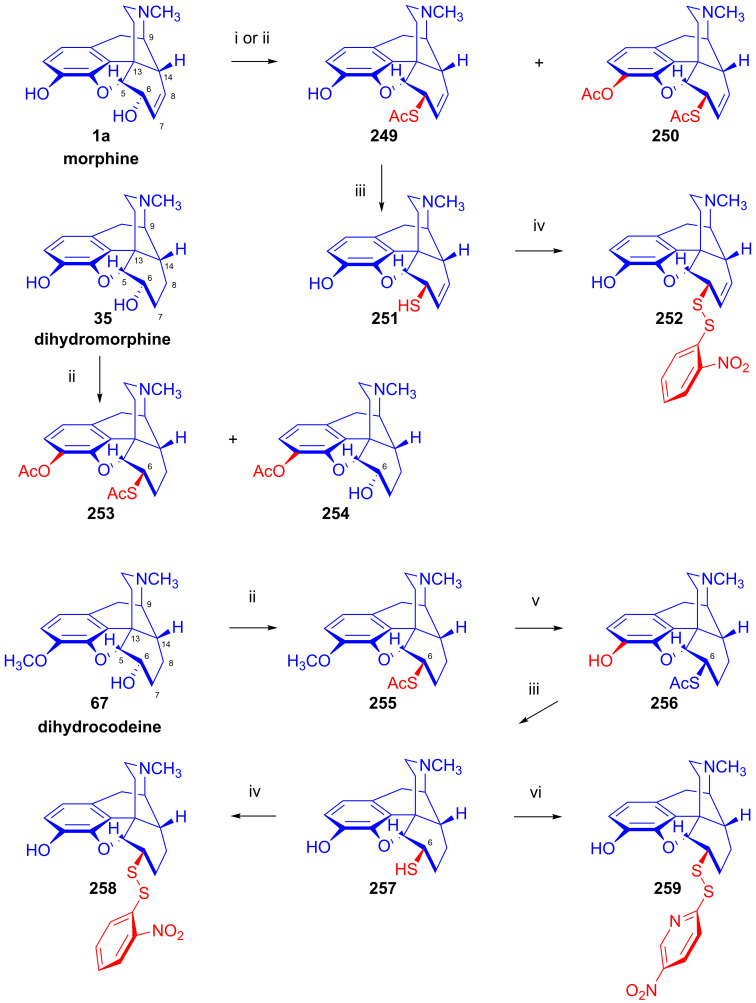
Reaction of morphine, dihydromorphine and dihydrocodeine with thioacetic acid under Mitsunobu conditions. *Reagents and conditions*: (i): thioacetic acid, *N*,*N*-dimethylformamide dineopentyl acetal, toluene, 80 °C; (ii): thioacetic acid, DIAD, TPP, THF, 0 °C; (iii): 0.2 M KOH, ethanol, N_2_; (iv): 2-nitrobenzenesulfenyl chloride, acetonitrile or CHCl_3_, 0 °C; (v): BBr_3_, CHCl_3_, 86% (vi): 5-nitro-2-pyridinesulfenyl chloride, CH_2_Cl_2_, 0 °C, 84%.

**Figure 62 ijms-26-02736-f062:**
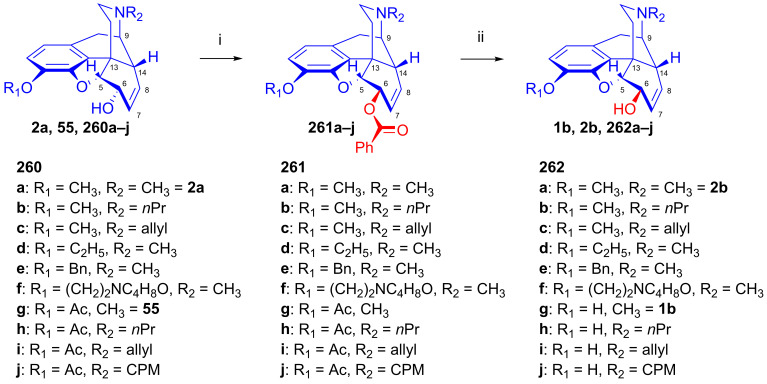
Synthesis of isocodeine and isomorphine derivatives via Mitsunobu reaction. *Reagents and conditions*: (i): 2 equiv. benzoic acid, 2 equiv. DEAD, 2 equiv. TPP, benzene, room temperature, 1 h, 29–75%; (ii): 10% KOH (aq.), ethanol, reflux, 10 min, 29–100%.

**Figure 63 ijms-26-02736-f063:**
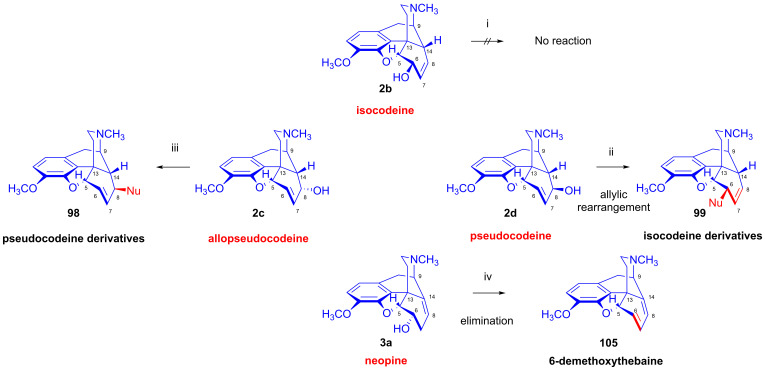
Reaction of isocodeine, allopseudocodeine, pseudocodeine and neopine with various reagents under Mitsunobu conditions. *Reagents and conditions*: (i): H-Nu = benzoic acid or phthalimide, DEAD, TPP; (ii): H-Nu = 4-NO_2_-benzoic acid or phthalimide, DEAD, TPP; (iii): **A**: when H-Nu = benzoic acid or 4-NO_2_-benzoic acid, DEAD, TPP, no reaction was observed, **B**: when H-Nu = phthalimide, DEAD, TPP, formation of 8β-phthalimido-derivative was observed; (iv): **A**: H-Nu = benzoic acid or phthalimide, DEAD, TPP; **B**: DEAD, TPP.

**Figure 64 ijms-26-02736-f064:**
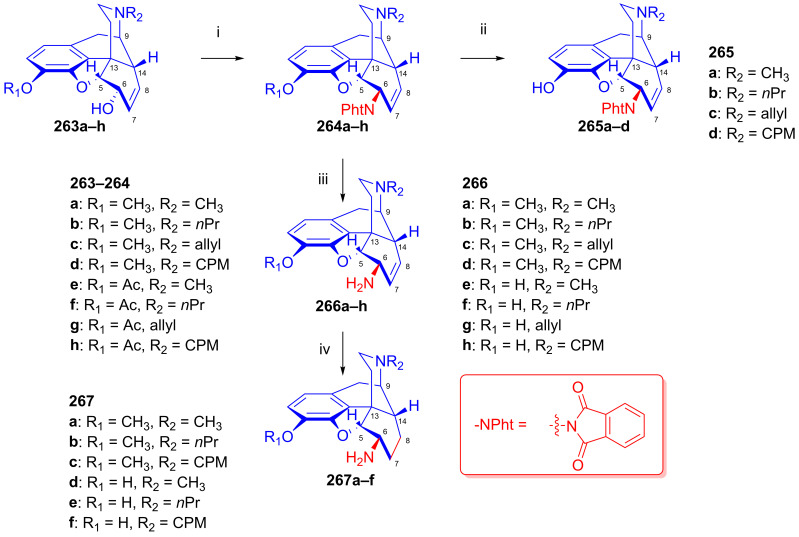
Synthesis of *N*^17^-substituted-6β-aminocodeine and 6β-aminomorphine derivatives. *Reagents and conditions*: (i): 2 equiv. phthalimide, 2 equiv. DEAD, 2 equiv. TPP, benzene, 39–92%; (ii): hydroxylamine hydrochloride, H_2_O, EtOH, 50 °C, 10 min; 35–79%; (iii): 98% hydrazine hydrate, EtOH, 50–98%; (iv): H_2_, 10% Pd-C, EtOH, 1 Bar.

**Figure 65 ijms-26-02736-f065:**
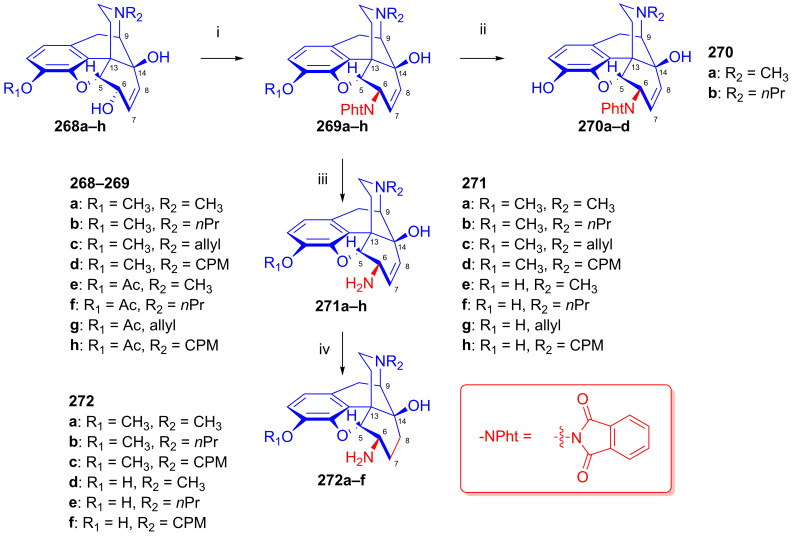
Synthesis of *N*^17^-substituted-6β-amino-14-hydroxycodeine and 6β-amino-14-hydroxymorphine derivatives. *Reagents and conditions*: (i): 2 equiv. phthalimide, 2 equiv. DEAD, 2 equiv. TPP, anhydrous benzene, RT, 1 h, 30–65%; (ii): hydroxylamine hydrochloride, H_2_O, EtOH, 50 °C, 10 min; 20–74%; (iii): 98% hydrazine hydrate, EtOH, 10–90%; (iv): H_2_, 10% Pd-C, EtOH, 1 Bar.

**Figure 66 ijms-26-02736-f066:**
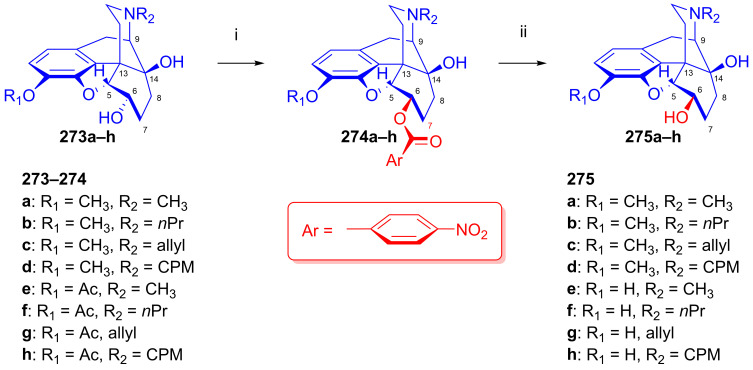
Synthesis of 14-hydroxydihydroisocodeine and 14-hydroxyisomorphine derivatives. *Reagents and conditions*: (i): 2 equiv. 4-nitrobenzoic acid, 2 equiv. TPP, 2 equiv. DEAD, anhydrous benzene, RT, 1 h, 33–79%; (ii): 10% KOH aqueous solution, EtOH, reflux, 10 min, 37–90%.

**Figure 67 ijms-26-02736-f067:**
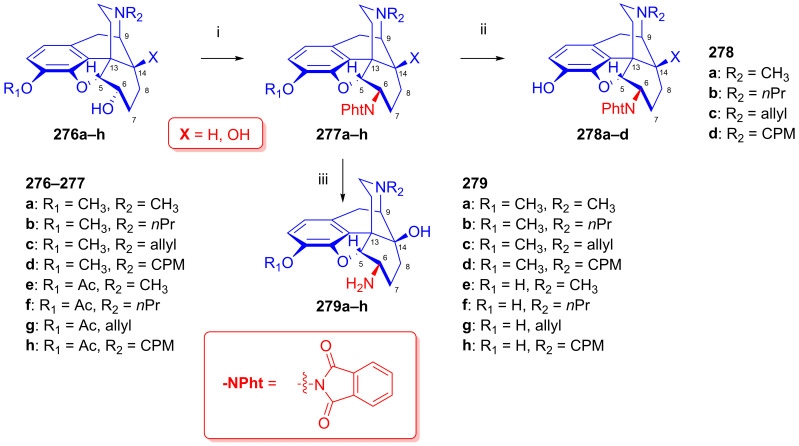
Synthesis of *N*^17^-substituted-6β-amino-14-hydroxydihydrocodeine and 6β-amino-14-hydroxydihydromorphine derivatives. *Reagents and conditions*: (i): 2 equiv. phthalimide, 2 equiv. DEAD, 2 equiv. TPP, anhydrous benzene, RT, 1 h, 33–90%; (ii): hydroxylamine hydrochloride, H_2_O, EtOH, 50 °C, 10 min; 25–70%; (iii): 98% hydrazine hydrate, EtOH, 35–98%.

**Figure 68 ijms-26-02736-f068:**
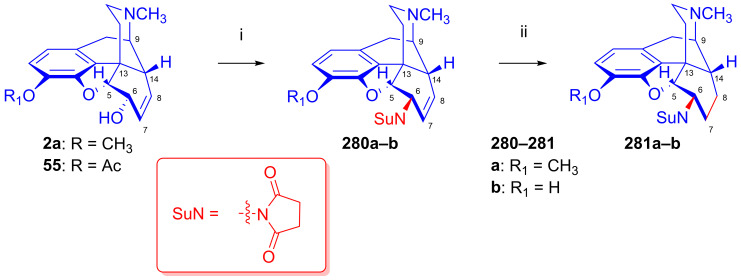
Synthesis of 6β-succinimido codeine and morphine derivatives. Reagents and conditions: (i): 2 equiv. succinimide, 2 equiv. DEAD, 2 equiv. TPP, benzene; (ii): H_2_, 10% Pd-C, EtOH, atmospheric pressure.

**Figure 69 ijms-26-02736-f069:**
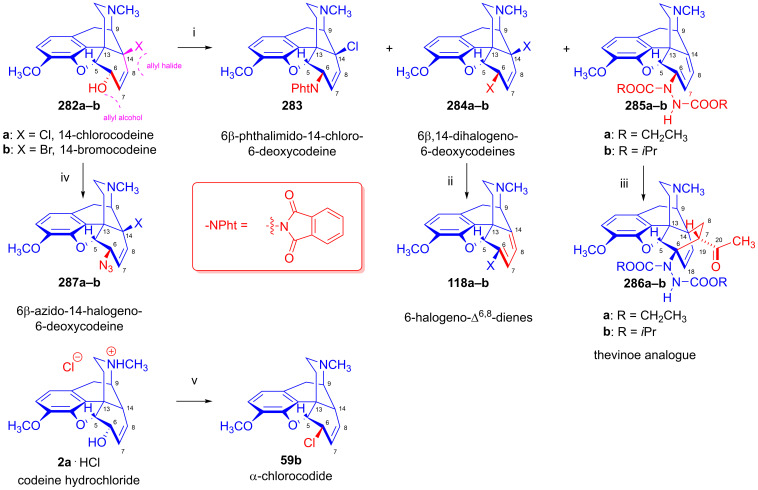
Mitsunobu reaction of 14-halogenocodeines with phthalimide and diphenylphosphoryl azide. *Reagents and conditions*: (i): phthalimide, DEAD, TPP, benzene, RT, 1 h; (ii): *N*,*N*-dimethylformamide, 100 °C, 1 h, [155]; (iii): methyl vinyl ketone, reflux, 1 h; (iv): DPPA, DEAD or DIAD, TPP; (v): DEAD, TPP, benzene, RT, 1 h, 70%.

**Figure 70 ijms-26-02736-f070:**
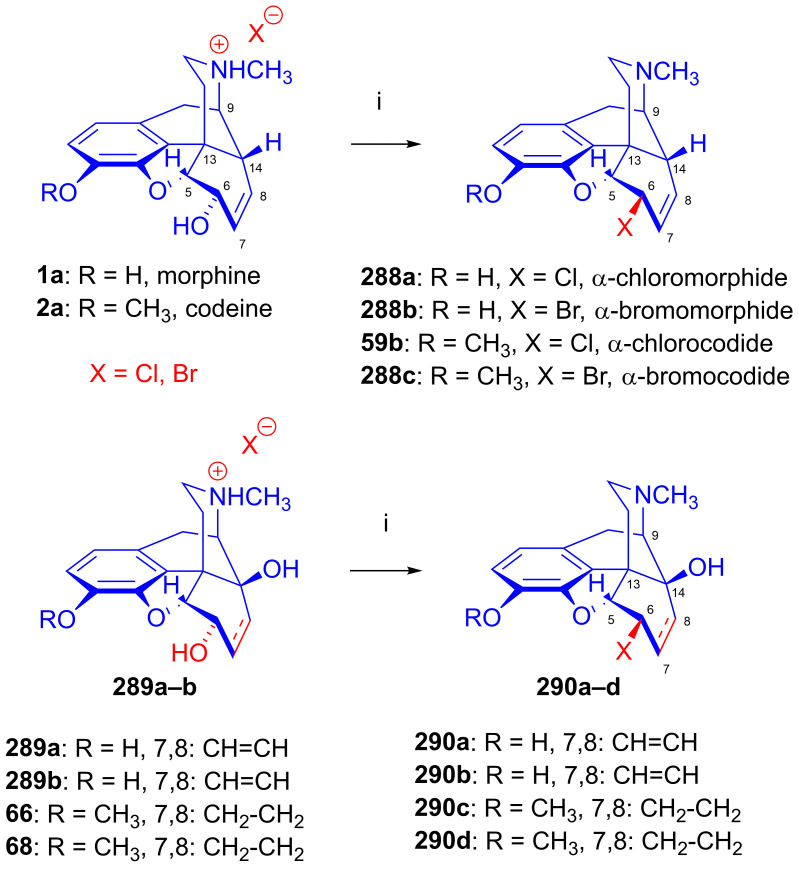
Reaction of the hydrogen halide salts of morphinans under Mitsunobu conditions. *Reagents and conditions*: (i): DEAD or DIAD, TPP, toluene or benzene, room temperature, 1–4 h.

**Figure 71 ijms-26-02736-f071:**
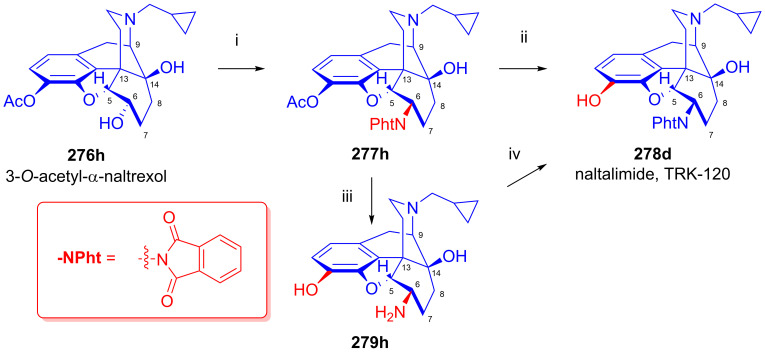
Synthesis of naltalimide. *Reagents and conditions*: (i): 2 equiv. phthalimide, 2 equiv. DEAD, 2 equiv. TPP, anhydrous benzene, RT, 1 h; (ii): hydroxylamine hydrochloride, H_2_O, EtOH, 50 °C, 10 min; (iii): 98% hydrazine hydrate, EtOH; (iv): phthalic anhydride, Et_3_N, DMF; 140 °C, 4h.

**Figure 72 ijms-26-02736-f072:**
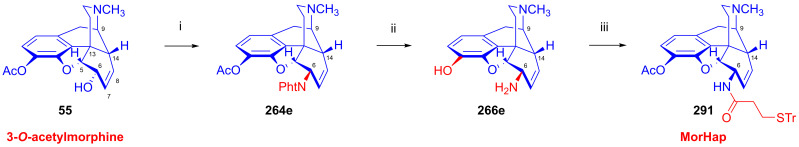
Synthesis of heroin vaccine hapten (MorHap) from 6β-amino-desoxymorphine. *Reagents and conditions*: (i): phthalimide, DIAD, TPP, toluene, room temperature, 2 h, 88%; (ii): hydrazine hydrate, 95% EtOH, 55 °C, 90 min, 81%; (iii): 1.5 equiv. 2,5-dioxopyrrolidin-1-yl 3-(tritylthio)propanoate, CH_2_Cl_2_, 2 equiv. Et_3_N, room temperature 48 h, 55%.

**Figure 73 ijms-26-02736-f073:**
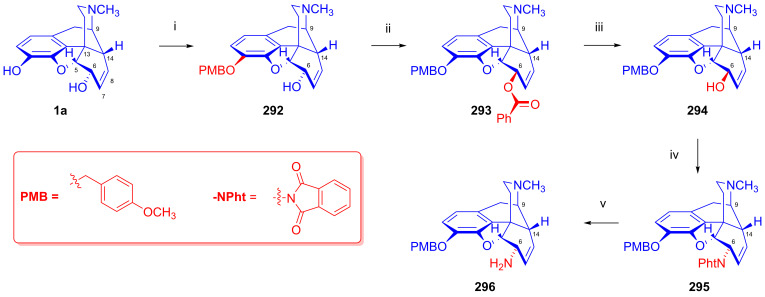
Precursors for the synthesis of fluorescently labeled morphinan-sulfo-Cy5 conjugates. *Reagents and conditions*: (i): PMB-Cl, 3M KOH, DMF, MeOH, room temperature, 4 h, 46%; (ii): 1.5 equiv. benzoic acid, 1.5 equiv. DIAD, 1.5 equiv. TPP, toluene, 0 °C to room temperature, 4 h; (iii): 1M potassium hydroxide, EtOH, H_2_O, reflux, 20 min, 78% (ii + iii); (iv): 1.5 equiv. phthalimide, 1.5 equiv. DIAD, 1.5 equiv. TPP, toluene, 0 °C to room temperature, 4 h; (v): hydrazine hydrate, EtOH, reflux, 1 h, 34% (iv + v).

**Figure 74 ijms-26-02736-f074:**
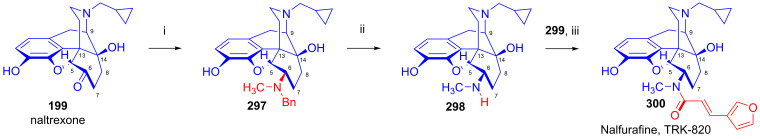
Synthesis of nalfurafine (TRK-820) a selective κ-opioid receptor agonist. *Reagents and conditions*: (i): **A**. *N*-benzylmethylamine, benzoic acid, benzene, 110 °C, 8 h, **B**. NaBH_3_CN, room temperature, 2 h; (ii): H_2_, 10% Pd-C, EtOH, atmospheric pressure, room temperature, 24 h; (iii): (*E*)-3-(3-furyl)acrylic acid (**299**), Et_3_N, DMF, CH_2_Cl_2_, room temperature, 16 h.

**Figure 75 ijms-26-02736-f075:**
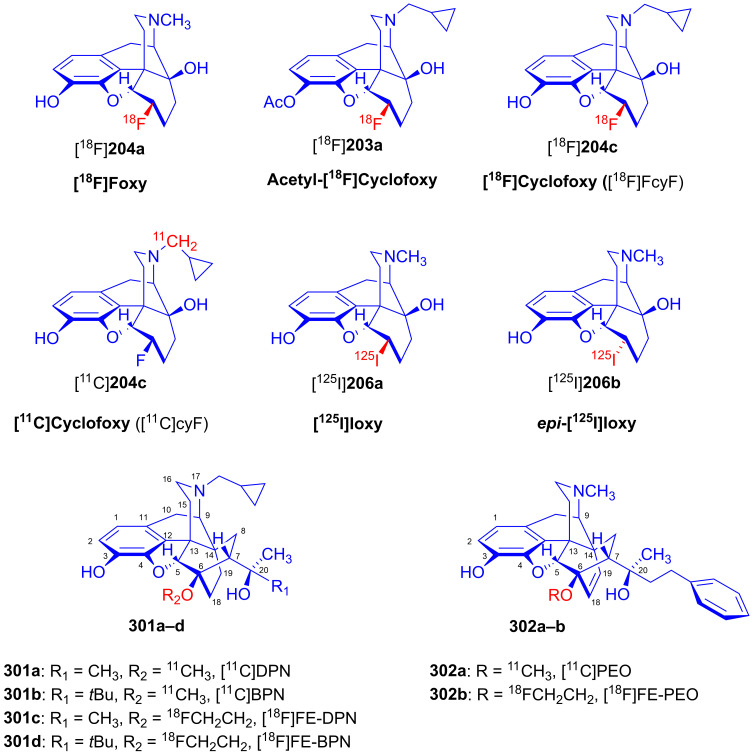
Structures of selected labeled morphinan derivatives.

**Table 1 ijms-26-02736-t001:** Absolute configuration of the selected morphinan alkaloids and their semisynthetic derivatives depicted in Figure 4.

Comp.	Name	Position of the C=C Double Bond	Position					
			5	6	8	9	13	14
**1a**	morphine	Δ^7,8^	5*R*	6*S*	-	9*R*	13*S*	14*R*
**1b**	isomorphine *	Δ^7,8^	5*R*	6*R*	-	9*R*	13*S*	14*R*
**1c**	allopseudomorphine **	Δ^6,7^	5*S*	-	8*R*	9*R*	13*S*	14*R*
**1d**	γ-isomorphine	Δ^6,7^	5*S*	-	8*S*	9*R*	13*S*	14*R*
**2a**	codeine	Δ^7,8^	5*R*	6*S*	-	9*R*	13*S*	14*R*
**2b**	isocodeine	Δ^7,8^	5*R*	6*R*	-	9*R*	13*S*	14*R*
**2c**	allopseudocodeine	Δ^6,7^	5*S*	-	8*R*	9*R*	13*S*	14*R*
**2d**	pseudocodeine	Δ^6,7^	5*S*	-	8*S*	9*R*	13*S*	14*R*
**3a**	neopine	Δ^8,14^	5*R*	6*S*	-	9*R*	13*S*	-
**3b**	isoneopine	Δ^8,14^	5*R*	6*R*	-	9*R*	13*S*	-
**3c**	neomorphine	Δ^8,14^	5*R*	6*S*	-	9*R*	13*S*	-
**3d**	isoneomorphine	Δ^8,14^	5*R*	6*R*	-	9*R*	13*S*	-
**4**	thebaine	Δ^6,7^ and Δ^8,14^	5*R*	-	-	9*R*	13*S*	-
**5**	oripavine	Δ^6,7^ and Δ^8,14^	5*R*	-	-	9*R*	13*S*	-

* Isomorphine: known also as α-isomorphine; ** allopseudomorphine: known also as β-isomorphine.

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
