# Peer review of "Morphinan Alkaloids and Their Transformations: A Historical Perspective of a Century of Opioid Research in Hungaryâ€"

_ijms, 2025, doi:10.3390/ijms26062736_

Round 1
Reviewer 1 Report
Comments and Suggestions for Authors
In this review, the authors describe in great detail the chemical modifications of alkaloids derived from morphinan. It should be emphasised that the entire work is written and edited very carefully. Contains an extensive introduction to the history of the discovery and description of the first opium alkaloids. Of course, the topic is very interesting from the point of view of chemistry, medicine and pharmacognosy. However, it seems to me quite a niche and such a detailed and extensive description of morphinan derivatives may be of interest mainly to people dealing with this topic in their scientific work. The volume of the work and the number of references are admirable, but also make reading this publication very time-consuming, which may result in its poor reception. However, it is not my role to judge the length of work. There are a few linguistic errors in the work that should be corrected. It seems to me that some schemes or diagrams could be larger or of better quality to make them more legible. I rate the entire work highly and believe that it will be a valuable read for people particularly interested in the topic of opioids.
Author Response
Answers to referee-1
We would like to thank referee-1 for thoughtful evaluation of our manuscript, and concur with their judgment that our survey will provide a substantial aid for researchers and students who wish to deepen their knowledge/expertise in the field of opiate chemistry.
“In this review, the authors describe in great detail the chemical modifications of alkaloids derived from morphinan. It should be emphasised that the entire work is written and edited very carefully. Contains an extensive introduction to the history of the discovery and description of the first opium alkaloids. Of course, the topic is very interesting from the point of view of chemistry, medicine and pharmacognosy. However, it seems to me quite a niche and such a detailed and extensive description of morphinan derivatives may be of interest mainly to people dealing with this topic in their scientific work. The volume of the work and the number of references are admirable, but also make reading this publication very time-consuming, which may result in its poor reception. However, it is not my role to judge the length of work.
There are a few linguistic errors in the work that should be corrected.”
Answer:
Linguistical errors: Our native English-speaking co-authors (PC/KCR) have carefully proofread the revised version.
“It seems to me that some schemes or diagrams could be larger or of better quality to make them more legible.”
Answer:
Figures quality: We now provide the editors all 75 figures in the form of high-quality 720 dpi TIFF files.
“I rate the entire work highly and believe that it will be a valuable read for people particularly interested in the topic of opioids.”
We are grateful for this support and kind words.

Reviewer 2 Report
Comments and Suggestions for Authors
Dear Editor,
The present review is focusing to opiate alkaloid chemistry, with emphasis to be given on the extensive contributions in recent decades of the Hungarian research groups. In this direction, there is a historical presentation in introduction of different companies and research groups since 1926 till recently. Thereafter, authors presented the stereochemistry, and the biosynthesis of morphinan alkaloids.
The importance of Poppy alkaloids and the stereochemistry of morphinans is not presented. Authors just described their structures.
A lot of work has been done, and a chaotic review was presented at the end. For me the importance of all these compounds is missing. I do not know if the aim of this review was just to present the chemistry and the synthesis of all these compounds.
It was mentioned that in 1968, Makleit and Bodnár [48] proposed the introduction of a new nomenclature for morphine derivatives. Ok, but I cannot see the problem and a new nomenclature was required.
It seems that a boring review has been done. The direction and the aim of this review is missing. Probably a table is needed with added compounds and their applications.
Author Response
Answers to referee-2
We are thankful for this positive evaluation, which has helped to improve our manuscript.
“The present review is focusing to opiate alkaloid chemistry, with emphasis to be given on the extensive contributions in recent decades of the Hungarian research groups. In this direction, there is a historical presentation in introduction of different companies and research groups since 1926 till recently. Thereafter, authors presented the stereochemistry, and the biosynthesis of morphinan alkaloids.”
“The importance of Poppy alkaloids and the stereochemistry of morphinans is not presented. Authors just described their structures.”
Answer: The stereochemistry of morphinans is presented in section «2.2. The stereochemistry of morphinans» (page 4–6, Line 151–199). We have inserted the following additional information:
«The morphinan scaffold contains three asymmetric centres (9,13,14). The stereochemistry of morphinan derivatives and other type of OR ligands are of fundamental importance in determining their pharmacological selectivity between the OR subtypes (μ, δ, κ, NOP) or off-target binding to other receptor types, e.g., NMDA receptors. For example, levorphan (N17-methyl-3-hydroxy-morphinan), which has an R absolute configuration at all three asymmetric carbons (9R,13R,14R, Figure 3, structure IIa), is an OR agonist with seven-fold higher affinity than morphine (1a). Dextrorphan (N17-methyl-3-hydroxy-morphinan) with 9S,13S,14S absolute configuration has a different pharmacological profile. It has a selectivity for the binding site of the NMDA glutamate receptors. Dextromethorphan ((9S,13S,14S)-N17-methyl-3-methoxy-morphinan) has little analgesic activity at μORs, but is a commonly applied cough suppressant [23]. There are also semi-synthetic 6,14-ethenomorphinans with up to seven asymmetric carbons (5,6,7,9,13,14,20), with the absolute configuration of the C-20 chiral carbon being of partic-ular importance for the OR affinity and selectivity. While the 20S version of etorphine has about 40-times higher potency than morphine (1a), 20R-etorphine surpass its diastere-omer by far, being 3200-fold that of morphine (1a).»
“A lot of work has been done, and a chaotic review was presented at the end. For me the importance of all these compounds is missing. I do not know if the aim of this review was just to present the chemistry and the synthesis of all these compounds.”
Answer: Our intention in writing this manuscript was to convey and commemorate the outstanding past activities of the «Alkaloida Chemical Company» and members of the morphine-alkaloid research group of the «University of Debrecen», all of whom had received their training in organic chemistry (please see «Supplementary Figure 1»). Due to this history, we focused our survey mainly on the organic chemical transformations of poppy alkaloids. On the other hand, we structured our review to follow in chronological sequence the studies of this research group over the decades. We would like to elaborate the logic of the organisation of our paper using as an example section «2.2. Nucleophilic substitution reaction in the morphine series». We grouped the substrates based on their main structural properties: 7,8-dihydro compounds, Δ7,8-unsaturated derivatives, pseudocodeine tosylate (Δ6,7-unsaturated derivative), neopine derivatives (Δ8,14-unsaturated), alkyl mesylate and allyl halide structural units in the same molecule, etc. (please see sections 2.8.1–2.8.6 please), and report their reactions with different type of nucleophiles. As such, our organizational structure follows a strict logic.
For illustration/visualization of the molecules, we applied the method elaborated by the Maat/Delft morphine-alkaloid research group in presenting 3D structures in our manuscript. The The «Maat-visualization» clearly shows the T-shape of the morphinan skeleton in 2D, and makes a clear distinction between the morphinan and isomorphinan structures, depending on the orientation of the substituent in the 14-position. To highlight the major changes in the structures in a given synthesis step we apply different colours to impart a clearer understanding of the chemical transformation.
“It was mentioned that in 1968, Makleit and Bodnár [48] proposed the introduction of a new nomenclature for morphine derivatives. Ok, but I cannot see the problem and a new nomenclature was required.”
The Makleit-Bognár nomenclature, similar to the Maat nomenclature in the field of 6,14-ethenomorphinans, is a rational system that would allow starting from four codeine isomers (codeine (2a), isocodeine (2b), allopseudocodeine (2c), pseudocodeine (2d), Figure 4), to easily derive and rationalize all other/new in ring-C substituted morphinans. This would simplify the accessibility of chemistry and ease the memorizing of names and the corresponding stereochemistry for students. We find it regrettable that this nomenclature did not yet become videly adopted.
“It seems that a boring review has been done. The direction and the aim of this review is missing. Probably a table is needed with added compounds and their applications.”
Our review depicts chemical research in Hungary over a span of several decades during which the standards for pharmacological evaluation were rapidly evolving. Unfortunately, key pharmacological data is frequently missing, especialy from early studies preceding the pharmacological and genetic characterization of the several receptor types. As such, we fear that the table proposed by the reviewer would bring little clarity.

Round 2
Reviewer 2 Report
Comments and Suggestions for Authors
Accept in the present form.